# Quantitative dissection of transcription in development yields evidence for transcription-factor-driven chromatin accessibility

Elizabeth Eck[1†], Jonathan Liu[2†], Maryam Kazemzadeh-Atoufi[3], Sydney Ghoreishi[4], Shelby A Blythe[5], Hernan G Garcia[1,2,4,6*]

[1]Biophysics Graduate Group, University of California at Berkeley, Berkeley, United States; [2]Department of Physics, University of California at Berkeley, Berkeley, United States; [3]Department of Materials Science and Engineering, Northwestern University, Evanston, United States; [4]Department of Molecular and Cell Biology, University of California at Berkeley, Berkeley, United States; [5]Department of Molecular Biosciences, Northwestern University, Evanston, United States; [6]Institute for Quantitative Biosciences-QB3, University of California at Berkeley, Berkeley, United States

**Abstract** Thermodynamic models of gene regulation can predict transcriptional regulation in bacteria, but in eukaryotes, chromatin accessibility and energy expenditure may call for a different framework. Here, we systematically tested the predictive power of models of DNA accessibility based on the Monod-Wyman-Changeux (MWC) model of allostery, which posits that chromatin fluctuates between accessible and inaccessible states. We dissected the regulatory dynamics of *hunchback* by the activator Bicoid and the pioneer-like transcription factor Zelda in living *Drosophila* embryos and showed that no thermodynamic or non-equilibrium MWC model can recapitulate *hunchback* transcription. Therefore, we explored a model where DNA accessibility is not the result of thermal fluctuations but is catalyzed by Bicoid and Zelda, possibly through histone acetylation, and found that this model can predict *hunchback* dynamics. Thus, our theory-experiment dialogue uncovered potential molecular mechanisms of transcriptional regulatory dynamics, a key step toward reaching a predictive understanding of developmental decision-making.

*For correspondence:
hggarcia@berkeley.edu

[†]These authors contributed equally to this work

Competing interests: The authors declare that no competing interests exist.

## Introduction

Over the last decade, hopeful analogies between genetic and electronic circuits have posed the challenge of predicting the output gene expression of a DNA regulatory sequence in much the same way that the output current of an electronic circuit can be predicted from its wiring diagram (*Endy, 2005*). This challenge has been met with a plethora of theoretical works, including thermodynamic models, which use equilibrium statistical mechanics to calculate the probability of finding transcription factors bound to DNA and to relate this probability to the output rate of mRNA production (*Ackers et al., 1982*; *Buchler et al., 2003*; *Vilar and Leibler, 2003*; *Bolouri and Davidson, 2003*; *Bintu et al., 2005a*; *Bintu et al., 2005b*; *Sherman and Cohen, 2012*). Thermodynamic models of bacterial transcription launched a dialogue between theory and experiments that has largely confirmed their predictive power for several operons (*Ackers et al., 1982*; *Bakk et al., 2004*; *Zeng et al., 2010*; *He et al., 2010*; *Garcia and Phillips, 2011*; *Brewster et al., 2012*; *Cui et al.,*

**eLife digest** Cells in the brain, liver and skin, as well as many other organs, all contain the same DNA, yet behave in very different ways. This is because before a gene can produce its corresponding protein, it must first be transcribed into messenger RNA. As an organism grows, the transcription of certain genes is switched on or off by regulatory molecules called transcription factors, which guide cells towards a specific 'fate'.

These molecules bind to specific locations within the regulatory regions of DNA, and for decades biologist have tried to use the arrangement of these sites to predict which proteins a cell will make. Theoretical models known as thermodynamic models have been able to successfully predict transcription in bacteria. However, this has proved more challenging to do in eukaryotes, such as yeast, fruit flies and humans.

One of the key differences is that DNA in eukaryotes is typically tightly wound into bundles called nucleosomes, which must be disentangled in order for transcription factors to access the DNA. Previous thermodynamic models have suggested that DNA in eukaryotes randomly switches between being in a wound and unwound state. The models assume that once unwound, regulatory proteins stabilize the DNA in this form, making it easier for other transcription factors to bind to the DNA.

Now, Eck, Liu et al. have tested some of these models by studying the transcription of a gene involved in the development of fruit flies. The experiments showed that no thermodynamic model could accurately mimic how this gene is regulated in the embryos of fruit flies. This led Eck, Liu et al. to identify a model that is better at predicting the activation pattern of this developmental gene. In this model, instead of just 'locking' DNA into an unwound shape, transcription factors can also actively speed up the unwinding of DNA.

This improved understanding builds towards the goal of predicting gene regulation, where DNA sequences can be used to tell where and when cell decisions will be made. In the future, this could allow the development of new types of therapies that can regulate transcription in different diseases.

*2013*; *Brewster et al., 2014*; *Sepúlveda et al., 2016*; *Razo-Mejia et al., 2018*) with a few potential exceptions (*Garcia et al., 2012*; *Hammar et al., 2014*).

Following these successes, thermodynamic models have been widely applied to eukaryotes to describe transcriptional regulation in yeast (*Segal et al., 2006*; *Gertz et al., 2009*; *Sharon et al., 2012*; *Zeigler and Cohen, 2014*), human cells (*Giorgetti et al., 2010*), and the fruit fly *Drosophila melanogaster* (*Jaeger et al., 2004a*; *Zinzen et al., 2006*; *Segal et al., 2008*; *Fakhouri et al., 2010*; *Parker et al., 2011*; *Kanodia et al., 2012*; *White et al., 2012*; *Samee et al., 2015*; *Sayal et al., 2016*). However, two key differences between bacteria and eukaryotes cast doubt on the applicability of thermodynamic models to predict transcriptional regulation in the latter. First, in eukaryotes, DNA is tightly packed in nucleosomes and must become accessible in order for transcription factor binding and transcription to occur (*Polach and Widom, 1995*; *Levine, 2010*; *Schulze and Wallrath, 2007*; *Lam et al., 2008*; *Raveh-Sadka et al., 2009*; *Li et al., 2011*; *Fussner et al., 2011*; *Bai et al., 2011*; *Li et al., 2014b*; *Hansen and O'Shea, 2015*). Second, recent reports have speculated that, unlike in bacteria, the equilibrium framework may be insufficient to account for the energy-expending steps involved in eukaryotic transcriptional regulation, such as histone modifications and nucleosome remodeling, calling for non-equilibrium models of transcriptional regulation (*Kim and O'Shea, 2008*; *Estrada et al., 2016*; *Li et al., 2018*; *Park et al., 2019*).

Recently, various theoretical models have incorporated chromatin accessibility and energy expenditure in theoretical descriptions of eukaryotic transcriptional regulation. First, models by *Mirny, 2010*, *Narula and Igoshin, 2010*, and *Marzen et al., 2013* accounted for chromatin occluding transcription-factor binding by extending thermodynamic models to incorporate the Monod-Wyman-Changeux (MWC) model of allostery (*Figure 1A*; *Monod et al., 1965*). This thermodynamic MWC model assumes that chromatin rapidly transitions between accessible and inaccessible states via thermal fluctuations, and that the binding of transcription factors to accessible DNA shifts this equilibrium toward the accessible state. Like all thermodynamic models, this model relies on the

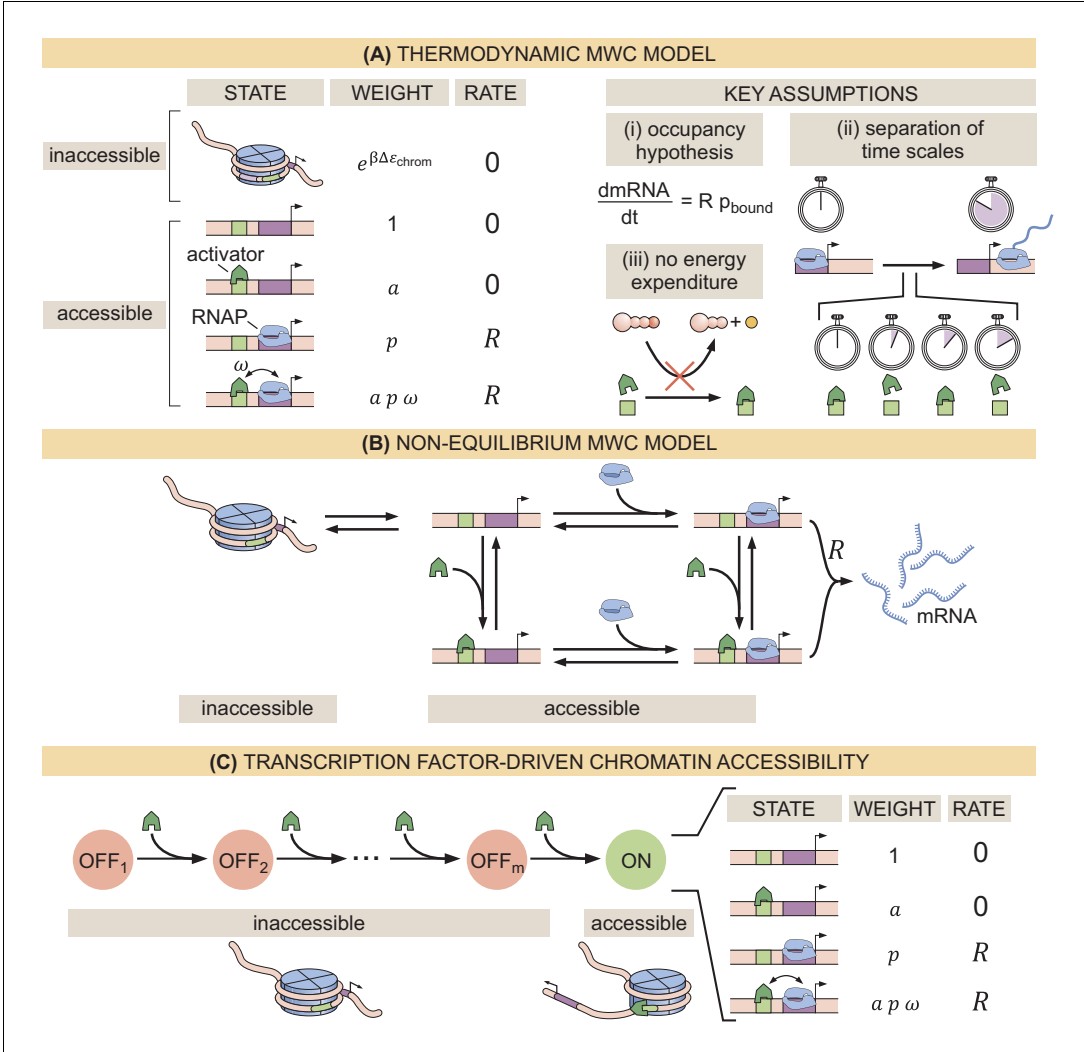

**Figure 1.** Three models of chromatin accessibility and transcriptional regulation. (**A**) Thermodynamic MWC model where chromatin can be inaccessible or accessible to transcription factor binding. Each state is associated with a statistical weight given by the Boltzmann distribution and with a rate of transcriptional initiation. $\Delta\varepsilon_{\mathrm{chrom}}$ is the energy cost associated with making the DNA accessible and $\omega$ is an interaction energy between the activator and RNAP. $a = [\mathrm{activator}]/K_a$ and $p = [\mathrm{RNAP}]/K_p$ with $K_a$ and $K_p$ being the dissociation constants of the activator and RNAP, respectively. This model assumes the occupancy hypothesis, separation of time scales, and lack of energy expenditure described in the text. (**B**) Non-equilibrium MWC model where no assumptions about separation of time scales or energy expenditure are made. Transition rates that depend on the concentration of the activator or RNAP are indicated by an arrow incorporating the respective protein. (**C**) Transcription-factor-driven chromatin accessibility model where the activator catalyzes irreversible transitions of the DNA through $m$ silent states before it becomes accessible. Once this accessible state is reached, the system is in equilibrium.

'occupancy hypothesis' (*Hammar et al., 2014*; *Garcia et al., 2012*; *Phillips et al., 2019*): the probability $p_{bound}$ of finding RNA polymerase (RNAP) bound to the promoter, a quantity that can be easily computed, is linearly related to the rate of mRNA production $\frac{\mathrm{dmRNA}}{\mathrm{d}t}$, a quantity that can be experimentally measured, such that

$$\frac{\mathrm{dmRNA}}{\mathrm{d}t} = R p_{bound}. \tag{1}$$

Here, $R$ is the rate of mRNA production when the system is in an RNAP-bound state (see Appendix section 1.1 for a more detailed overview). Additionally, in all thermodynamic models, the

transitions between states are assumed to be much faster than both the rate of transcriptional initiation and changes in transcription factor concentrations. This separation of time scales, combined with a lack of energy dissipation in the process of regulation, makes it possible to consider the states to be in equilibrium such that the probability of each state can be computed using its Boltzmann weight (*Garcia et al., 2007*).

Despite the predictive power of thermodynamic models, eukaryotic transcription may not adhere to the requirements imposed by the thermodynamic framework. Indeed, *Narula and Igoshin, 2010*, *Hammar et al., 2014*, *Estrada et al., 2016*, *Scholes et al., 2017*, and *Li et al., 2018* have proposed theoretical treatments of transcriptional regulation that maintain the occupancy hypothesis, but make no assumptions about separation of time scales or energy expenditure in the process of regulation. When combined with the MWC mechanism of DNA allostery, these models result in a non-equilibrium MWC model (*Figure 1B*). Here, no constraints are imposed on the relative values of the transition rates between states and energy can be dissipated over time. To our knowledge, neither the thermodynamic MWC model nor the non-equilibrium MWC model have been tested experimentally in eukaryotic transcriptional regulation.

Here, we performed a systematic dissection of the predictive power of these MWC models of DNA allostery in the embryonic development of the fruit fly *Drosophila melanogaster* in the context of the step-like activation of the *hunchback* gene by the Bicoid activator and the pioneer-like transcription factor Zelda (*Driever et al., 1989*; *Nien et al., 2011*; *Xu et al., 2014*). Specifically, we compared the predictions from these MWC models against dynamical measurements of input Bicoid and Zelda concentrations and output *hunchback* transcriptional activity. Using this approach, we discovered that no thermodynamic or non-equilibrium MWC model featuring the regulation of *hunchback* by Bicoid and Zelda could describe the transcriptional dynamics of this gene. Following recent reports of the regulation of *hunchback* and *snail* (*Desponds et al., 2016*; *Dufourt et al., 2018*) and inspired by discussions of non-equilibrium schemes of transcriptional regulation (*Coulon et al., 2013*; *Wong and Gunawardena, 2020*), we proposed a model in which Bicoid and Zelda, rather than passively biasing thermal fluctuations of chromatin toward the accessible state, actively assist the overcoming of an energetic barrier to make chromatin accessible through the recruitment of energy-consuming histone modifiers or chromatin remodelers. This model (*Figure 1C*) recapitulated all of our experimental observations. This interplay between theory and experiment establishes a clear path to identify the molecular steps that make DNA accessible, to systematically test our model of transcription-factor-driven chromatin accessibility, and to make progress toward a predictive understanding of transcriptional regulation in development.

## Results

### A thermodynamic MWC model of activation and chromatin accessibility by Bicoid and Zelda

During the first 2 hr of embryonic development, the *hunchback* P2 minimal enhancer (*Margolis et al., 1995*; *Driever et al., 1989*; *Perry et al., 2012*; *Park et al., 2019*) is believed to be devoid of significant input signals other than activation by Bicoid and regulation of chromatin accessibility by both Bicoid and Zelda (*Perry et al., 2012*; *Xu et al., 2014*; *Hannon et al., 2017*). As a result, the early regulation of *hunchback* provides an ideal scaffold for a stringent test of simple theoretical models of eukaryotic transcriptional regulation.

Our implementation of the thermodynamic MWC model (*Figure 1A*) in the context of *hunchback* states that in the inaccessible state, neither Bicoid nor Zelda can bind DNA. In the accessible state, DNA is unwrapped and the binding sites become accessible to these transcription factors. Due to the energetic cost of opening the chromatin ($\Delta\varepsilon_{\mathrm{chrom}}$), the accessible state is less likely to occur than the inaccessible one. However, the binding of Bicoid or Zelda can shift the equilibrium toward the accessible state (*Adams and Workman, 1995*; *Miller and Widom, 2003*; *Mirny, 2010*; *Narula and Igoshin, 2010*; *Marzen et al., 2013*).

In our model, we assume that all binding sites for a given molecular species have the same binding affinity. Relaxing this assumption does not affect any of our conclusions (as we will see below in Sections 'The thermodynamic MWC model fails to predict activation of hunchback in the absence of Zelda' and 'No thermodynamic model can recapitulate the activation of hunchback by Bicoid alone').

Bicoid upregulates transcription by recruiting RNAP through a protein-protein interaction characterized by the parameter $\omega_{bp}$. We allow cooperative protein-protein interactions between Bicoid molecules, described by $\omega_b$. However, since to our knowledge there is no evidence of direct interaction between Zelda and any other proteins, we assume no interaction between Zelda and Bicoid, or between Zelda and RNAP.

In *Figure 2A*, we illustrate the simplified case of two Bicoid binding sites and one Zelda binding site, plus the corresponding statistical weights of each state given by their Boltzmann factors. Note that the actual model utilized throughout this work accounts for at least 6 Bicoid-binding sites and 10 Zelda-binding sites that have been identified within the *hunchback* P2 enhancer (Section 'Predicting Zelda binding sites'; *Driever and Nüsslein-Volhard, 1988*; *Driever and Nüsslein-Volhard, 1989*; *Park et al., 2019*). This general model is described in detail in Appendix section 1.2.

The probability of finding RNAP bound to the promoter is calculated by dividing the sum of all statistical weights featuring RNAP by the sum of the weights corresponding to all possible system states. This leads to

$$p_{bound} = \frac{\left(1+z\right)^{n_z} p \left(1 + \sum_{i=1}^{n_b} \binom{n_b}{i} b^i \omega_b^{i-1} \omega_{bp}^i\right)}{\underbrace{e^{\Delta\varepsilon_{chrom}/k_B T}}_{\text{inaccessible state}} + \underbrace{\left(1+z\right)^{n_z}}_{\text{Zelda binding}} \underbrace{\left(1 + p + \sum_{j=0,1}\sum_{i=1}^{n_b} \binom{n_b}{i} b^i \omega_b^{i-1} p^j \omega_{bp}^{ij}\right)}_{\text{Bicoid and RNAP binding}}}, \tag{2}$$

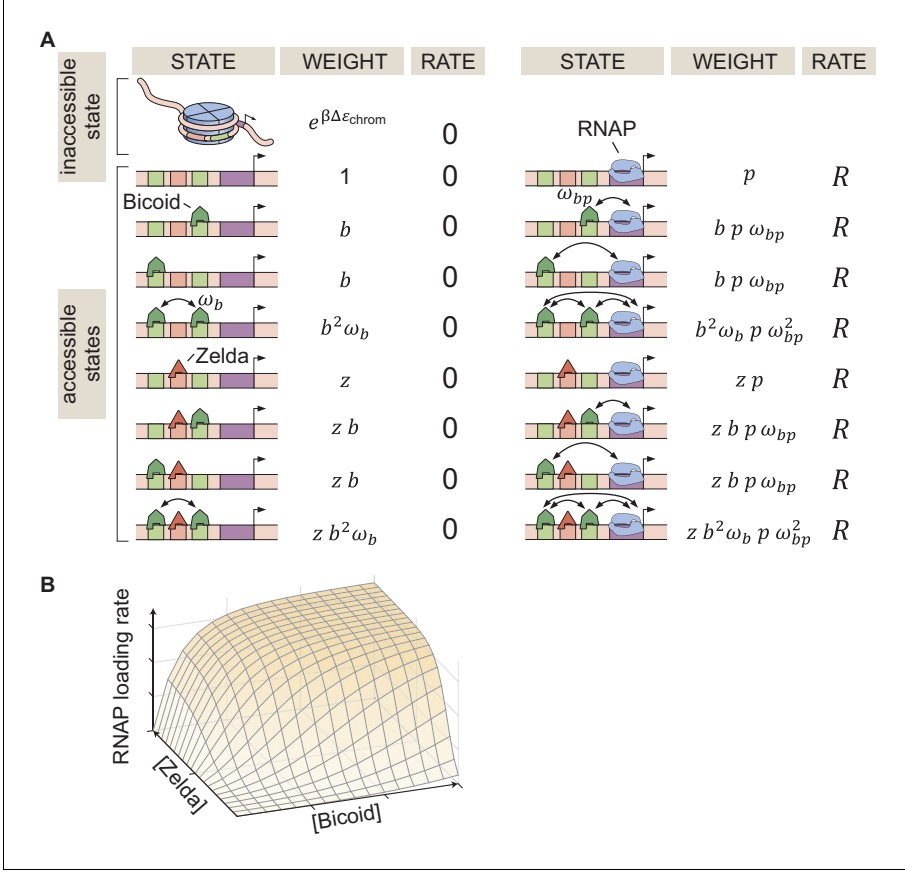

**Figure 2.** Thermodynamic MWC model of transcriptional regulation by Bicoid and Zelda. (**A**) States and statistical weights for a simplified version of the *hunchback* P2 enhancer. In this model, we assume that chromatin occluded by nucleosomes is not accessible to transcription factors or RNAP. Parameters are defined in the text. (**B**) 3D input-output function predicting the rate of RNAP loading (and of transcriptional initiation) as a function of Bicoid and Zelda concentrations for a given set of model parameters.

where $b = [Bicoid]/K_b$, $z = [Zelda]/K_z$, and $p = [RNAP]/K_p$, with $[Bicoid]$, $[Zelda]$, and $[RNAP]$ being the concentrations of Bicoid, Zelda, and RNAP, respectively, and $K_b$, $K_z$, and $K_p$ their dissociation constants (see Appendix sections 1.1 and 1.2 for a detailed derivation). Given a set of model parameters, plugging $p_{bound}$ into *Equation 1* predicts the rate of RNAP loading as a function of Bicoid and Zelda concentrations as shown in *Figure 2B*. Note that in this work, we treat the rate of transcriptional initiation and the rate of RNAP loading interchangeably.

## Dynamical prediction and measurement of input-output functions in development

In order to experimentally test the theoretical model in *Figure 2*, it is necessary to measure both the inputs – the concentrations of Bicoid and Zelda – as well as the output rate of RNAP loading. Typically, when testing models of transcriptional regulation in bacteria and eukaryotes, input transcription-factor concentrations are assumed to not be modulated in time: regulation is in steady state (*Ackers et al., 1982*; *Bakk et al., 2004*; *Segal et al., 2008*; *Garcia and Phillips, 2011*; *Sherman and Cohen, 2012*; *Cui et al., 2013*; *Little et al., 2013*; *Raveh-Sadka et al., 2009*; *Sharon et al., 2012*; *Zeigler and Cohen, 2014*; *Xu et al., 2015*; *Sepúlveda et al., 2016*; *Estrada et al., 2016*; *Razo-Mejia et al., 2018*; *Zoller et al., 2018*; *Park et al., 2019*). However, embryonic development is a highly dynamic process in which the concentrations of transcription factors are constantly changing due to their nuclear import and export dynamics, and due to protein production, diffusion, and degradation (*Edgar and Schubiger, 1986*; *Edgar et al., 1987*; *Jaeger et al., 2004b*; *Gregor et al., 2007b*). As a result, it is necessary to go beyond steady-state assumptions and to predict and measure how the *instantaneous*, time-varying concentrations of Bicoid and Zelda at each point in space dictate *hunchback* output transcriptional dynamics.

In order to quantify the concentration dynamics of Bicoid, we utilized an established Bicoid-eGFP line (Sections 'Fly Strains', 'Sample preparation and data collection' and 'Image analysis'; *Figure 3A* and *Appendix 1—figure 3A*; *Video 1*; *Gregor et al., 2007b*; *Liu et al., 2013*). As expected, this

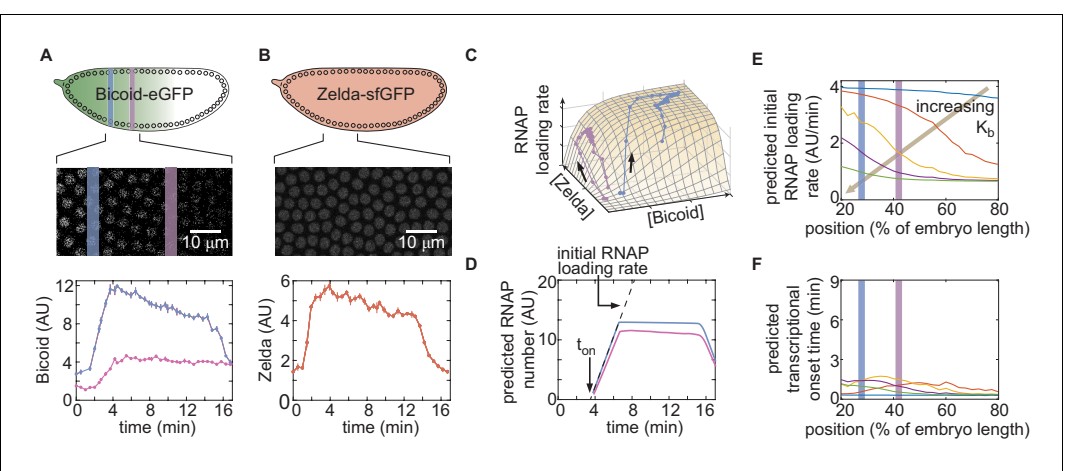

**Figure 3.** Prediction and measurement of dynamical input-output functions. (A) Measurement of Bicoid concentration dynamics in nuclear cycle 13. Color denotes different positions along the embryo and time is defined with respect to anaphase. (B) Zelda concentration dynamics. These dynamics are uniform throughout the embryo. (C) Trajectories defined by the input concentration dynamics of Bicoid and Zelda along the predicted input-output surface. Each trajectory corresponds to the RNAP loading-rate dynamics experienced by nuclei at the positions indicated in (A). (D) Predicted number of RNAP molecules actively transcribing the gene as a function of time and position along the embryo, and calculation of the corresponding initial rate of RNAP loading and the time of transcriptional onset, $t_{on}$. (E, F) Predicted *hunchback* (E) initial rate of RNAP loading and (F) $t_{on}$ as a function of position along the embryo for varying values of the Bicoid dissociation constant $K_b$. (A, B, error bars are standard error of the mean nuclear fluorescence in an individual embryo, averaged across all nuclei at a given position; D, the standard error of the mean predicted RNAP number in a single embryo, propagated from the errors in A and B, is thinner than the curve itself; E, F, only mean predictions are shown so as to not obscure differences between them; we imaged n=6 Bicoid-GFP and n=3 Zelda-GFP embryos.)

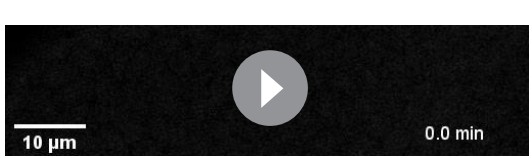

**Video 1.** Measurement of eGFP-Bicoid. Movie of eGFP-Bicoid fusion in an embryo in nuclear cycle 13. Time is defined with respect to the previous anaphase.
https://elifesciences.org/articles/56429#video1

line displayed the exponential Bicoid gradient across the length of the embryo (Appendix section 2.1; *Appendix 1—figure 3B*).We measured mean Bicoid nuclear concentration dynamics along the anterior-posterior axis of the embryo, as exemplified for two positions in *Figure 3A*. As previously reported (*Gregor et al., 2007b*), after anaphase and nuclear envelope formation, the Bicoid nuclear concentration quickly increases as a result of nuclear import. These measurements were used as inputs into the theoretical model in *Figure 2*.

Zelda concentration dynamics were measured in a Zelda-sfGFP line (Sections 'Fly Strains', 'Sample preparation and data collection', and 'Image analysis'; *Figure 3B*; *Video 2*; *Hamm et al., 2017*). Consistent with previous results (*Staudt et al., 2006*; *Liang et al., 2008*; *Dufourt et al., 2018*), the Zelda concentration was spatially uniform along the embryo (*Appendix 1—figure 3*). Contrasting *Figure 3A and B* reveals that the overall concentration dynamics of both Bicoid and Zelda are qualitatively comparable. As a result of Zelda's spatial uniformity, we used mean Zelda nuclear concentration dynamics averaged across all nuclei within the field of view to test our model (Appendix section 2.1; *Figure 3B*).

Given the high reproducibility of the concentration dynamics of Bicoid and Zelda (*Appendix 1—figure 3*), we combined measurements from multiple embryos by synchronizing their anaphase in order to create an 'averaged embryo' (Appendix section 2.1), an approach that has been repeatedly used to describe protein and transcriptional dynamics in the early fly embryo (*Garcia et al., 2013*; *Bothma et al., 2014*; *Bothma et al., 2015*; *Berrocal et al., 2018*; *Lammers et al., 2020*).

Our model assumes that *hunchback* output depends on the instantaneous concentration of input transcription factors. As a result, at each position along the anterior-posterior axis of the embryo, the combined Bicoid and Zelda concentration dynamics define a trajectory over time along the predicted input-output function surface (*Figure 3C*). The resulting trajectory predicts the rate of RNAP loading as a function of time. However, instead of focusing on calculating RNAP loading rate, we used it to compute the number of RNAP molecules actively transcribing *hunchback* at each point in space and time, a more experimentally accessible quantity (Section 'The thermodynamic MWC model fails to predict activation of *hunchback* in the absence of Zelda'). This quantity can be obtained by accounting for the RNAP elongation rate and the cleavage of nascent RNA upon termination (Appendix section 2.2; *Appendix 1—figure 4*; *Bothma et al., 2014*; *Lammers et al., 2020*) yielding the predictions shown in *Figure 3D*.

Instead of examining the full time-dependent nature of our data, we analyzed two main dynamical features stemming from our prediction of the number of RNAP molecules actively transcribing *hunchback*: the initial rate of RNAP loading and the transcriptional onset time, $t_{on}$, defined by the slope of the initial rise in the predicted number of RNAP molecules, and the time after anaphase at which transcription starts as determined by the x-intercept of the linear fit to the initial rise, respectively (*Figure 3D*).

Examples of the predictions generated by our theoretical model are shown in *Figure 3E and F*, where we calculate the initial rate of RNAP loading and $t_{on}$ for different values of the Bicoid dissociation constant $K_b$. This framework for quantitatively investigating dynamic input-output functions in living embryos is a necessary step toward testing the predictions of theoretical models of transcriptional regulation in development.

## The thermodynamic MWC model fails to predict activation of *hunchback* in the absence of Zelda

In order to test the predictions of the thermodynamic MWC model (*Figure 3E and F*), we used the MS2 system (*Bertrand et al., 1998*; *Garcia et al., 2013*; *Lucas et al., 2013*). Here,

**Video 2.** Measurement of Zelda-sfGFP. Movie of Zelda-sfGFP fusion in an embryo in nuclear cycle 13. Time is defined with respect to the previous anaphase.
https://elifesciences.org/articles/56429#video2

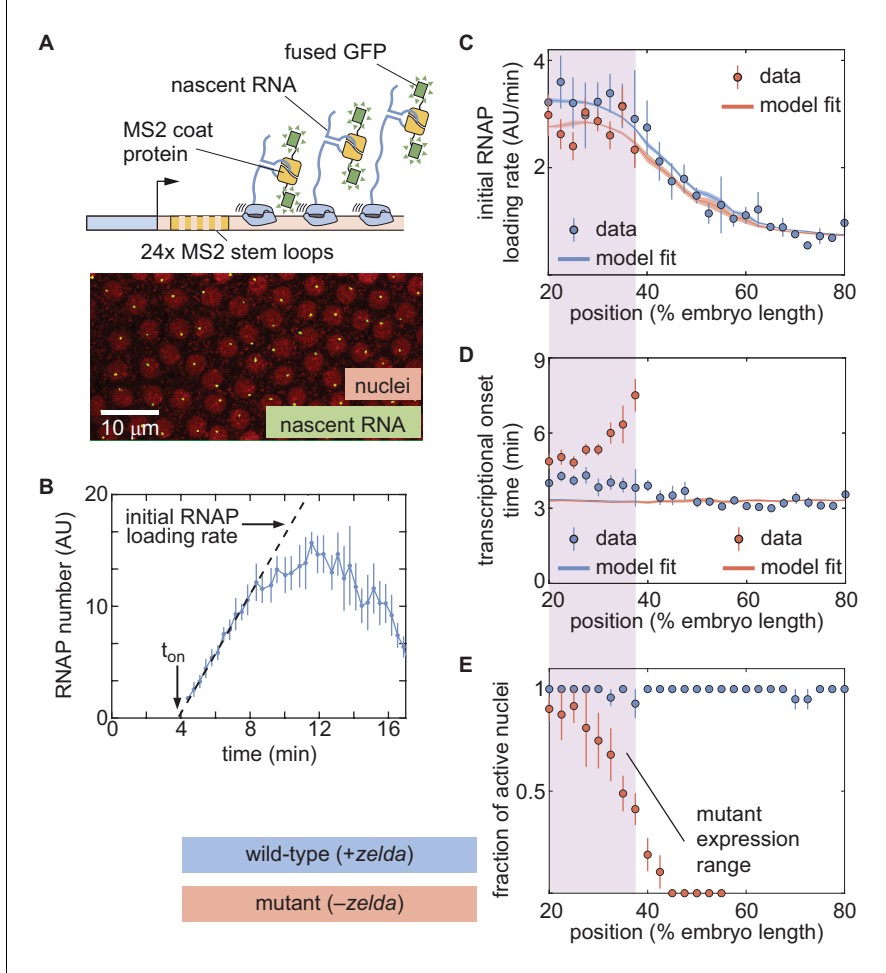

**Figure 4.** The thermodynamic MWC model can explain *hunchback* transcriptional dynamics in wild-type, but not *zelda⁻*, embryos. (**A**) The MS2 system measures the number of RNAP molecules actively transcribing the *hunchback* reporter gene in live embryos. (**B**) Representative MS2 trace featuring the quantification of the initial rate of RNAP loading and $t_{on}$. (**C**) Initial RNAP loading rate and (**D**) $t_{on}$ for wild-type (blue points) and *zelda⁻* (red points) embryos, compared with best fit to the thermodynamic MWC model (lines). The red and blue fit lines are close enough to overlap substantially. (**E**) Fraction of transcriptionally active nuclei for wild-type (blue) and *zelda⁻* (red) embryos. Active nuclei are defined as nuclei that exhibited an MS2 spot at any time during the nuclear cycle. Purple shading indicates the spatial range over which at least 30% of nuclei in the *zelda⁻* background display transcription. (B, error bars are standard error of the mean observed RNAP number, averaged across nuclei in a single embryo; C, D solid lines indicate mean predictions of the model, shading represents standard error of the mean; C, D, E, error bars in data points represent standard error of the mean over 11 wild-type embryos (blue) or 12 *zelda⁻* embryos (red)).

24 repeats of the MS2 loop are inserted in the 5′ untranslated region of the *hunchback* P2 reporter (*Garcia et al., 2013*), resulting in the fluorescent labeling of sites of nascent transcript formation (*Figure 4A*; *Video 3*). This fluorescence is proportional to the number of RNAP molecules actively transcribing the gene (*Garcia et al., 2013*). The experimental mean fluorescence as a function of time measured in a narrow window (2.5% of the total embryo length, averaged across nuclei in the window) along the length of the embryo (*Figure 4B*) is in qualitative agreement with the theoretical prediction (*Figure 3D*).

To compare theory and experiment, we next obtained the initial RNAP loading rates (*Figure 4C*, blue points) and $t_{on}$ (*Figure 4D*, blue points) from the experimental data (Appendix section 2.3; *Appendix 1—figure 5B*). The step-like shape of the RNAP loading rate (*Figure 4C*, blue points) agrees with previous measurements performed on this same reporter construct (*Garcia et al.,*

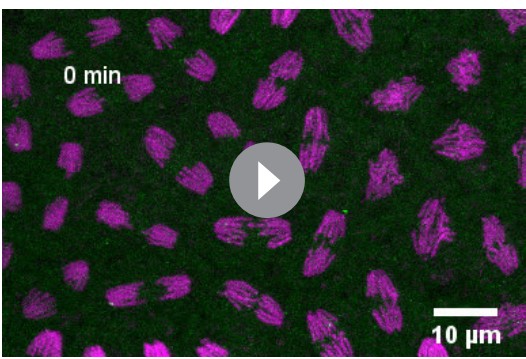

**Video 3.** Measurement of MS2 fluorescence in a wild-type background. Movie of MS2 fluorescent spots in a wild-type background embryo in nuclear cycle 13. Time is defined with respect to the previous anaphase.
https://elifesciences.org/articles/56429#video3

*2013*). The plateaus at the extreme anterior and posterior positions were used to constrain the maximum and minimum theoretically allowed values in the model (Appendix section 1.3). With these constraints in place, we attempted to simultaneously fit the thermodynamic MWC model to both the initial rate of RNAP loading and $t_{on}$. For a given set of model parameters, the measurements of Bicoid and Zelda concentration dynamics predicted a corresponding initial rate of RNAP loading and $t_{on}$ (*Figure 3E and F*). The model parameters were then iterated using standard curve-fitting techniques (Section 'Data analysis') until the best fit to the experimental data was achieved (*Figure 4C and D*, blue lines).

Although the model accounted for the initial rate of RNAP loading (*Figure 4C*, blue line), it produced transcriptional onset times that were much lower than those that we experimentally observed (*Appendix 1—figure 6B*, purple line). We hypothesized that this disagreement was due to our model not accounting for mitotic repression, when the transcriptional machinery appears to be silent immediately after cell division (*Shermoen and O'Farrell, 1991*; *Gottesfeld and Forbes, 1997*; *Parsons and Spencer, 1997*; *Garcia et al., 2013*). Thus, we modified the thermodynamic MWC model to include a mitotic repression window term, implemented as a time window at the start of the nuclear cycle during which no transcription could occur; the rate of mRNA production is thus given by

$$\frac{\mathrm{dmRNA}}{\mathrm{d}t} = \begin{cases} 0 & \text{if } t < t_{MitRep} \\ R p_{bound} & \text{if } t \geq t_{MitRep} \end{cases}, \tag{3}$$

where $R$ and $p_{bound}$ are as defined in *Equations 1 and 2*, respectively, and $t_{MitRep}$ is the mitotic repression time window over which no transcription can take place after anaphase (Appendix sections 1.2 and 3). After incorporating mitotic repression, the thermodynamic MWC model successfully fit both the rates of RNAP loading and $t_{on}$ (*Figure 4C and D*, blue lines, *Appendix 1—figure 6A and B*, blue lines).

Given this success, we next challenged the model to perform the simpler task of explaining Bicoid-mediated regulation in the absence of Zelda. This scenario corresponds to setting the concentration of Zelda to zero in the models in Appendix section 1.2 and *Figure 2*. In order to test this seemingly simpler model, we repeated our measurements in embryos devoid of Zelda protein (*Video 4*). These *zelda⁻* embryos were created by inducing clones of non-functional *zelda* mutant ($zelda^{294}$) germ cells in female adults (Sections 'Fly Strains', 'Zelda germline clones'; *Liang et al., 2008*). All embryos from these mothers lack maternally deposited Zelda; female embryos still have a functional copy of *zelda* from their father, but this copy is not transcribed until after the maternal-to-zygotic transition, during nuclear cycle 14 (*Liang et al., 2008*). We confirmed that the absence of Zelda did not have a substantial effect on the spatiotemporal pattern of Bicoid (Appendix section 4; *Xu et al., 2014*).

While close to 100% of nuclei in wild-type embryos exhibited transcription along the length of the embryo (*Figure 4E*, blue; *Video 5*), measurements in the *zelda⁻* background revealed that some nuclei never displayed any transcription during the entire nuclear cycle (*Video 6*). Specifically, transcription occurred only in the anterior part of the embryo, with transcription disappearing completely in positions posterior to about 40% of the embryo length (*Figure 4E*, red). We confirmed that no visible transcription spots were present in *zelda⁻* embryo posteriors by imaging in the posteriors of three *zelda⁻* embryos. These embryos are not included in our total embryo counts.

From those positions in the mutant embryos that did exhibit transcription in at least 30% of observed nuclei, we extracted the initial rate of RNAP loading and $t_{on}$ as a function of position. Interestingly, these RNAP loading rates were comparable to the corresponding rates in wild-type embryos (*Figure 4C*, red points). However, unlike in the wild-type case (*Figure 4D*, blue points), $t_{on}$

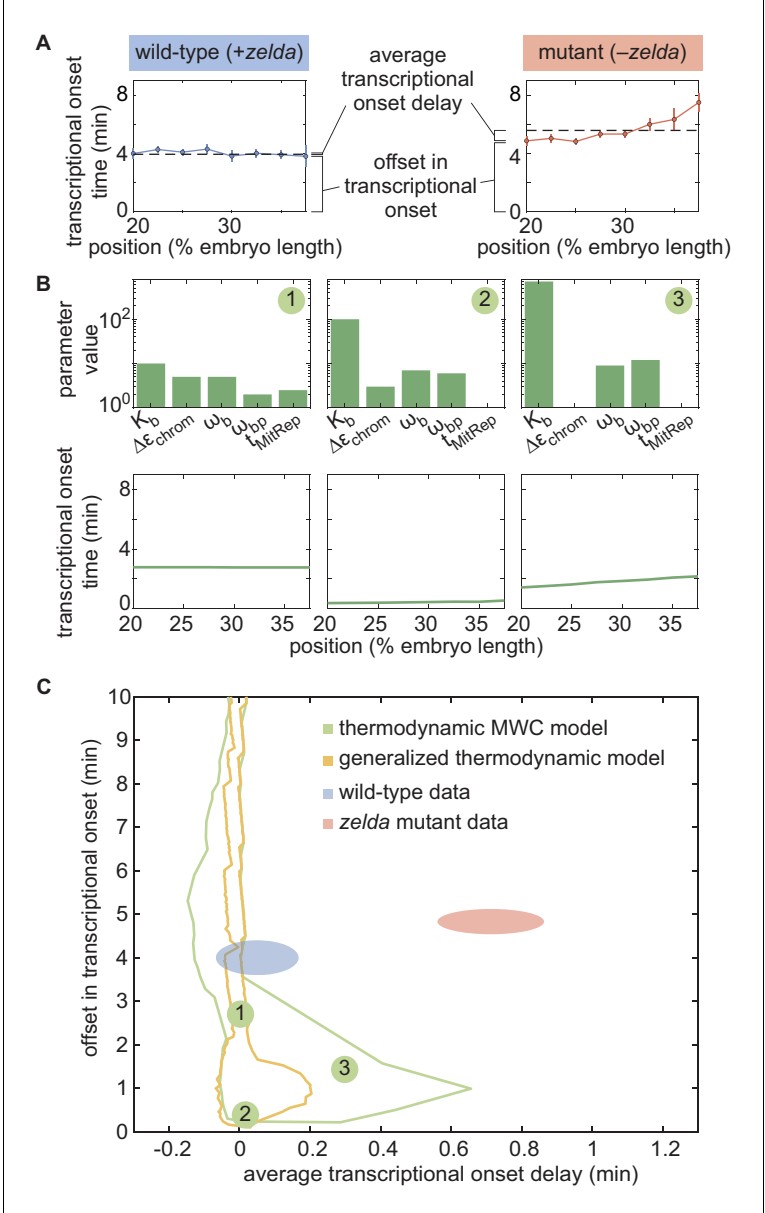

**Figure 5.** Failure of thermodynamic models to describe Bicoid-dependent activation of *hunchback*. (**A**) Experimentally determined $t_{on}$ with offset and average delay. Horizontal dashed lines indicate the average $t_{on}$ delay with respect to the offset in $t_{on}$ at 20% along the embryo for wild-type and $zelda^-$ data sets. (**B**) Exploration of $t_{on}$ offset and average $t_{on}$ delay from the thermodynamic MWC model. Each choice of model parameters predicts a distinct $t_{on}$ profile of along the embryo. (**C**) Predicted range of $t_{on}$ offset and average $t_{on}$ delay for the three cases featured in B (green points), for all possible parameter choices of the thermodynamic MWC model (green region), as well as for all thermodynamic models considering 12 Bicoid-binding sites (yellow region), compared with experimental data (red and blue regions). (A, C, error bars/ellipses represent standard error of the mean over 11 and 12 embryos for the wild-type and $zelda^-$ datasets, respectively; B, solid lines indicate mean predictions of the model).

was not constant in the $zelda^-$ background. Instead, $t_{on}$ became increasingly delayed in more posterior positions until transcription ceased posterior to 40% of the embryo length (*Figure 4D*, red points). Together, these observations indicated that removing Zelda primarily results in a delay of transcription with only negligible effects on the underlying rates of RNAP loading, consistent with previous fixed-embryo experiments (*Nien et al., 2011*; *Foo et al., 2014*) and with recent live-

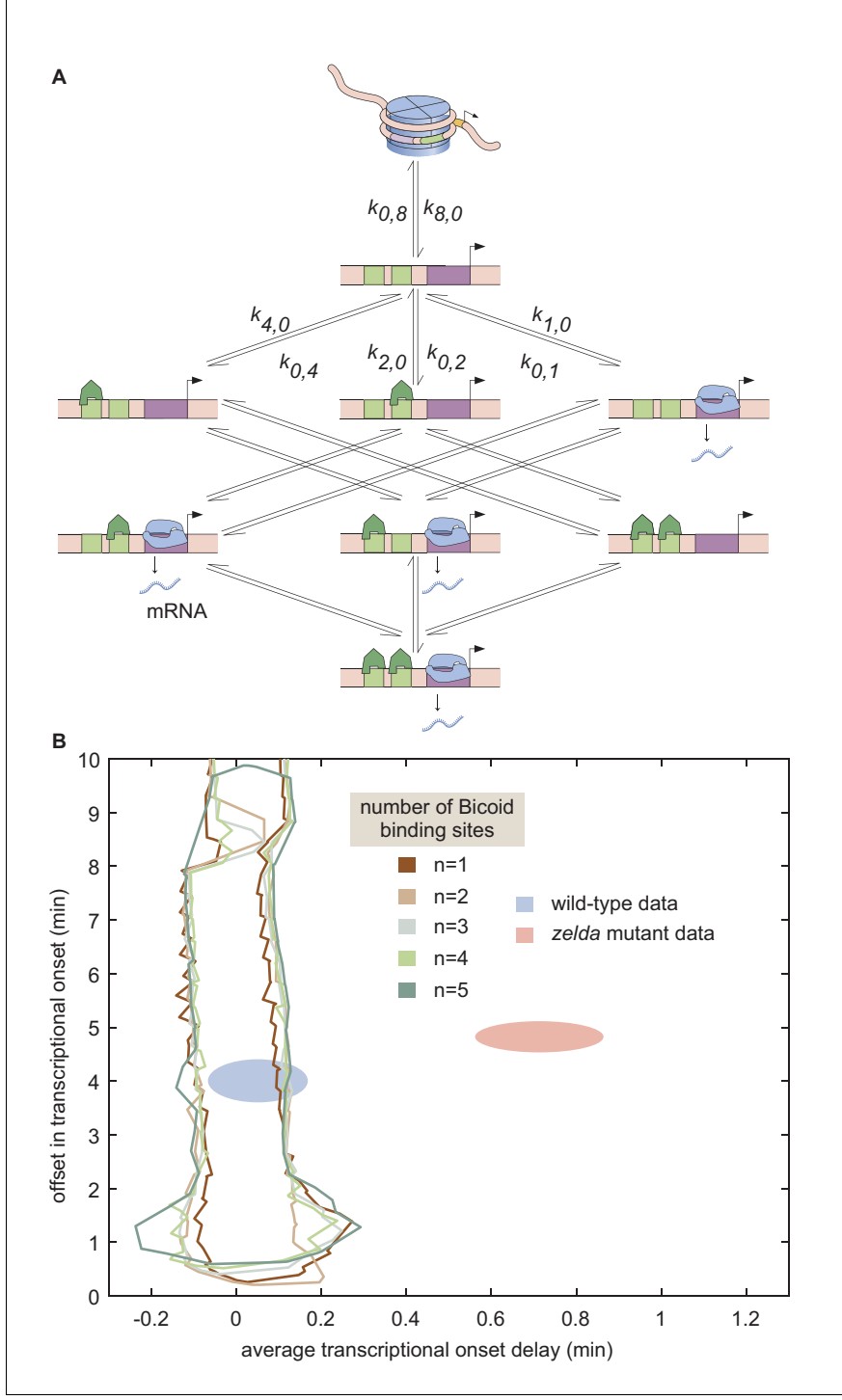

**Figure 6.** Non-equilibrium MWC model of transcriptional regulation cannot predict the observed $t_{on}$ delay. (A) Model that makes no assumptions about the relative transition rates between states or about energy expenditure. Each transition rate $i, j$ represents the rate of switching from state $i$ to state $j$. See Appendix section 7.1 for details on how the individual states are labeled. (B) Exploration of $t_{on}$ offset and average $t_{on}$ delay attainable by the non-equilibrium MWC models as a function of the number of Bicoid-binding sites compared to the experimentally obtained values corresponding to the wild-type and *zelda*− mutant backgrounds. While the non-equilibrium MWC model can explain the wild-type data, the exploration reveals that it fails to explain the *zelda*− data, for up to five Bicoid-binding sites. (B, ellipses represent standard error of the mean over 11 and 12 embryos for the wild-type and *zelda*− datasets, respectively).

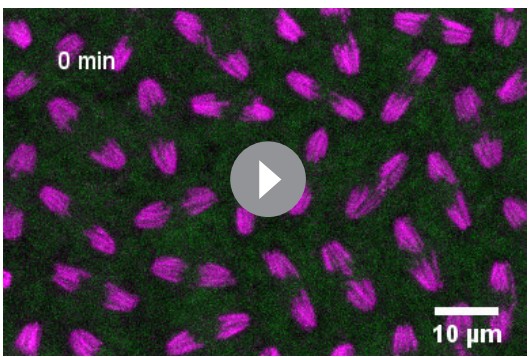

**Video 4.** Measurement of MS2 fluorescence in a *zelda⁻* background. Movie of MS2 fluorescent spots in a *zelda⁻* background embryo in nuclear cycle 13. Time is defined with respect to the previous anaphase.
https://elifesciences.org/articles/56429#video4

imaging measurements in which Zelda binding was reduced at specific enhancers (*Dufourt et al., 2018*; *Yamada et al., 2019*). We speculate that the loss of transcriptionally active nuclei posterior to 40% of the embryo length is a direct result of this delay in $t_{on}$: by the time that onset would occur in those nuclei, the processes leading to the next mitosis have already started and repressed transcriptional activity.

Next, we attempted to simultaneously fit the model to the initial rates of RNAP loading and $t_{on}$ in the *zelda⁻* mutant background. Although the model recapitulated the observed initial RNAP loading rates (*Figure 4C*, red line), we noticed a discrepancy between the observed and fitted transcriptional onset times of up to ~5 min (*Figure 4D*, red). While the mutant data exhibited a substantial delay in more posterior nuclei, the model did not produce significant delays (*Figure 4D*, red line). Further, our model could not account for the lack of transcriptional activity posterior to 40% of the embryo length in the *zelda⁻* mutant (*Figure 4E*, red).

These discrepancies suggest that the thermodynamic MWC model cannot fully describe the transcriptional regulation of the *hunchback* promoter by Bicoid and Zelda. However, the attempted fits in *Figure 4C and D* correspond to a particular set of model parameters and therefore do not completely rule out the possibility that there exists some parameter set of the thermodynamic MWC model capable of recapitulating the *zelda⁻* data.

In order to determine whether this model is *at all* capable of accounting for the *zelda⁻* transcriptional behavior, we systematically explored how its parameters dictate its predictions. To characterize and visualize the limits of our model, we examined two relevant quantitative features of our data. First, we defined the offset in the transcriptional onset time as the value of the onset time at the position 20% along the embryo length, the most anterior position studied here (*Figure 5A*), namely

$$\text{offset} = t_{on}(x = 20\%) \tag{4}$$

where $x$ is the position along the embryo. Second, we measured the average transcriptional onset delay along the anterior-posterior axis (*Figure 5A*). This quantity is defined as the area under the curve of $t_{on}$ versus embryo position, from 20% to 37.5% along the embryo (the positions where the *zelda⁻* embryos display transcription in at least 30% of nuclei), divided by the corresponding distance along the embryo

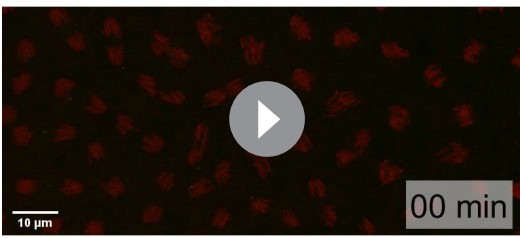

**Video 5.** Transcriptionally active nuclei in a wild-type background. Movie of MS2 fluorescent spots in a wild-type background embryo in nuclear cycle 13, with transcriptionally active nuclei labeled with an overlay. Time is defined with respect to the previous anaphase.
https://elifesciences.org/articles/56429#video5

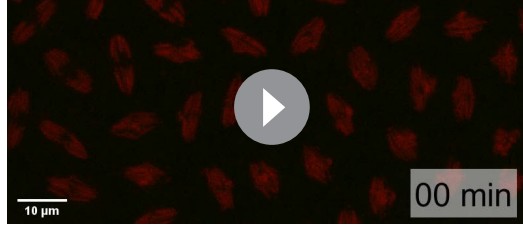

**Video 6.** Transcriptionally active nuclei in a *zelda⁻* background. Movie of MS2 fluorescent spots in a *zelda⁻* background embryo in nuclear cycle 13, with transcriptionally active nuclei labeled with an overlay. Time is defined with respect to the previous anaphase.
https://elifesciences.org/articles/56429#video6

$$\langle \text{onset delay} \rangle = \frac{1}{37.5\% - 20\%} \int_{20\%}^{37.5\%} (t_{on}(x) - t_{on}(x = 20\%)) dx, \tag{5}$$

where the offset in the onset time was used to define the zero of this integral (Appendix section 5.1). While the offset in $t_{on}$ is similar for both wild-type and *zelda⁻* backgrounds (approximately 4 min), the average $t_{on}$ delay corresponding to the wild-type data is close to 0 min, and is different from the value of about 0.7 min obtained from measurements in the *zelda⁻* background within experimental error (*Figure 5C*, ellipses).

Based on *Estrada et al., 2016* and as detailed in Appendix section 5.1, we used an algorithm to efficiently sample the parameter space of the thermodynamic MWC model (dissociation constants $K_b$ and $K_z$, protein-protein interaction terms $\omega_b$ and $\omega_{bp}$, energy to make the DNA accessible $\Delta\varepsilon_{\text{chrom}}$, and length of the mitotic repression window $t_{MitRep}$), and to calculate the corresponding $t_{on}$ offset and average $t_{on}$ delay for each parameter set. *Figure 5B* features three specific realizations of this parameter search; for each combination of parameters considered, the predicted $t_{on}$ is calculated and the corresponding $t_{on}$ offset and average $t_{on}$ delay computed. Although the wild-type data overlap with the thermodynamic MWC model region, the range of the $t_{on}$ offset and average $t_{on}$ delay predicted by the model (*Figure 5C*, green) did not overlap with that of the *zelda⁻* data. We concluded that our thermodynamic MWC model is not sufficient to explain the regulation of *hunchback* by Bicoid and Zelda.

## No thermodynamic model can recapitulate the activation of *hunchback* by Bicoid alone

Since the failure of the thermodynamic MWC model to predict the *zelda⁻* data does not necessarily rule out the existence of another thermodynamic model that can account for our experimental measurements, we considered other possible thermodynamic models. Conveniently, an arbitrary thermodynamic model featuring $n_b$ Bicoid binding sites can be generalized using the mathematical expression

$$\frac{d\text{mRNA}}{dt} = \frac{\left(\sum_{i=0}^{n_b} P_{1,i} R [Bicoid]^i \right)}{p_{inacc} + \sum_{r=0}^{1} \sum_{i=0}^{n_b} P_{r,i} [Bicoid]^i}, \tag{6}$$

where $p_{inacc}$ and $P_{r,i}$ are *arbitrary* weights describing the states in our generalized thermodynamic model, $R$ is a rate constant that relates promoter occupancy to transcription rate, and the $r$ and $i$ summations refer to the numbers of RNAP and Bicoid molecules bound to the enhancer, respectively (Appendix section 6.1; *Bintu et al., 2005a*; *Estrada et al., 2016*; *Scholes et al., 2017*). Note, that this generalized thermodynamic model also included the possibility of Bicoid binding to the inaccessible chromatin state (Appendix section 6.3).

Although this generalized thermodynamic model contains many more parameters than the thermodynamic MWC model previously considered, we could still systematically explore reasonable values of these parameters and the resulting $t_{on}$ offsets and average $t_{on}$ delays (Appendix section 6.2). For added generality, and to account for recent reports suggesting the presence of more than six Bicoid-binding sites in the *hunchback* minimal enhancer (*Park et al., 2019*), we expanded this model to include up to 12 Bicoid-binding sites.

The generalized thermodynamic model also failed to explain the *zelda⁻* data (Appendix section 6.2; *Figure 5C*, yellow). Note that the region of parameter space occupied by the generalized thermodynamic model does not entirely include that of the thermodynamic MWC model due to differences in the constraints of parameter values used in the parameter exploration, as described in Appendix sections 1.3 and 6.2. Nevertheless, our results strongly suggest that no thermodynamic model of Bicoid-activated *hunchback* transcription can predict transcriptional onset in the absence of Zelda, casting doubt on the general applicability of these models to transcriptional regulation in development.

Qualitatively, the reason for the failure of thermodynamic models to predict *hunchback* transcriptional is revealed by comparing Bicoid and Zelda concentration dynamics to those of the MS2 output signal (*Appendix 1—figure 10*). The thermodynamic models investigated in this work have assumed that the system responds *instantaneously* to any changes in input transcription factor concentration.

As a result, since Bicoid and Zelda are imported into the nucleus by around 3 min into the nuclear cycle (*Figure 3A and B*), these models always predict that transcription will ensue at approximately that time. Thus, thermodynamic models cannot accommodate delays in the $t_{on}$ such as those revealed by the *zelda*$^-$ data (see Appendix section 6.4 for a more detailed explanation). Rather than further complicating our thermodynamic models with additional molecular players to attempt to describe the data, we instead decided to examine the broader purview of non-equilibrium models to attempt to reach an agreement between theory and experiment.

## A non-equilibrium MWC model also fails to describe the *zelda*$^-$ data

Thermodynamic models based on equilibrium statistical mechanics can be seen as limiting cases of more general kinetic models that lie out of equilibrium (Appendix section 6.5; *Figure 1B*). Following recent reports (*Estrada et al., 2016*; *Li et al., 2018*; *Park et al., 2019*) that the theoretical description of transcriptional regulation in eukaryotes may call for models rooted in non-equilibrium processes – where the assumptions of separation of time scales and no energy expenditure may break down – we extended our earlier models to produce a non-equilibrium MWC model (Appendix sections 6.5 and 7.1; *Kim and O'Shea, 2008*; *Narula and Igoshin, 2010*). This model, shown for the case of two Bicoid binding sites in *Figure 6A*, accounts for the dynamics of the MWC mechanism by positing transition rates between the inaccessible and accessible chromatin states, but makes no assumptions about the relative magnitudes of these rates, or about the rates of Bicoid and RNAP binding and unbinding.

Since this model can operate out of steady state, we calculate the probabilities of each state as a function of time by solving the system of coupled ordinary differential equations (ODEs) associated with the system shown in *Figure 6A*. Consistent with prior measurements (*Blythe and Wieschaus, 2016*), we assume that chromatin is inaccessible at the start of the nuclear cycle. Over time, the system evolves such that the probability of it occupying each state becomes nonzero, making it possible to calculate the fraction of time RNAP is bound to the promoter and, through the occupancy hypothesis, the rate of RNAP loading. Mitotic repression is still incorporated using the term $t_{MitRep}$. For times $t < t_{MitRep}$, the system can evolve in time but the ensuing transcription rate is fixed at zero.

We systematically varied the magnitudes of the transition rates and solved the system of ODEs in order to calculate the corresponding $t_{on}$ offset and average $t_{on}$ delay. Due to the combinatorial increase of free parameters as more Bicoid-binding sites are included in the model, we could only explore the parameter space for models containing up to five Bicoid-binding sites (Appendix section 7.2; *Figure 6B* and *Appendix 1—figure 9*). Regardless, none of the non-equilibrium MWC models with up to five Bicoid-binding sites came close to reaching the mutant $t_{on}$ offset and average $t_{on}$ delay (*Figure 6B*). Additionally, an alternative version of this non-equilibrium MWC model where the system could not evolve in time until after the mitotic repression window had elapsed yielded similar conclusions (see Appendix section 7.3 for details). We conjecture that the observed behavior extends to the biologically relevant case of six or more binding sites. Thus, we conclude that the more comprehensive non-equilibrium MWC model still cannot account for the experimental data, motivating an additional reexamination of our assumptions.

## Transcription-factor-driven chromatin accessibility can capture all aspects of the data

Since even non-equilibrium MWC models incorporating energy expenditure and non-steady behavior could not explain the *zelda*$^-$ data, we further revised the assumptions of our model in an effort to quantitatively predict the regulation of $t_{on}$ along the embryo. In accordance with the MWC model of allostery, all of our theoretical treatments so far have posited that the DNA is an allosteric molecule that transitions between open and closed states as a result of thermal fluctuations (*Narula and Igoshin, 2010*; *Mirny, 2010*; *Marzen et al., 2013*; *Phillips et al., 2013*).

In the MWC models considered here, the presence of Zelda and Bicoid does not affect the microscopic rates of DNA opening and closing; rather, their binding to open DNA shifts the equilibrium of the DNA conformation toward the accessible state. However, recent biochemical work has suggested that Zelda and Bicoid play a more direct role in making chromatin accessible. Specifically, Zelda has been implicated in the acetylation of chromatin, a histone modification that renders nucleosomes unstable and increases DNA accessibility (*Li et al., 2014a*; *Li and Eisen, 2018*). Further,

Bicoid has been shown to interact with the co-activator dCBP, which possesses histone acetyltransferase activity (*Fu et al., 2004*). Additionally, recent studies by *Desponds et al., 2016* in *hunchback* and by *Dufourt et al., 2018* in *snail* have proposed the existence of multiple transcriptionally silent steps that the promoter needs to transition through before transcriptional onset. These steps could correspond to, for example, the recruitment of histone modifiers, nucleosome remodelers, and the transcriptional machinery (*Li et al., 2014a*; *Park et al., 2019*), or to the step-wise unraveling of discrete histone-DNA contacts (*Culkin et al., 2017*). Further, *Dufourt et al., 2018* proposed that Zelda plays a role in modulating the number of these steps and their transition rates.

We therefore proposed a model of transcription-factor-driven chromatin accessibility in which, in order for the DNA to become accessible and transcription to ensue, the system slowly and irreversibly transitions through $m$ transcriptionally silent states (Appendix section 8.1; *Figure 7A*). We assume that the transitions between these states are all governed by the same rate constant $\pi$. Finally, in a stark deviation from the MWC framework, we posit that these transitions can be catalyzed by the presence of Bicoid and Zelda such that

$$\pi = c_b[Bicoid] + c_z[Zelda]. \qquad (7)$$

Here, $\pi$ describes the rate (in units of inverse time) of each irreversible step, expressed as a sum of rates that depend separately on the concentrations of Bicoid and Zelda, and $c_b$ and $c_z$ are rate constants that scale the relative contribution of each transcription factor to the overall rate (see Appendix section 8.2 for a more detailed discussion of this choice). We emphasize that this is only one potential model, and there may exist several other non-equilibrium models capable of describing our data.

In this model of transcription-factor-driven chromatin accessibility, once the DNA becomes irreversibly accessible after transitioning through the $m$ non-productive states, we assume that, for the rest of the nuclear cycle, the system equilibrates rapidly such that the probability of it occupying any of its possible states is still described by equilibrium statistical mechanics. Like in our previous models, transcription only occurs in the RNAP-bound states, obeying the occupancy hypothesis. Further, our model assumes that if the transcriptional onset time of a given nucleus exceeds that of the next mitosis, this nucleus will not engage in transcription. Thus, this transcription-factor-driven model is an extension of the non-equilibrium MWC model with two crucial differences: (i) we allow for multiple inaccessible states preceding the transcriptionally active state, and (ii) the transitions between these states are *actively* driven by Bicoid or Zelda.

Unlike the thermodynamic and non-equilibrium MWC models, this model of transcription-factor-driven chromatin accessibility quantitatively recapitulated the observation that posterior nuclei in *zelda*⁻ embryos do not engage in transcription as well as the initial rate of RNAP loading, and $t_{on}$ for both the wild-type and *zelda*⁻ mutant data (*Figure 7B and C*). Additionally, we found that a minimum of $m = 3$ steps was required to sufficiently explain the data (Appendix section 8.3; *Appendix 1—figure 14*). Interestingly, unlike all previously considered models, the model of transcription-factor-driven chromatin accessibility did not require mitotic repression to explain $t_{on}$ (Appendix sections 3 and 8.1). Instead, the timing of transcriptional output arose directly from the model's initial irreversible transitions (*Appendix 1—figure 14*), obviating the need for an arbitrary suppression window in the beginning of the nuclear cycle. The only substantive disagreement between our theoretical model and the experimental data was that the model predicted that no nuclei should transcribe posterior to 60% of the embryo length, whereas no transcription posterior to 40% was experimentally observed in the embryo (*Figure 7B and C*). Finally, note that this model encompasses a much larger region of parameter space than the thermodynamic and non-equilibrium MWC models and, as expected from the agreement between model and experiment described above, contained both the wild-type and *zelda*⁻ data points within its domain (*Figure 7D*).

## Discussion

For four decades, thermodynamic models rooted in equilibrium statistical mechanics have constituted the null theoretical model for calculating how the number, placement and affinity of transcription factor binding sites on regulatory DNA dictates gene expression (*Bintu et al., 2005a*; *Bintu et al., 2005b*). Further, the MWC mechanism of allostery has been proposed as an

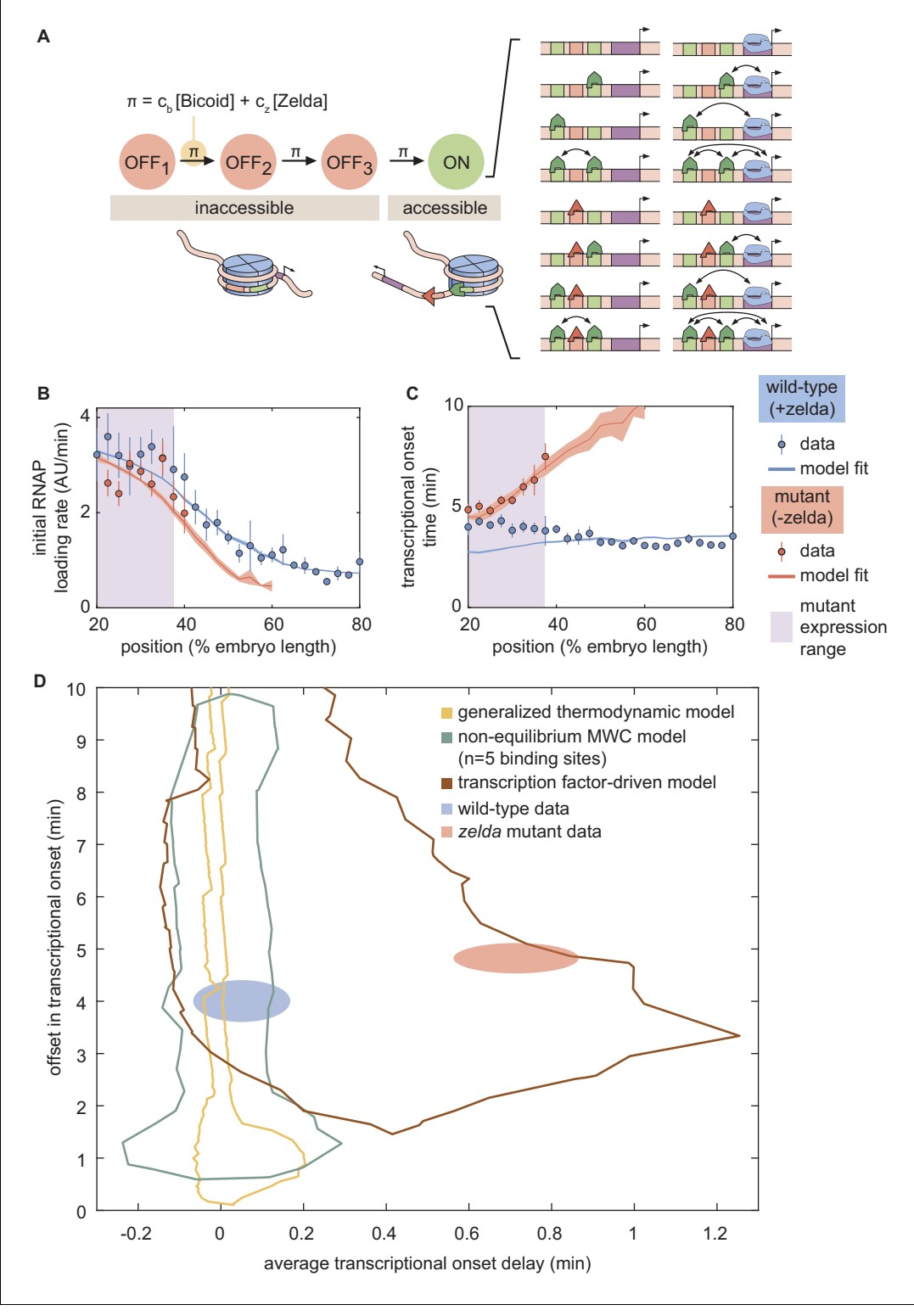

**Figure 7.** A model of transcription-factor-driven chromatin accessibility is sufficient to recapitulate *hunchback* transcriptional dynamics. (**A**) Overview of the proposed model, with three ($m = 3$) effectively irreversible Zelda and/ or Bicoid-mediated intermediate transitions from the inaccessible to the accessible state. (**B, C**) Experimentally fitted (**B**) initial RNAP loading rates and (**C**) $t_{on}$ for wild-type and *zelda⁻* embryos using a single set of parameters and assuming six Bicoid binding sites. (**D**) The domain of $t_{on}$ offset and average $t_{on}$ delay covered by this transcription-factor-driven chromatin accessibility model (brown) is much larger than those of the generalized

*Figure 7 continued on next page*

*Figure 7 continued*

thermodynamic model (yellow) and the non-equilibrium MWC models (green), and easily encompasses both experimental datasets (ellipses). (B-D, error bars/ellipses represent standard error of the mean over 11 and 12 embryos for the wild-type and *zelda*⁻ datasets, respectively).

extra layer that allows thermodynamic and more general non-equilibrium models to account for the regulation of chromatin accessibility (*Mirny, 2010*; *Narula and Igoshin, 2010*; *Marzen et al., 2013*).

In this investigation, we tested thermodynamic and non-equilibrium MWC models of chromatin accessibility and transcriptional regulation in the context of *hunchback* activation in the early embryo of the fruit fly *D. melanogaster* (*Driever et al., 1989*; *Nien et al., 2011*; *Xu et al., 2014*). While chromatin state (accessibility, post-translational modifications) is highly likely to influence transcriptional dynamics of associated promoters, specifically measuring the influence of chromatin state on transcriptional dynamics is challenging because of the sequential relationship between changes in chromatin state and transcriptional regulation. However, the *hunchback* P2 minimal enhancer provides a unique opportunity to dissect the relative contribution of chromatin regulation on transcriptional dynamics because, in the early embryo, chromatin accessibility at *hunchback* is granted by both Bicoid and Zelda (*Hannon et al., 2017*). The degree of *hunchback* transcriptional activity, however, is regulated directly by Bicoid (*Driever and Nüsslein-Volhard, 1989*; *Driever et al., 1989*; *Struhl et al., 1989*). Therefore, while genetic elimination of Zelda function interferes with acquisition of full chromatin accessibility, the *hunchback* locus retains a measurable degree of accessibility and transcriptional activity stemming from Bicoid function, allowing for a quantitative determination of the contribution of Zelda-dependent chromatin accessibility on the transcriptional dynamics of the locus.

With these attributes in mind, we constructed a thermodynamic MWC model which, given a set of parameters, predicted an output rate of *hunchback* transcription as a function of the input Bicoid and Zelda concentrations (*Figure 2B*). In order to test this model, it was necessary to acknowledge that development is not in steady-state, and that both Bicoid and Zelda concentrations change dramatically in space and time (*Figure 3A and B*). As a result, we went beyond widespread steady-state descriptions of development and introduced a novel approach that incorporated transient dynamics of input transcription-factor concentrations in order to predict the instantaneous output transcriptional dynamics of *hunchback* (*Figure 3C*). Given input dynamics quantified with fluorescent protein fusions to Bicoid and Zelda, we both predicted output transcriptional activity and measured it with an MS2 reporter (*Figures 3D* and *4B*).

This approach revealed that the thermodynamic MWC model sufficiently predicts the timing of the onset of transcription and the subsequent initial rate of RNAP loading as a function of Bicoid and Zelda concentration. However, when confronted with the much simpler case of Bicoid-only regulation in a *zelda* mutant, the thermodynamic MWC model failed to account for the observations that only a fraction of nuclei along the embryo engaged in transcription, and that the transcriptional onset time of those nuclei that do transcribe was significantly delayed with respect to the wild-type setting (*Figure 4D and E*). Our systematic exploration of all thermodynamic models (over a reasonable parameter range) showed that that no thermodynamic model featuring regulation by Bicoid alone could quantitatively recapitulate the measurements performed in the *zelda* mutant background (*Figure 5C*, yellow).

This disagreement could be resolved by invoking an unknown transcription factor that regulates the *hunchback* reporter in addition to Bicoid. However, at the early stages of development analyzed here, such a factor would need to be both maternally provided and patterned in a spatial gradient to produce the observed position-dependent transcriptional onset times. To our knowledge, none of the known maternal genes regulate the expression of this *hunchback* reporter in such a fashion (*Chen et al., 2012*; *Perry et al., 2012*; *Xu et al., 2014*). We conclude that the MWC thermodynamic model cannot accurately predict *hunchback* transcriptional dynamics.

To explore non-equilibrium models, we retained the MWC mechanism of chromatin accessibility, but did not demand that the accessible and inaccessible states be in thermal equilibrium. Further, we allowed for the process of Bicoid and RNAP binding, as well as their interactions, to consume energy. For up to five Bicoid-binding sites, no set of model parameters could quantitatively account for the transcriptional onset features in the *zelda* mutant background (*Figure 6B*). While we were

unable to investigate models with more than five Bicoid-binding sites due to computational complexity (*Estrada et al., 2016*), the substantial distance in parameter space between the mutant data and the investigated models (*Figure 6B*) suggested that a successful model with more than five Bicoid-binding sites would probably operate near the limits of its explanatory power, similar to the conclusions from studies that explored *hunchback* regulation under the steady-state assumption (*Park et al., 2019*). Thus, despite the simplicity and success of the MWC model in predicting the effects of protein allostery in a wide range of biological contexts (*Keymer et al., 2006*; *Swem et al., 2008*; *Martins and Swain, 2011*; *Marzen et al., 2013*; *Rapp and Yifrach, 2017*; *Razo-Mejia et al., 2018*; *Chure et al., 2019*; *Rapp and Yifrach, 2019*), the observed transcriptional onset times could not be described by any previously proposed thermodynamic MWC mechanism of chromatin accessibility, or even by a more generic non-equilibrium MWC model in which energy is continuously dissipated (*Tu, 2008*; *Kim and O'Shea, 2008*; *Narula and Igoshin, 2010*; *Estrada et al., 2016*; *Wang et al., 2017*).

Since Zelda is associated with histone acetylation, which is correlated with increased chromatin accessibility (*Li et al., 2014a*; *Li and Eisen, 2018*), and Bicoid interacts with the co-activator dCBP, which has histone acetyltransferase activity (*Fu et al., 2004*; *Fu and Ma, 2005*; *Park et al., 2019*), we suspect that both Bicoid and Zelda actively drive DNA accessibility. A molecular pathway shared by Bicoid and Zelda to render chromatin accessible is consistent with our results, and with recent genome-wide experiments showing that Bicoid can rescue the function of Zelda-dependent enhancers at high enough concentrations (*Hannon et al., 2017*). Thus, the binding of Bicoid and Zelda, rather than just biasing the equilibrium toward the open chromatin state as in the MWC mechanism, may trigger a set of molecular events that locks DNA into an accessible state. In addition, the promoters of *hunchback* (*Desponds et al., 2016*) and *snail* (*Dufourt et al., 2018*) may transition through a set of intermediate, non-productive states before transcription begins.

We therefore explored a model in which Bicoid and Zelda catalyze the transition of chromatin into the accessible state via a series of slow, effectively irreversible steps. These steps may be interpreted as energy barriers that are overcome through the action of Bicoid and Zelda, consistent with the coupling of these transcription factors to histone modifiers, nucleosome remodelers (*Fu et al., 2004*; *Li et al., 2014a*; *Li and Eisen, 2018*; *Park et al., 2019*), and with the step-wise breaking of discrete histone-DNA contacts to unwrap nucleosomal DNA (*Culkin et al., 2017*). In this model, once accessible, the chromatin remains in that state and the subsequent activation of *hunchback* by Bicoid is described by a thermodynamic model.

Crucially, this transcription-factor-driven chromatin accessibility model successfully replicated all of our experimental observations. A minimum of three transcriptionally silent states were necessary to explain our data (*Figure 7D* and *Appendix 1—figure 14C*). Interestingly, recent work dissecting the transcriptional onset time distribution of *snail* also suggested the existence of three such intermediate steps in the context of that gene (*Dufourt et al., 2018*). Given that, as in *hunchback*, the removal and addition of Zelda modulates the timing of transcriptional onset of *sog* and *snail* (*Dufourt et al., 2018*; *Yamada et al., 2019*), we speculate that transcription-factor-driven chromatin accessibility may also be at play in these pathways. Thus, taken in consideration with similar works examining the dynamics of transcription onset (*Desponds et al., 2016*; *Dufourt et al., 2018*; *Fritzsch et al., 2018*; *Li et al., 2018*), our results strongly suggest that chromatin state does not fluctuate thermodynamically, but rather progresses through a series of stepwise, transcription-factor-driven transitions into a final RNAP-accessible configuration (*Coulon et al., 2013*).

Intriguingly, accounting for these intermediate states also obviated the need for the *ad hoc* imposition of a mitotic repression window (Appendix sections 3 and 8.1), which was required in the thermodynamic MWC model (*Appendix 1—figure 6*). Our results thus suggest a mechanistic interpretation of the phenomenon of mitotic repression after anaphase, where the promoter must traverse through intermediary transcriptionally silent states before transcriptional onset can occur.

These clues into the molecular mechanisms of action of Bicoid, Zelda, and their associated modifications to the chromatin landscape pertain to a time scale of a few minutes, a temporal scale that is inaccessible with widespread genome-wide and fixed-tissue approaches. Here, we revealed the regulatory action of Bicoid and Zelda by utilizing the dynamic information provided by live imaging to analyze the transient nature of the transcriptional onset time, highlighting the need for descriptions of development that go beyond steady state and acknowledge the highly dynamic changes in transcription-factor concentrations that drive developmental programs.

While we showed that one model incorporating transcription-factor-driven chromatin accessibility could recapitulate *hunchback* transcriptional regulation by Bicoid and Zelda, and is consistent with molecular evidence on the modes of action of these transcription factors, other models may have comparable explanatory power. In the future, a systematic exploration of different classes of models and their unique predictions will identify measurements that determine *which* specific model is the most appropriate description of transcriptional regulation in development and *how* it is implemented at the molecular level. While all the analyses in this work relied on mean levels of input concentrations and output transcription levels, detailed studies of single-cell features of transcriptional dynamics such as the distribution of transcriptional onset times (*Narula and Igoshin, 2010*; *Dufourt et al., 2018*; *Fritzsch et al., 2018*) could shed light on these chromatin-regulating mechanisms. Simultaneous measurement of local transcription-factor concentrations at sites of transcription and of transcriptional initiation with high spatiotemporal resolution, such as afforded by lattice light-sheet microscopy (*Mir et al., 2018*), could provide further information about chromatin accessibility dynamics. Finally, different theoretical models may make distinct predictions about the effect of modulating the number, placement, and affinity of Bicoid and Zelda sites (and even of nucleosomes) in the *hunchback* enhancer. These models could be tested with future experiments that implement these modulations in reporter constructs.

In sum, here we engaged in a theory-experiment dialogue to respond to the theoretical challenges of proposing a passive MWC mechanism for chromatin accessibility in eukaryotes (*Mirny, 2010*; *Narula and Igoshin, 2010*; *Marzen et al., 2013*); we also questioned the suitability of thermodynamic models in the context of development (*Estrada et al., 2016*). At least regarding the activation of *hunchback*, and likely similar developmental genes such as *snail* and *sog* (*Dufourt et al., 2018*; *Yamada et al., 2019*), we speculate that Bicoid and Zelda actively drive chromatin accessibility, possibly through histone acetylation. Once chromatin becomes accessible, thermodynamic models can predict *hunchback* transcription without the need to invoke energy expenditure and non-equilibrium models. Regardless of whether we have identified the only possible model of chromatin accessibility and regulation, we have demonstrated that this dialogue between theoretical models and the experimental testing of their predictions at high spatiotemporal resolution is a powerful tool for biological discovery. The new insights afforded by this dialogue will undoubtedly refine theoretical descriptions of transcriptional regulation as a further step toward a predictive understanding of cellular decision-making in development.

# Materials and methods

### Predicting Zelda-binding sites

Zelda-binding sites in the *hunchback* promoter were identified as heptamers scoring three or higher using a Zelda alignment matrix (*Harrison et al., 2011*) and the Advanced PASTER entry form online (http://stormo.wustl.edu/consensus/cgi-bin/Server/Interface/patser.cgi) (*Hertz et al., 1990*; *Hertz and Stormo, 1999*). PATSER was run with setting 'Seq. Alphabet and Normalization' as 'a:t 3 g:c 2' to provide the approximate background frequencies as annotated in the Berkeley *Drosophila* Genome Project (BDGP)/Celera Release 1. Reverse complementary sequences were also scored.

### Fly strains

Bicoid nuclear concentration was imaged in embryos from line *yw;his2av-mrfp1;bicoidE1,egfp-bicoid* (*Gregor et al., 2007b*). Similarly, Zelda nuclear concentration was determined by imaging embryos from line *sfgfp-zelda;+;his-irfp*. The *sfgfp-zelda* transgene was obtained from *Hamm et al., 2017* and the *his-iRFP* transgene is courtesy of Kenneth Irvine and Yuanwang Pan.

Transcription from the *hunchback* promoter was measured by imaging embryos resulting from crossing female virgins *yw;HistoneRFP;MCP-NoNLS(2)* with male *yw;P2P-MS2-LacZ/cyo;+* (*Garcia et al., 2013*).

In order to image transcription in embryos lacking maternally deposited Zelda protein, we crossed mother flies whose germline was *w,his2av-mrfp1,zelda(294),FRT19A;+;MCP-egfp(4F)/+* obtained through germline clones (see below) with fathers carrying the *yw;P2P-MS2-LacZ/cyo;+* reporter. The $zelda^{294}$ transgene is courtesy of Christine Rushlow (*Liang et al., 2008*). The *MCP-egfp*

*(4F)* transgene expresses approximately double the amount of MCP than the *MCP-egfp(2)* (*Garcia et al., 2013*), ensuring similar levels of MCP in the embryo in all experiments.

Imaging Bicoid nuclear concentration in embryos lacking maternally deposited Zelda protein was accomplished by replacing the *MCP-egfp(4F)* transgene described in the previous paragraph with the *bicoidE1,egfp-bicoid* transgene used for imaging nuclear Bicoid in a wild-type background. We crossed mother flies whose germline was *w,his2av-mrfp1,zelda(294),FRT19A;+;bicoidE1,egfp-bicoid/+* obtained through germline clones (see below) with *yw* fathers.

### Zelda germline clones

In order to generate mother flies containing a germline homozygous null for *zelda*, we first crossed virgin females of *w,his2av-mrfp1,zelda(294),FRT19A/FM7,y,B;+;MCP-egfp(4F)/TM3,ser* (or *w,his2av-mrfp1,zelda(294),FRT19A;+;bicoidE1,egfp-bicoid/+* to image nuclear Bicoid) with males of *ovoD,hs-FLP,FRT19A;+;+* (*Liang et al., 2008*). The resulting heterozygotic offspring were heat-shocked in order to create maternal germline clones as described in *Liang et al., 2008*. The resulting female virgins were crossed with male *yw;P2P-MS2-LacZ/cyo;+* (*Garcia et al., 2013*) to image transcription or male *yw* to image nuclear Bicoid concentration.

Male offspring are null for zygotic *zelda*. Female offspring are heterozygotic for functional *zelda*, but zygotic *zelda* is not transcribed until nuclear cycle 14 (*Liang et al., 2008*), which occurs after the analysis in this work. All embryos lacking maternally deposited Zelda showed aberrant morphology in nuclear size and shape (data not shown), as previously reported (*Liang et al., 2008*; *Staudt et al., 2006*).

### Sample preparation and data collection

Sample preparation followed procedures described in *Bothma et al., 2014*, *Garcia and Gregor, 2018*, and *Lammers et al., 2020*.

Embryos were collected and mounted in halocarbon oil 27 between a semipermeable membrane (Lumox film, Starstedt, Germany) and a coverslip. Data collection was performed using a Leica SP8 scanning confocal microscope (Leica Microsystems, Biberach, Germany). Imaging settings for the MS2 experiments were the same as in *Lammers et al., 2020*, except the Hybrid Detector (HyD) for the His-RFP signal used a spectral window of 556–715 nm. The settings for the Bicoid-GFP measurements were the same, except for the following. The power setting for the 488 nm line was 10 μW. The confocal stack was only 10 slices in this case, rather than 21, resulting in a spacing of 1.11 μm between planes. The images were acquired at a time resolution of 30 s, using an image resolution of $512 \times 128$ pixels.

The settings for the Zelda-sfGFP measurements were the same as the Bicoid-GFP measurements, except different laser lines were used for the different fluorophores. The sf-GFP excitation line was set at 485 nm, using a power setting of 10 μW. The His-iRFP excitation line was set at 670 nm. The HyD for the His-iRFP signal was set at a 680–800 nm spectral window. All specimens were imaged over the duration of nuclear cycle 13.

### Image analysis

Images were analyzed using custom-written software following the protocol in *Garcia et al., 2013*. Briefly, this procedure involved segmenting individual nuclei using the histone signal as a nulear mask, segmenting each transcription spot based on its fluorescence, and calculating the intensity of each MCP-GFP transcriptional spot inside a nucleus as a function of time.

Additionally, the nuclear protein fluorescences of the Bicoid-GFP and Zelda-sfGFP fly lines were calculated as follows. Using the histone-labeled nuclear mask for each individual nucleus, the fluorescence signal within the mask was extracted in xyz, as well as through time. For each timepoint, the xy signal was averaged to give an average nuclear fluorescence as a function of z and time. This signal was then maximum projected in z, resulting in an average nuclear concentration as a function of time, per single nucleus. These single nucleus concentrations were then averaged over anterior-posterior position to create the protein concentrations reported in the main text.

## Data analysis

All fits in the main text were performed by minimizing the least-squares error between the data and the model predictions. Unless stated otherwise, error bars reflect standard error of the mean over multiple embryo measurements. See Appendix section 2.1 for more details on how this was carried out for model predictions.

## Acknowledgements

We are grateful to Jack Bateman, Jacques Bothma, Mike Eisen, Jeremy Gunawardena, Jane Kondev, Oleg Igoshin, Rob Phillips, Christine Rushlow and Peter Whitney for their guidance and comments on our manuscript. We thank Kenneth Irvine and Yuanwang Pan for providing the *his-irfp* fly line. This work was supported by the Burroughs Wellcome Fund Career Award at the Scientific Interface, the Sloan Research Foundation, the Human Frontiers Science Program, the Searle Scholars Program, the Shurl and Kay Curci Foundation, the Hellman Foundation, the NIH Director's New Innovator Award (DP2 OD024541-01), and an NSF CAREER Award (1652236) (HGG), an NSF GRFP (DGE 1752814) and UC Berkeley Chancellor's Fellowship (EE), and the DoD NDSEG graduate fellowship (JL).

## Additional information

### Funding

| Funder | Grant reference number | Author |
|---|---|---|
| National Science Foundation | DGE1752814 | Elizabeth Eck |
| University of California Berkeley | Chancellor's Fellowship | Elizabeth Eck |
| Department of Defense | Graduate Student Fellowship | Jonathan Liu |
| Burroughs Wellcome Fund | Career Award | Hernan G Garcia |
| Sloan Research Foundation | | Hernan G Garcia |
| Human Frontier Science Program | | Hernan G Garcia |
| Searle Scholars Program | | Hernan G Garcia |
| Shurl and Kay Curci Foundation | | Hernan G Garcia |
| Hellman Foundation | | Hernan G Garcia |
| National Institutes of Health | DP2 OD024541-01 | Hernan G Garcia |
| National Science Foundation | 1652236 | Hernan G Garcia |

The funders had no role in study design, data collection and interpretation, or the decision to submit the work for publication.

### Author contributions

Elizabeth Eck, Jonathan Liu, Data curation, Software, Formal analysis, Funding acquisition, Validation, Investigation, Visualization, Methodology, Writing - original draft, Writing - review and editing; Maryam Kazemzadeh-Atoufi, Sydney Ghoreishi, Data curation, Validation, Investigation; Shelby A Blythe, Resources, Writing - review and editing; Hernan G Garcia, Conceptualization, Resources, Supervision, Funding acquisition, Investigation, Writing - original draft, Project administration, Writing - review and editing

### Author ORCIDs

Elizabeth Eck https://orcid.org/0000-0003-0139-3865
Jonathan Liu https://orcid.org/0000-0003-0204-0105

Shelby A Blythe [ID] http://orcid.org/0000-0003-4986-2579
Hernan G Garcia [ID] https://orcid.org/0000-0002-5212-3649

**Decision letter and Author response**
Decision letter https://doi.org/10.7554/eLife.56429.sa1
Author response https://doi.org/10.7554/eLife.56429.sa2

## Additional files

### Supplementary files

- Source code 1. mRNADynamics image analysis software.

- Transparent reporting form

### Data availability

Processed microscopy data have been deposited in Dryad (https://datadryad.org/stash/share/zak-b7AqU2233pgWls1mMAKyDiTQi4BXtnP0-Uu93xI0).

The following dataset was generated:

| Author(s) | Year | Dataset title | Dataset URL | Database and Identifier |
|---|---|---|---|---|
| Elizabeth Eck, Jonathan Liu, Maryam Kazemzadeh-Atoufi, Sydney Ghoreishi, Shelby A Blythe, Hernan G Garcia | 2020 | Quantitative dissection of transcription in development yields evidence for transcription factor-driven chromatin accessibility | https://datadryad.org/stash/dataset/doi:10.6078/D1GX19 | Dryad, 10.6078/D1GX19 |

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

## Appendix 1

### 1. Equilibrium models of transcription

#### 1.1 An overview of equilibrium thermodynamics models of transcription

In this section, we give a brief overview of the theoretical concepts behind equilibrium thermodynamics models of transcription. For a more detailed overview, we refer the reader to *Bintu et al., 2005b* and *Bintu et al., 2005a*. These models invoke statistical mechanics in order to to calculate bulk properties of a system by enumerating the probability of each possible microstate of the system. The probability of a given microstate is proportional to its Boltzmann weight $e^{-\beta\varepsilon}$, where $\varepsilon$ is the energy of the microstate and $\beta = (k_B T)^{-1}$ with $k_B$ being the Boltzmann constant and $T$ the absolute temperature of the system (*Garcia et al., 2007*).

Specific examples of these microstates in the context of simple activation are featured in *Appendix 1—figure 1*. As reviewed in *Garcia et al., 2007*, the Boltzmann weight of each of these microstates can also be written in a thermodynamic language that accounts for the concentration of the molecular species, their dissociation constant to DNA, and a cooperativity term ω that accounts for the protein-protein interactions between the activator and RNAP. To calculate the probability of finding RNAP bound to the promoter $p_{bound}$, we divide the sum of the weights of the RNAP-bound states by the sum of all possible states

$$p_{bound} = \frac{\frac{[P]}{K_p} + \omega \frac{[P]\,[A]}{K_p\,K_a}}{1 + \frac{[P]}{K_p} + \frac{[A]}{K_a} + \omega \frac{[P]\,[A]}{K_p\,K_a}}. \tag{1}$$

Here, $[P]$ and $[A]$ are the concentrations of RNAP and activator, respectively. $K_p$ and $K_a$ are their corresponding dissociation constants, and $\omega$ indicates an interaction between activator and RNAP: $\omega > 1$ corresponds to cooperativity, whereas $0 < \omega < 1$ corresponds to anti-cooperativity.

| STATE | WEIGHT | RATE |
|---|---|---|
| | $1$ | $0$ |
| | $\dfrac{[P]}{K_p}$ | $R$ |
| | $\dfrac{[A]}{K_a}$ | $0$ |
| | $\omega \dfrac{[P]}{K_p}\dfrac{[A]}{K_a}$ | $R$ |

**Appendix 1—figure 1.** Equilibrium thermodynamic model of simple activation. A promoter region with one binding site for an activator molecule has four possible microstates, each with its corresponding statistical weight and rate of RNAP loading.

Using $p_{bound}$, we write the subsequent rate of mRNA production by assuming the *occupancy hypothesis*, which states that

$$\frac{\mathrm{dmRNA}}{\mathrm{d}t} = R\, p_{bound}, \tag{2}$$

where $R$ is an underlying rate of transcriptional initiation (usually interpreted as the rate of loading RNAP from the promoter-bound state). In the case of simple activation illustrated in *Appendix 1—figure 1*, the overall transcriptional initiation rate is then given by

$$\frac{\mathrm{dmRNA}}{\mathrm{d}t} = R \frac{\frac{[P]}{K_p} + \omega \frac{[P]\,[A]}{K_p\,K_a}}{1 + \frac{[P]}{K_p} + \frac{[A]}{K_a} + \omega \frac{[P]\,[A]}{K_p\,K_a}}. \tag{3}$$

From Appendix *Equation 1*, one can derive the Hill equation that is frequently used to model biophysical binding. In the limit of high cooperativity, $\omega \frac{[P]}{K_p} \gg 1$ and $\omega \frac{[A]}{K_a} \gg 1$ such that

$$p_{bound} = \frac{\omega \frac{[P]}{K_p} \frac{[A]}{K_a}}{1 + \omega \frac{[P]}{K_p} \frac{[A]}{K_a}}. \tag{4}$$

If we then define a new binding constant $K_a' = \frac{K_a K_p}{\omega [P]}$, we get the familiar Hill equation of order 1 with a binding constant $K_a'$

$$p_{bound} = \frac{\frac{[A]}{K_a'}}{1 + \frac{[A]}{K_a'}} \tag{5}$$

In general, any Hill equation of order $n$ can be derived from a more fundamental equilibrium thermodynamic model of simple activation possessing $n$ activator-binding sites in the appropriate limits of high cooperativity. Thus, any time a Hill equation is invoked, equilibrium thermodynamics is implicitly used, bringing with it all of the underlying assumptions described in Appendix section 6.5. This highlights the importance of rigorously grounding the assumptions made in any model of transcription, to better discriminate between the effects of equilibrium and non-equilibrium processes.

## 1.2 Thermodynamic MWC model

In the thermodynamic MWC model, we consider a system with 6 Bicoid-binding sites and 10 Zelda-binding sites. In addition, we allow for RNAP binding to the promoter.

In our model, the DNA can be in either an accessible or an inaccessible state. The difference in free energy between the two states is given by $-\Delta\varepsilon_{\mathrm{chrom}}$, where $\Delta\varepsilon_{\mathrm{chrom}}$ is defined as

$$\Delta\varepsilon_{\mathrm{chrom}} = \varepsilon_{\mathrm{accessible}} - \varepsilon_{\mathrm{inaccessible}}. \tag{6}$$

Here, $\varepsilon_{\mathrm{accessible}}$ and $\varepsilon_{\mathrm{inaccessible}}$ are the energies of the accessible and inaccessible states, respectively. A positive $\Delta\varepsilon_{\mathrm{chrom}}$ signifies that the inaccessible state is at a lower energy level, and therefore more probable, than the accessible state. We assume that all binding sites for a given molecular species have the same binding affinity, and that all accessible states exist at the same energy level compared to the inaccessible state. Thus, the total number of states is determined by the combinations of occupancy states of the three types of binding sites as well as the presence of the inaccessible, unbound state. We choose to not allow any transcription factor or RNAP binding when the DNA is inaccessible.

In this equilibrium model, the statistical weight of each accessible microstate is given by the thermodynamic dissociation constants $K_b$, $K_z$, and $K_p$ of Bicoid, Zelda, and RNAP, respectively. The statistical weight for the inaccessible state is $e^{\Delta\varepsilon_{\mathrm{chrom}}}$. We allow for a protein-protein interaction term $\omega_b$ between nearest-neighbor Bicoid molecules, as well as a pairwise cooperativity $\omega_{bp}$ between Bicoid and RNAP. However, we posit that Zelda does not interact directly with either Bicoid or RNAP. For notational convenience, we express the statistical weights in terms of the non-dimensionalized concentrations of Bicoid, Zelda, and RNAP, given by $b$, $z$ and $p$, respectively, such that, for example, $b \equiv \frac{[Bicoid]}{K_b}$. *Appendix 1—figure 2* shows the states and statistical weights for this thermodynamic MWC model, with all the associated parameters.

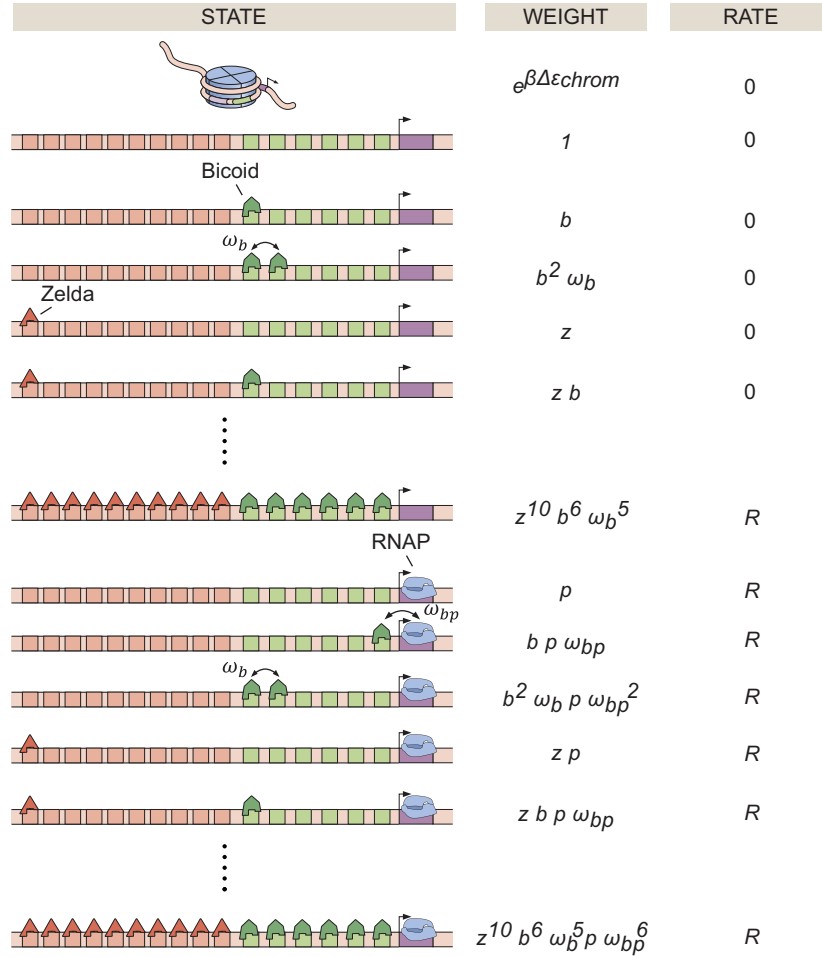

| STATE | WEIGHT | RATE |
|---|---|---|
| | $e^{\beta\Delta\varepsilon chrom}$ | 0 |
| | 1 | 0 |
| Bicoid | $b$ | 0 |
| $\omega_b$ | $b^2\,\omega_b$ | 0 |
| Zelda | $z$ | 0 |
| | $z\,b$ | 0 |
| | $z^{10}\,b^6\,\omega_b^5$ | R |
| RNAP | $p$ | R |
| $\omega_{bp}$ | $b\,p\,\omega_{bp}$ | R |
| $\omega_b$ | $b^2\,\omega_b\,p\,\omega_{bp}^2$ | R |
| | $z\,p$ | R |
| | $z\,b\,p\,\omega_{bp}$ | R |
| | $z^{10}\,b^6\,\omega_b^5 p\,\omega_{bp}^6$ | R |

**Appendix 1—figure 2.** States, weights, and rate of RNAP loading diagram for the thermodynamic MWC model, containing 6 Bicoid-binding sites, 10 Zelda-binding sites, and a promoter.

Incorporating all the microstates, we can calculate a statistical mechanical partition function, the sum of all possible weights, which is given by

$$Z = e^{\Delta\varepsilon_{chrom}/k_BT} + \underbrace{(1+z)^{10}}_{\text{Zelda binding}} \underbrace{\left(1+b+b^2\omega_b+...+b^6\omega_b^5+p+pb\omega_{bp}+...+pb^6\omega_b^5\omega_{bp}^6\right)}_{\text{Bicoid and RNAP binding}}. \tag{7}$$

Using the binomial theorem

$$(a+b)^N = \sum_{n=0}^{N}\binom{N}{n}a^n b^{N-n},$$

Appendix *Equation 7* can be expressed more compactly as

$$Z = e^{\Delta\varepsilon_{chrom}/k_BT} + (1+z)^{10}(1+p+\sum_{j=0,1}\sum_{i=1}^{6}\binom{6}{i}b^i\omega_b^{i-1}p^j\omega_{bp}^{ij}). \tag{8}$$

From this partition function, we can calculate $p_{bound}$, the probability of being in an RNAP-bound state. This term is given by the sum of the statistical weights of the RNAP-bound states divided by the partition function

$$p_{bound} = \frac{1}{Z}((1+z)^{10}p(1+\sum_{i=1}^{6}\binom{6}{i}b^i\omega_b^{i-1}\omega_{bp}^i)). \tag{9}$$

In this model, we once again assume that the transcription associated with each microstate is zero unless RNAP is bound, in which case the associated rate is $R$. Then, the overall transcriptional initiation rate is given by the product of $p_{bound}$ and $R$

$$\frac{d\mathrm{mRNA}}{dt} = R\frac{1}{Z}((1+z)^{10}p(1+\sum_{i=1}^{6}\binom{6}{i}b^i\omega_b^{i-1}\omega_{bp}^i)). \tag{10}$$

Note that since the MS2 technology only measures nascent transcripts, we can ignore the effects of mRNA degradation and focus on transcriptional initiation.

## 1.3 Constraining model parameters

The transcription rate $R$ of the RNAP-bound states can be experimentally constrained by making use of the fact that the *hunchback* minimal reporter used in this work produces a step-like pattern of transcription across the length of the fly embryo (*Figure 4C*, blue points). Since in the anterior end of the embryo, the observed transcription appears to level out to a maximum value, we assume that Bicoid binding is saturated in this anterior end of the embryo such that

$$p_{bound}(b \to \infty) \approx 1. \tag{11}$$

In this limit, Appendix *Equation 10* can be written as

$$\frac{d\mathrm{mRNA}}{dt} = R_{max} \approx R, \tag{12}$$

where $R_{max}$ is the maximum possible transcription rate. Importantly, $R_{max}$ is an experimentally observed quantity rather than a free parameter. As a result, the model parameter $R$ is determined by experimentally measurable quantity $R_{max}$.

The value of $p$ can also be constrained by measuring the transcription rate in the embryo's posterior, where we assume Bicoid concentration to be negligible. Here, the observed transcription bottoms out to a minimum level $R_{min}$ (*Figure 4C*, blue points), which we can connect with the model's theoretical minimum rate. Specifically, in this limit, $b$ approaches zero in Appendix *Equation 10* such that all Bicoid-dependent terms drop out, resulting in

$$\frac{d\mathrm{mRNA}}{dt} = R_{min} \approx \frac{1}{Z}((1+z)^{10}p)R_{max}, \tag{13}$$

where we have replaced $R$ with $R_{max}$ as described above. Next, we can express $p$ in terms of the other parameters such that

$$p \approx \frac{R_{min}(e^{\Delta\varepsilon_{chrom}/k_BT} + (1+z)^{10})}{(R_{max} - R_{min})(1+z)^{10}}. \tag{14}$$

Thus, $p$ is no longer a free parameter, but is instead constrained by the experimentally observed maximum and minimum rates of transcription $R_{max}$ and $R_{min}$, as well as our choices of $K_z$ and $\Delta\varepsilon_{chrom}$. In our analysis, $R_{max}$ and $R_{min}$ are calculated by taking the mean RNAP loading rate across all embryos from the anterior and posterior of the embryo respectively, extrapolated using the trapezoidal fitting scheme described in Appendix section 2.3.

Finally, we expand this thermodynamic MWC model to also account for suppression of transcription in the beginning of the nuclear cycle via mechanisms such as mitotic repression (Appendix section 3). To make this possible, we include a trigger time term $t_{MitRep}$, before which we posit that no readout of Bicoid or Zelda by *hunchback* is possible and the rate of RNAP loading is fixed at 0. For times $t > t_{MitRep}$, the system behaves according to Appendix *Equation 10*. Thus, given the constraints stemming from direct measurements of $R_{max}$ and $R_{min}$, the model has six free parameters: $\Delta\varepsilon_{chrom}$, $\omega_b$, $\omega_{bp}$, $K_b$, $K_z$, and $t_{MitRep}$. The final calculated transcription rate is then integrated in time to produce a predicted MS2 fluorescence as a function of time (Appendix section 2.2).

For subsequent parameter exploration of this model (Appendix section 5.1), constraints were placed on the parameters to ensure sensible results. Each parameter was constrained to be strictly positive such that:

- $\Delta\varepsilon_{chrom} > 0$
- $K_b > 0$
- $K_z > 0$
- $\omega_b > 0$
- $\omega_{bp} > 0$
- $0 < t_{MitRep} < 10$.

where an upper limit of 10 min was placed on the mitotic repression term to ensure efficient parameter exploration. This was justified because none of the observed transcriptional onset times in the data were larger than this value (*Figure 4D*).

## 2. Input-output measurements, predictions, and characterization

### 2.1 Input measurement methodology

Input transcription-factor measurements were carried out separately in individual embryos containing a eGFP-Bicoid transgene in a *bicoid* null mutant background (*Gregor et al., 2007b*) or a Zelda-sfGFP CRISPR-mediated homologous recombination at the endogenous *zelda* locus (*Hamm et al., 2017*). Over the course of nuclear cycle 13, the fluorescence inside each nucleus was extracted (details given in Section 'Image analysis'), resulting in a measurement of the nuclear concentration of each transcription factor over time. Six eGFP-Bicoid and three Zelda-sfGFP embryos were imaged.

Representative fluorescence traces of eGFP-Bicoid for a single embryo indicate that the magnitude of eGFP-Bicoid fluorescence decreases for nuclei located toward the posterior of the embryo (*Appendix 1—figure 3A*). Further, the nuclear fluorescence of eGFP-Bicoid at 8 min into nuclear cycle 13 (*Appendix 1—figure 3B*) exhibited the known exponential decay of Bicoid, with a mean decay length of 23.5% ± 0.6% of the total embryo length, consistent with but slightly different than previous measurements that suggested a mean decay length of 19.1% ± 0.8% (*Liu et al., 2013*). This discrepancy could stem, for example, from minor differences in acquisition from the laser-scanning two-photon microscope used in *Liu et al., 2013* versus the laser-scanning confocal microscope used here, such as differences in axial resolution (due both to different choices of objectives and the inherent differences in axial resolution of one-photon and two-photon fluorescence excitation processes). Nevertheless, the difference was minute enough that we felt confident in our eGFP-Bicoid measurements.

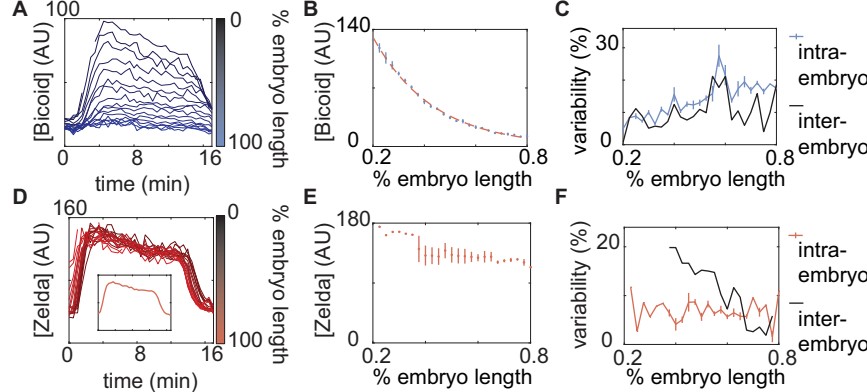

**Appendix 1—figure 3.** Measurements of input transcription-factor concentration dynamics . (**A**) Nuclear eGFP-Bicoid concentration as a function of time into nuclear cycle 13 across various positions along the anterior-posterior axis of a single embryo. (**B**) eGFP-Bicoid concentration at 8 min into nuclear cycle 13 as a function of position along the embryo averaged over all measured embryos (n = 6). The fit of the concentration profile to an exponential function (dashed line) results in a decay length of $23\% \pm 0.6\%$ embryo length. (**C**) Intra- and inter-embryo variability in eGFP-Bicoid

*Appendix 1—figure 3 continued on next page*

*Appendix 1—figure 3 continued*

nuclear fluorescence along the anterior-posterior axis. (**D**) Zelda-sfGFP concentration as a function of time into nuclear cycle 13 across various anterior-posterior positions of a single embryo. (D, inset) Zelda-sfGFP concentration averaged over the data shown in D. (**E**) Zelda-sfGFP concentration at 8 min into nuclear cycle 13 as a function of position along the anterior-posterior axis of the embryo averaged over all measured embryos (n = 3). Note that anterior of 40% and posterior of 77.5% only a single embryo was measured; no error bars were calculated. (**F**) Intra- and inter-embryo variability in Zelda-sfGFP nuclear fluorescence along the anterior-posterior axis. (B,E, error bars represent standard error of the mean nuclear fluorescence, measured across embryos; C,F, error bars represent standard error of the mean intra-embryo variability, measured across embryos.).

Intra-embryo variability in eGFP-Bicoid nuclear fluorescence, defined by the standard deviation across nuclei within a single embryo divided by the mean, was in the range of 10–30%, as was the inter-embryo variability, defined by the standard deviation of the mean amongst nuclei, across different embryos (*Appendix 1—figure 3C*, blue and black, respectively). Six separate eGFP-Bicoid embryos were measured.

Similarly, representative fluorescence time traces of Zelda-sfGFP for a single embryo are shown in *Appendix 1—figure 3D*. Unlike the eGFP-Bicoid profile, the Zelda-sfGFP nuclear fluorescence was approximately uniform across embryo position (*Appendix 1—figure 3E*), consistent with previous fixed-tissue measurements (*Staudt et al., 2006*; *Liang et al., 2008*). Intra-embryo variability in Zelda-sfGFP nuclear fluorescence was very low (less than 10%), whereas inter-embryo variability was relatively higher, up to 20% (*Appendix 1—figure 3F*, red and black, respectively). Three separate Zelda-sfGFP embryos were measured.

Due to the consistency of Zelda-sfGFP nuclear fluorescence, we assumed the Zelda profile to be spatially uniform in our analysis, and thus created a mean Zelda-sfGFP measurement for each individual embryo by averaging all mean nuclear fluorescence traces in space across the anterior-posterior axis of the embryo (*Appendix 1—figure 3D*, inset). This mean measurement was used as an input in the theoretical models. However, we still retained inter-embryo variability in Zelda, as described below.

To combine multiple embryo datasets as inputs to the models explored throughout this work, the fluorescence traces corresponding to each dataset were aligned at the start of nuclear cycle 13, defined as the start of anaphase. Because each embryo may have possessed slightly different nuclear cycle lengths and/or experimental sampling rates (due to the manual realignment of the z-stack to keep nuclei in focus), the individual datasets were not combined in order to create average Bicoid and Zelda profiles across embryos. Instead, a simulation and model prediction were performed for each combination of measured input Bicoid and Zelda datasets, essentially an in silico experiment covering a portion of the full embryo length. In all, outputs at each embryo position were predicted in at least three separate simulations. Subsequent analyses used the mean and standard error of the mean of these amalgamated simulations. With six GFP-Bicoid datasets and three Zelda-GFP datasets, there were 18 unique combinations of input embryo datasets; for a single set of parameters used in a particular model, each derived metric (e.g. $t_{on}$) was calculated using predicted outputs from each of the 18 possible input combinations. This procedure provided full embryo coverage and resulted in a distribution of the derived metric for that particular set of parameters. From this distribution, the mean and standard error of the mean were calculated, leading to the lines and shading in plots such as *Appendix 1—figure 6*.

## 2.2 MS2 fluorescence simulation protocol

To calculate a predicted MS2 fluorescence trace from measured Bicoid and Zelda inputs for a given theoretical model, we utilized a simple model of transcription initiation, elongation, and termination. First, the dynamic transcription-factor concentrations were used as inputs to each of the theoretical models outlined throughout the paper. These models generated a rate of RNAP loading as a function of time and space across the embryo over the course of nuclear cycle 13.

For each position along the anterior-posterior axis, the predicted rate of RNAP loading was integrated over time to generate a predicted MS2 fluorescence trace. Given the known reporter construct length $L$ of 5.2 kb (*Garcia et al., 2013*), we assume that RNAP molecules are loaded onto the

start of the gene at a rate $R(t)$ predicted by the particular model under consideration (**Appendix 1—figure 4**; see Appendix sections 1.2, 6.1, 7.1, and 8.1 for model details). Each RNAP molecule traverses the gene at a constant velocity $v$ of 1.54 kb/min, as measured experimentally by **Garcia et al., 2013**. With these numbers, we calculate an elongation time

$$t_{elon} = \frac{L}{v}. \tag{15}$$

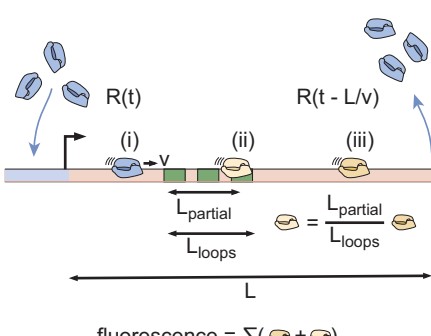

fluorescence = $\sum(\text{⬬} + \text{⬬})$

**Appendix 1—figure 4.** MS2 fluorescence calculation protocol. RNAP molecules load onto the reporter gene at a time-dependent rate $R(t)$, after which they elongate at a constant velocity $v$. Upon reaching the end of the gene after a length $L$ has been transcribed, they are assumed to terminate and disappear instantly, given by the time-shifted rate $R(t - \frac{L}{v})$. The time-dependent MS2 fluorescence is calculated by summing the contributions of RNAP molecules that are located before, within, or after the MS2 stem loop sequence (i, ii, and iii, respectively).

Finally, we assume that upon reaching the end of the reporter gene, the RNAP molecules terminate and disappear instantly such that they no longer contribute to spot fluorescence.

The MS2 fluorescence signal reports on the number of RNAP molecules actively occupying the gene at any given time and, under the assumptions outlined above, is given by the integral

$$F(t) = \alpha \int_0^t (R(t') - R(t' - t_{elon}))dt', \tag{16}$$

where $F(t)$ is the predicted fluorescence value, $R(t)$ is the RNAP loading rate predicted by each specific model, $R(t - t_{elon})$ is the time-shifted loading rate that accounts for RNAP molecules finishing transcription at the end of the gene, and $\alpha$ is an arbitrary scaling factor to convert from absolute numbers of RNAP molecules to arbitrary fluorescence units. The predicted value $F(t)$ was scaled by $\alpha$ to match the experimental data.

The final predicted MS2 signal was modified in a few additional ways. First, any RNAP molecule that had not yet reached the position of the MS2 stem loops had its fluorescence value set to zero (**Appendix 1—figure 4, i**), since only RNAP molecules downstream of the MS2 stem loop sequence exhibit a fluorescent signal. Second, RNAP molecules that were only partially done elongating the MS2 stem loops contributed a partial fluorescence intensity, given by the ratio of the distance traversed through the stem loops to the total length of the stem loops

$$F_{partial} = \frac{L_{partial}}{L_{loops}},$$

where $F_{partial}$ is the partial fluorescence contributed by an RNAP molecule within the stem loop sequence region, $L_{partial}$ is the distance within the stem loop sequence traversed, and $L_{loops}$ is the length of the stem loop sequence (**Appendix 1—figure 4**, ii). For this reporter construct, the length of the stem loops was approximately $L_{loops} = 1.28 kb$. RNAP molecules that had finished transcribing the MS2 stem loops contributed the full amount of fluorescence (**Appendix 1—figure 4**, iii). Finally, to make this simulation compatible with the trapezoidal fitting scheme in Appendix section 2.3, we included a falling signal at the end of the nuclear cycle, achieved by setting $R(t) = 0$ after 17 min into the nuclear cycle and thus preventing new transcription initiation events.

Given the predicted MS2 fluorescence trace, the rate of RNAP loading and $t_{on}$ were extracted with the fitting procedure used on the experimental data (Appendix section 2.3).

## 2.3 Extracting initial RNAP loading rate and transcriptional onset time

To extract the initial rate of RNAP loading and the transcriptional onset time $t_{on}$ used in the data analysis, we fit both the experimental and calculated MS2 signals to a constant loading rate model, the trapezoidal model (*Garcia et al., 2013*).

The trapezoidal model provides a heuristic fit of the main features of the MS2 signal by assuming that the RNAP loading rate is either zero or some constant value $r$ (*Appendix 1—figure 5A*). At time $t_{on}$, the loading rate switches from zero to this constant value $r$, producing a linear rise in the MS2 signal. After the elongation time $t_{elong}$, the loading of new RNAP molecules onto the gene is balanced by the loss of RNAP molecules at the end of the gene, producing a plateau in the MS2 signal. Finally, at the end of the nuclear cycle, transcription ceases at $t_{off}$ and the RNAP loading rate switches back to zero, producing the falling edge of the MS2 signal and completing the trapezoidal shape. Because we only consider the initial dynamics of transcription in the nuclear cycle in this investigation, we do not explore the behavior of $t_{off}$.

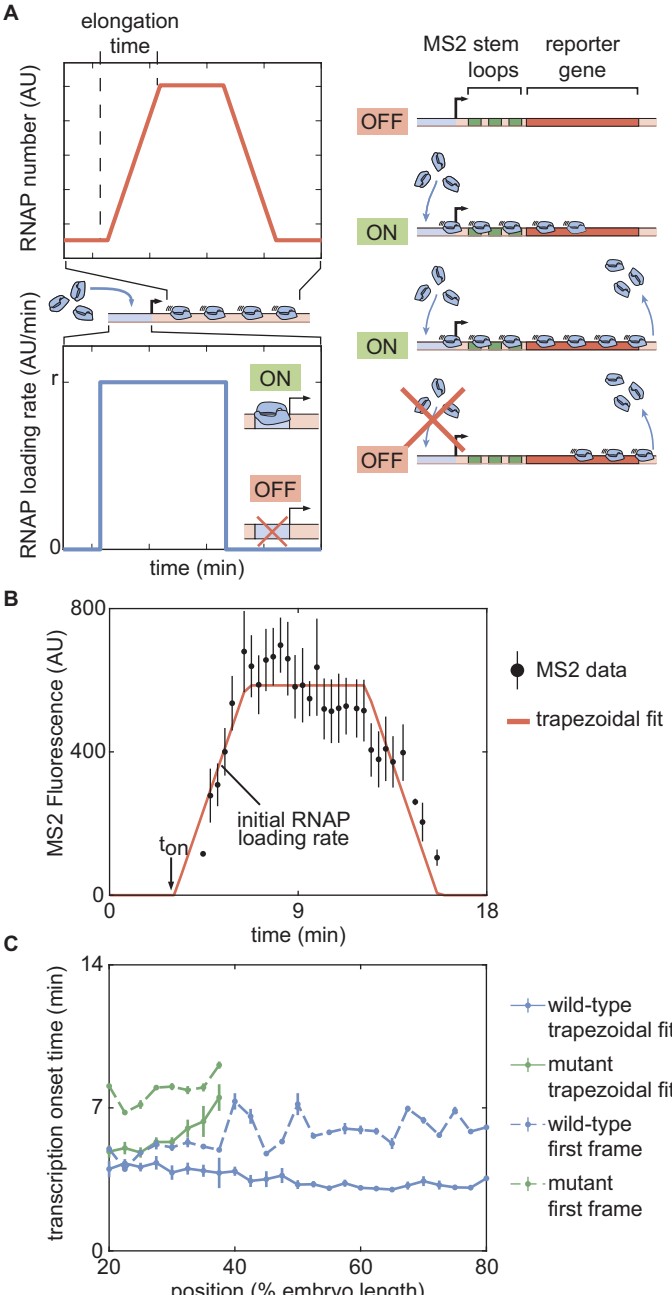

**Appendix 1—figure 5.** Outline of fitting to the trapezoidal model of transcription. (**A**) The trapezoidal model of transcription, where transcription begins at an onset time $t_{on}$ and loads RNAP molecules with a constant rate $r$. (**B**) Results of fitting the MS2 fluorescence data from a position in a single embryo to the trapezoidal model to extract $t_{on}$ and the initial rate of RNAP loading. (**C**) Comparison of inferred $t_{on}$ values between the trapezoidal model (solid lines) and using the time of first detection of signal in a fluorescence spot (dashed lines) for both wild-type and *zelda⁻* backgrounds. (B, error bars are standard error of the mean averaged over multiple nuclei within the embryo, for data in a wild-type background at 50% along the embryo length; C, error bars are standard error of the mean, averaged across embryos).

*Appendix 1—figure 5B* shows the results of fitting the mean MS2 fluorescence from a narrow window within a single embryo to the trapezoidal model. With this fit, we can extract the initial rate

of RNAP loading (given by the initial slope) as well as $t_{on}$ (given by the intercept of the fit onto the x-axis).

As a consistency check, the $t_{on}$ values extrapolated from the trapezoidal fit of the data were compared with the experimental time points at which the first MS2 spots were observed for both the wild-type and *zelda⁻* mutant experiments (*Appendix 1—figure 5C*). Due to the detection limit of the microscope, this latter method reports on the time at which a few RNAP molecules have already begun transcribing the reporter gene, rather than a 'true' transcriptional onset time. Using the first frame of spot detection yields similar trends to the trapezoidal fits, except that the measured first frame times are systematically larger, especially in the mutant data. Additionally, utilizing the first frame of detection to measure $t_{on}$ appears to be a noisier method, likely because the actual MS2 spots cannot be observed below a finite signal-detection limit, whereas the extrapolated $t_{on}$ from the trapezoidal fit corresponds to a 'true' onset time below the signal-detection limit. For this reason, we decided to rely on the trapezoidal fit to extract $t_{on}$, rather than using the first frame of spot detection.

## 3. Mitotic repression is necessary to recapitulate Bicoid- and Zelda-mediated regulation of *hunchback* using the thermodynamic MWC model

As described in Section 'The thermodynamic MWC model fails to predict activation of *hunchback* in the absence of Zelda' of the main text, a mitotic repression window was incorporated into the thermodynamic MWC model (Appendix section 1.2) in order to explain the observed transcriptional onset times of *hunchback*. Here, we justify and explain this theoretical modification in greater detail.

*Appendix 1—figure 6A and B* depicts the experimentally observed initial rates of RNAP loading and $t_{on}$ across the length of the embryo (blue points) for the wild-type background. After constraining the maximum and minimum theoretically allowed rates of RNAP loading (Appendix section 1.3), we attempted to simultaneously fit the thermodynamic MWC model to both the rate of RNAP loading and $t_{on}$.

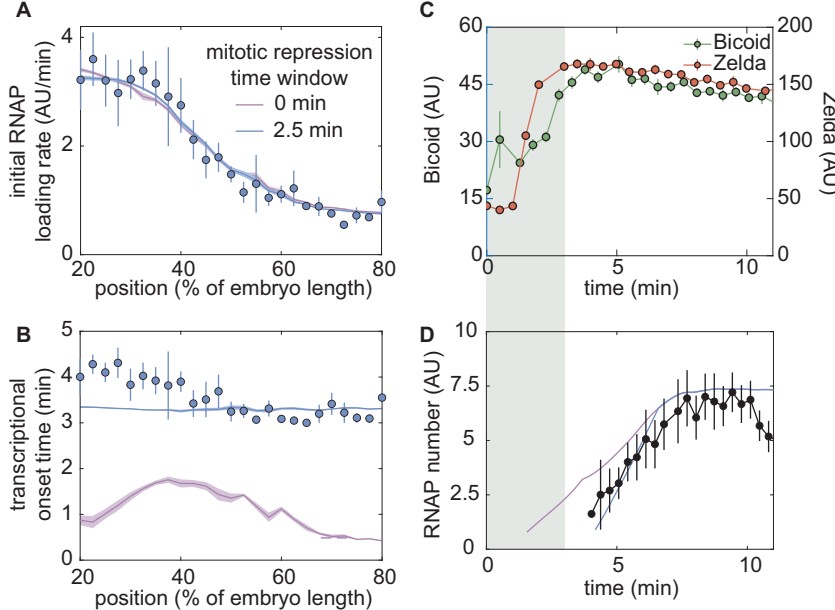

**Appendix 1—figure 6.** A thermodynamic MWC model including mitotic repression can recapitulate *hunchback* regulation by Bicoid and Zelda. (**A**) Measured initial rates of RNAP loading and (**B**) $t_{on}$ (blue points) across the length of the embryo, compared to fits to the thermodynamic MWC model with and without accounting for mitotic repression (blue and purple curves, respectively). (**C**) Nuclear concentration dynamics of Bicoid and Zelda with proposed mitotic repression window (gray shading). (**D**) Predicted MS2 dynamics with no mitotic repression term or a 3 min mitotic repression

*Appendix 1—figure 6 continued on next page*

*Appendix 1—figure 6 continued*

window compared to experimental measurements. (A,B, solid lines indicate mean predictions of the model and shading represents standard error of the mean, while points indicate data and error bars represent the standard error of the mean, across 11 embryos; C, D, data from single embryos at 45% of the embryo length with error bars representing the standard error of the mean across nuclei, errors in model predictions in D were negligible and are obscured by the prediction curve; fitted parameter values for a 3 min mitotic repression window were $\Delta\epsilon_{chrom} = 10\ k_BT$, $K_b = 34\ AU$, $K_z = 500\ AU$, with different arbitrary fluorescent units for Bicoid and Zelda, $\omega_b = 10$, $\omega_{bp} = 0.4$, for a model assuming six Bicoid-binding sites.).

The fit results demonstrate that while the thermodynamic MWC model can recapitulate the measured step-like rate of RNAP loading at *hunchback* (*Appendix 1—figure 6A*, purple line), it fails to predict the $t_{on}$ throughout the embryo (*Appendix 1—figure 6B*, purple line; see Appendix sections 2.2 and 2.3 for details about experimental and theoretical calculations). This model yields values of $t_{on}$ that are much smaller than those experimentally observed, a trend that holds throughout the length of the embryo. This disagreement becomes more evident when comparing the output transcriptional activity reported by the measured MS2 fluorescence with the input concentrations of Bicoid and Zelda. Specifically, the Bicoid and Zelda concentration measurements at 45% along the embryo, shown for a single embryo in *Appendix 1—figure 6C*, are used in conjunction with the previously mentioned best-fit model parameters to predict the output MS2 signal at the same position. This prediction can then be directly compared with experimental data (*Appendix 1—figure 6D*, purple line vs. black points, respectively). Although the model predicts that transcription will commence around 1 min after anaphase due to the concurrent increase in the Bicoid and Zelda concentrations, the observed MS2 signal begins to increase around 4 min after anaphase (*Appendix 1—figure 6D*). As a result, the predicted transcriptional dynamics in *Appendix 1—figure 6D* are systematically shifted in time with respect to the observed data.

The observed disagreement in $t_{on}$ suggests that in this model, transcription is prevented from starting at the time dictated solely by the increase of Bicoid and Zelda concentrations. While we speculate that this effect could stem from processes such as RNAP escape from the promoter, DNA replication at the start of the cell cycle, and post-mitotic nucleosome clearance from the promoter, we choose not to commit to a detailed molecular picture and instead ascribe this transcriptional refractory period at the beginning of the nuclear cycle to mitotic repression, the observation that the transcriptional machinery cannot operate during mitosis (*Shermoen and O'Farrell, 1991*; *Gottesfeld and Forbes, 1997*; *Parsons and Spencer, 1997*; *Garcia et al., 2013*). To account for this phenomenon, we revised our thermodynamic MWC model by stating that *hunchback* can only read out the inputs and begin transcription after a specified mitotic repression time window following the previous anaphase (Appendix section 1.3).

Since we expect mitotic repression to operate independently of position along the length of the embryo (*Shermoen and O'Farrell, 1991*), we assumed that the duration of mitotic repression was uniform throughout the embryo. After incorporating a uniform 3 min mitotic repression window into the thermodynamic MWC model (*Appendix 1—figure 6C and D*, grey-shaded region), the model successfully recapitulates $t_{on}$ throughout the embryo (*Appendix 1—figure 6B and D*, blue curves), while still explaining the observed rates of RNAP loading (*Appendix 1—figure 6A*, blue curve). Thus, once mitotic repression is accounted for, the thermodynamic MWC model based on statistical mechanics can quantitatively recapitulate the regulation of *hunchback* transcription by Bicoid and Zelda.

## 4. The effect of the *zelda⁻* background on the Bicoid concentration spatiotemporal profile

Our models rest on the assumption that the Bicoid gradient remains unaltered regardless of whether these measurements are made in the wild-type or *zelda⁻* backgrounds. To confirm this assumption, we measured eGFP-Bicoid concentrations in a *zelda⁻* background. These flies were heterozygous for eGFP-labeled Bicoid and for wild-type Bicoid, resulting in roughly 50% of total Bicoid being labeled with eGFP. As shown in *Appendix 1—figure 7A and B*, the resultant eGFP-Bicoid nuclear fluorescence levels in nuclear cycle 13 in the *zelda⁻* background (red) were roughly half the magnitude of

the equivalent measurements in the wild-type background (blue), a trend that held both in time and along the embryo. After doubling the heterozygote eGFP-Bicoid nuclear fluorescence measurements to rescale them (*Appendix 1—figure 7B*, black), the two eGFP-Bicoid curves became similar, although the *zelda⁻* eGFP-Bicoid values were systematically lower than in the wild-type background. The normalized difference, defined as the absolute value of the difference between the wild-type and *zelda⁻* profiles at each position in the embryo divided by the value of the wild-type profile at the position, averaged across all measured positions, was $15\% \pm 2\%$. This value is within the range of the inter-embryo variability of eGFP Bicoid in wild-type background embryos (*Appendix 1—figure 3C*). Measuring the decay length of the eGFP-Bicoid profile in the *zelda⁻* background also yielded a slightly different result: $21\% \pm 1\%$ of the total embryo length, as opposed to $23.5\% \pm 0.6\%$ in the wild-type background (dashed curves, see also *Appendix 1—figure 3B*).

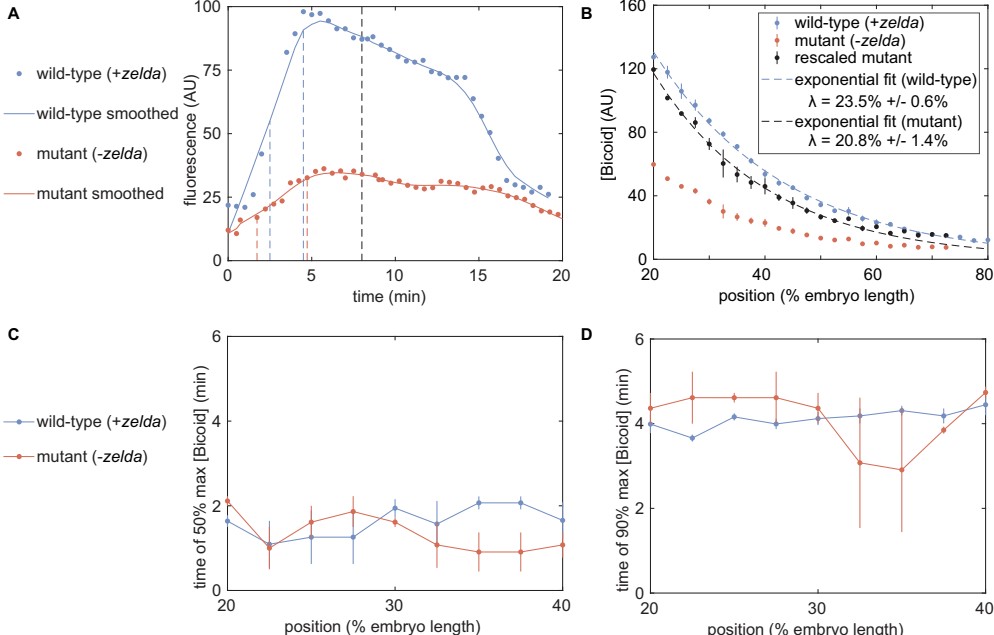

**Appendix 1—figure 7.** Comparison of eGFP-Bicoid measurements in wild-type and *zelda⁻* embryos. (**A**) Example mean nuclear eGFP-Bicoid concentrations for a single embryo at 30% along the embryo's length, for wild-type (blue) and *zelda⁻* (red) backgrounds. Datapoints are raw values and lines are smoothed results. The raw fluorescence at 8 min into the nuclear cycle, indicated by the black dashed line, is calculated to yield (**B**). The blue and red dashed lines correspond to the times to reach 50% and 90% of the maximum fluorescence for the smoothed wild-type and *zelda⁻* signals, respectively. (**B**) eGFP-Bicoid measurements in wild-type (blue) and *zelda⁻* mutant embryos (red), along with rescaled mutant profiles (black). Fits to an exponentially decaying function yield decay lengths in each background (blue and black dashed curves). (**C, D**) Time to reach 50% and 90% of maximum nuclear eGFP-Bicoid fluorescence for wild-type (blue) and *zelda⁻* (red) backgrounds. A total of n = 3 embryos were measured in the *zelda⁻* background, compared to n = 6 for wild-type. All error bars are standard error of mean across embryos.

Having compared the spatial profile of Bicoid in both backgrounds, we then contrasted the dynamics of nuclear Bicoid import. To quantify this analysis, we calculated the time to reach 50% and 90% of the maximum eGFP-Bicoid fluorescence signal for wild-type and *zelda⁻* embryos, at each position along the anterior-posterior axis (*Appendix 1—figure 7A*, blue and red dashed lines). Because the raw fluorescence signals were noisy enough to confound this calculation, we first smoothed the signals using a moving average filter of ten datapoints (*Appendix 1—figure 7A*, lines).

*Appendix 1—figure 7C and D* show the times to reach 50% and 90% of maximum fluorescence for the anterior positions in both embryo backgrounds, where transcription was observed, respectively. In both backgrounds, the 50% and 90% times are similar to within approximately 1 min, indicating that the dynamics of nuclear eGFP-Bicoid at the start of nuclear cycle 13 are quantitatively

comparable. Thus, we concluded that differences in transcription between the two embryo backgrounds do not stem from differences in Bicoid dynamics.

In summary, the dynamics of nuclear Bicoid concentration are quantitatively comparable in both wild-type and $zelda^-$ backgrounds, whereas the overall Bicoid concentration is slightly lower in the $zelda^-$ case. Nevertheless, these differences in concentration would have a negligible effect on our overall conclusions: in the context of our models, an overall rescaling in the magnitude of the Bicoid gradient between the wild-type and $zelda^-$ backgrounds can be compensated by a corresponding rescaling in the dissociation constant of Bicoid, $K_b$. Because our systematic exploration of theoretical models considers many possible parameter values (Appendix section 5.1), this rescaling has no effect on our conclusion that the equilibrium models are insufficient to explain the $zelda^-$ data. As a result, and given that our statistics for the wild-type eGFP-Bicoid data consisted of more embryos than the data for the $zelda^-$ background, we used this wild-type data in our analyses as an input to both the wild-type and $zelda^-$ model calculations.

## 5. State-space exploration of theoretical models

### 5.1 General methodology of state-space exploration

To help visualize the limits of our models, we collapsed our observations onto a three-dimensional state space, following a method similar to that described in *Estrada et al., 2016*. In this space, the x-axis was the average $t_{on}$ delay. This magnitude was computed by integrating the $t_{on}$ across 20% to 37.5% of the embryo length, corresponding to the range in which both wild-type and $zelda^-$ experiments exhibited transcription in at least 30% of observed nuclei (*Appendix 1—figures 8A* and *5A*, as defined in *Equation 5*). The offset in $t_{on}$ at 20% embryo length (*Equation 4*) was the y-axis in the state space. The z-axis was given by the average initial rate of RNAP loading between 20% and 37.5% of the embryo length (*Appendix 1—figure 8B*).

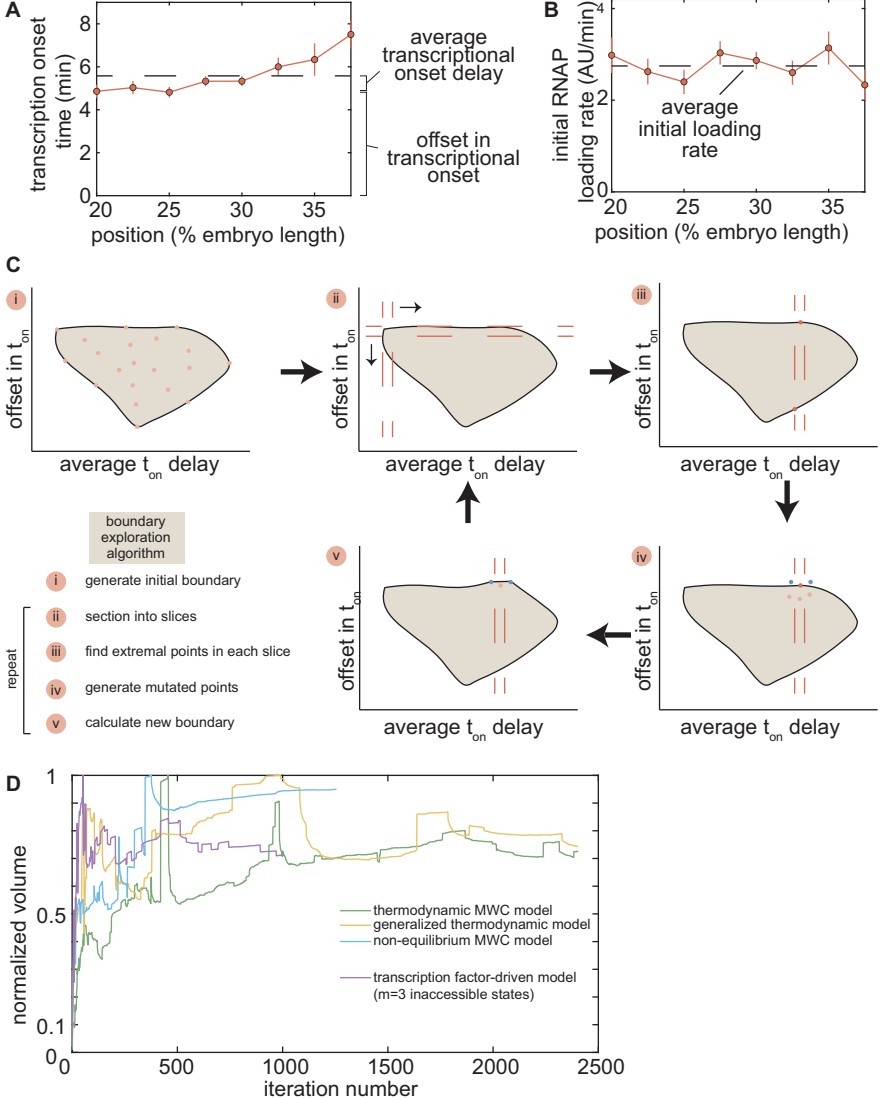

**Appendix 1—figure 8.** Description of state-space metrics and boundary-exploration algorithm. (**A**) Representative average $t_{on}$ delay (black dashed line) and $t_{on}$ offset for the *zelda⁻* background data in *Figure 4D*. (**B**) Average initial RNAP loading rate for the *zelda⁻* background data in *Figure 4C*. (**C**) Overview of the boundary-exploration algorithm for an example state space containing two dimensions. (**i**) A set of 50 points with random input parameters generates an initial state space of the investigated model. (**ii**) The space is sectioned into 10 horizontal and 10 vertical slices. (**iii**) The extremal points of each slice are found. (**iv**) For each extremal point, five new points are generated with input parameters in a small neighborhood around the parameters of this extremal point. (**v**) The new space is plotted with these new points, and steps (**ii**) - (**iv**) are repeated. (**D**) Normalized volume of state-space domain of each model investigated in this work as a function of algorithm iteration number. All volumes approach a steady value, indicating convergence.

Combined, the average $t_{on}$ delay, $t_{on}$ offset, and average initial RNAP loading rate provide a simplified description of our data as well as of our theoretical predictions. Each theoretical model inhabits a finite region in this three-dimensional state space, which we can calculate by systematically varying model parameters. *Appendix 1—figure 8A and B* show an example of how the three parameters are calculated using the *zelda⁻* background data presented in *Figure 4C and D* (red points) in the main text.

Due to the large number of parameters in each model explored, the corresponding state-space boundaries were generated by efficient sampling of the underlying high-dimensional parameter

space. Although in actuality the state space contained three dimensions, we illustrate the sampling process here with a two-dimensional example, using only the offset and average delay in transcriptional onset time, for ease of visualization (*Appendix 1—figure 8C*). The methodology is similar to the one described in *Estrada et al., 2016*. Briefly, a starting set of 50 points was generated, each with a randomized set of initial parameters, the specifics of which depended on the model being tested (*Appendix 1—figure 8Ci*). The state space was sectioned into 100 slices along each orthogonal axis (*Appendix 1—figure 8Cii*). The most extremal points in each slice were found, resulting in two extremal values each for the $t_{on}$ offset and average $t_{on}$ delay (*Appendix 1—figure 8Ciii*). For each of these points, a new set of five points was generated using random parameters within a small neighborhood of the seed points determined by the extremal points of the previous iteration (*Appendix 1—figure 8Civ*). These new points were plotted; some of these points may be more extreme than the previous set of points. Steps ii-iv were iterated, resulting in a growing boundary over time (*Appendix 1—figure 8Cv*). This algorithm was run in the full three-dimensional state space, where 100 three-dimensional columns along the orthogonal xy-, yz-, and xz-planes were used instead of two-dimensional slices.

Constraints imposed by the data were used to filter unrealistic results and ensure rapid convergence of the algorithm. First, if the simulated average $t_{on}$ delay was less than $-0.5$ min or greater than 2 min, the point was filtered out. This removal was justified experimentally, since none of the observed average $t_{on}$ delays were outside of this range (*Figure 5A*). Second, if the simulated average initial loading rate was smaller than 1 AU/min or greater than 4 AU/min, the point was also filtered out. This was also justified experimentally, since none of the observed initial RNAP loading rates between 20% and 37.5% embryo position lay outside this range (*Figure 4C*). Points that fulfilled these constraints were retained for the next iteration of the algorithm. This process was repeated until the resulting space of points no longer grew appreciably, resulting in an estimate of the size and shape of the state space for each of the models presented in Appendix sections 1.2, 6.1, 7.1, and 8.1.

To determine whether the algorithm had indeed converged, the total volume of each model's region in state space was tracked with each iteration number. If the algorithm worked well, then this volume would approach some maximum value. *Appendix 1—figure 8D* shows the volume of the state space corresponding to each model presented in this work (normalized by the volume at the final iteration number) as a function of the iteration number. Each model converged to a finite value, indicating that the parameter space occupied by the models had been thoroughly explored.

## 5.2 State space exploration with the thermodynamic MWC model

*Appendix 1—figure 9A* and *Appendix 1—video 1* show the resulting three-dimensional state space for the thermodynamic MWC model (green), as well as all of the theoretical models considered here. We plotted the wild-type and *zelda*⁻ data on the same state space, represented as small ellipsoids of uncertainty. Any successful model must occupy a region that overlaps both the wild-type and *zelda*⁻ data.

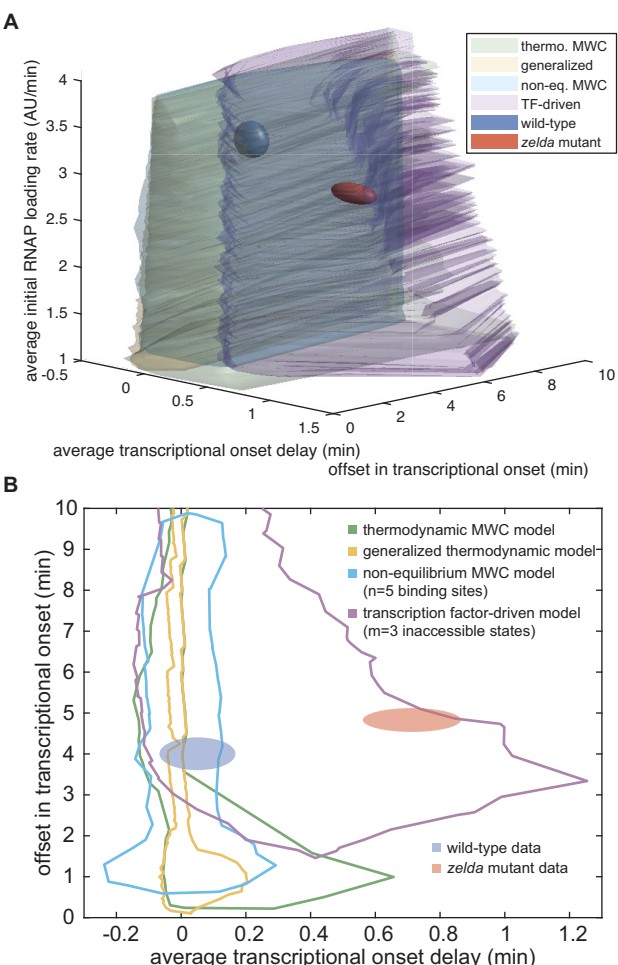

**Appendix 1—figure 9.** Exploration of state space. (**A**) Three-dimensional state-space exploration, showing the extents of state space of the wild-type (blue) and *zelda*⁻ (red) data as well as of various models explored in the main text. See also *Appendix 1—video 1* for an more comprehensive representation of our results. (**B**) Two-dimensional state-space exploration, created by projecting the three-dimensional state space in (**A**) for average initial loading rate values between 2.5 and 3.6 AU/min onto the xy-plane corresponding to the average $t_{on}$ delay and $t_{on}$ offset. Volumes (**A**) and areas (**B**) covered by the experimental data represent the standard error of the mean.

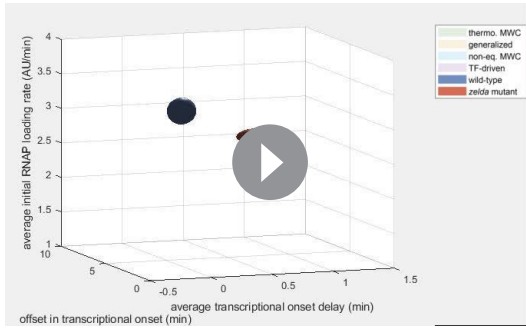

*Appendix 1—video 1 continued on next page*

*Appendix 1—video 1 continued*

**Appendix 1—video 1.** Exploration of three-dimensional space consisting of average initial RNAP loading rate and offset and average delay in transcriptional onset time. The models explored in the main text inhabit domains in this space, whereas the wild-type and *zelda*⁻ data inhabit ellipsoids of uncertainty. Whereas the thermodynamic MWC, generalized thermodynamic, and non-equilibrium MWC model with up to five Bicoid-binding sites cannot explain the *zelda*⁻ data, the transcription-factor-driven model with three inaccessible states can adequately encompass both datasets.
https://elifesciences.org/articles/56429#A1video1

As shown in *Appendix 1—figure 9A* and *Appendix 1—video 1*, the state space corresponding to the thermodynamic MWC model fails to overlap with the *zelda*⁻ data. To more clearly reveal this disagreement, this three-dimensional state space was projected onto the xy-plane, the space incorporating the average $t_{on}$ delay and $t_{on}$ offset information. To do this projection, we noticed that both the wild-type and *zelda*⁻ data only occupied average initial loading rate values between 2.5 AU/min and 3.6 AU/min (*Appendix 1—figure 9A* and *Appendix 1—video 1*). As a result, only points in that range of initial loading rates were retained for the projection. The resulting two-dimensional representation of our exploration is shown in *Appendix 1—figure 9B*. Even in this simplified representation, the failure of the thermodynamic MWC model (*Appendix 1—figure 9B*, green) is evident. Therefore, we utilized this representation throughout the main text and Appendix (*Figures 5C*, *6B* and *7D* and *Appendix 1—figures 13* and *14C*).

## 6. Failures and assumptions of thermodynamic models of transcription

### 6.1 Generalized thermodynamic model

The generalized thermodynamic model is an extension of the thermodynamic MWC model presented in Appendix section 1.2. For extra generality, we assume the presence of twelve Bicoid binding sites and one RNAP binding site, but do not include the action of Zelda since the objective was to attempt to recapitulate the *zelda*⁻ mutant experimental data. We still allow for an inaccessible DNA state.

In this generalized model, the weight of each microstate can be arbitrary, rather than determined by underlying biophysical parameters. Since $p_{bound}$ only depends on whether RNAP is bound, there is no need to distinguish between different microstates that have the same number of Bicoid molecules bound: the arbitrary coefficients allow separate microstates to effectively be combined together into the same weight. Thus, each microstate corresponds only to the overall number of bound molecules, regardless of binding site ordering. With 12 Bicoid sites, in addition to the inaccessible state, there are 27 total microstates and 26 free parameters describing the weights of each state (with the accessible, unbound microstate normalized to unity). Like with the thermodynamic MWC model, we assume that transcription only occurs when RNAP is bound, with the same constrained maximum rate of RNAP loading $R_{max}$. However, since the weights of each microstate are arbitrary, we no longer have a variable $p$ that can be constrained by $R_{min}$ like in Appendix *Equation 14*.

This generalized model is much more powerful than the thermodynamic MWC model due to a lack of coupling between individual microstate weights. Whereas in the previous model the underlying parameters $K_b$ and $\omega_b$ caused similar microstates to be related mathematically, now the statistical weights for each microstate are completely independent. Physically, this scenario can arise due to, for example, higher-order cooperativities or non-identical binding energies between binding sites (*Estrada et al., 2016*).

The partition function in this generalized thermodynamic model is given by the polynomial

$$Z = p_{inacc} + \sum_{r=0}^{1} \sum_{n=0}^{12} P_{r,n}[Bicoid]^n, \tag{17}$$

where $p_{inacc}$ is the weight of the inaccessible state and $P_{r,n}$ is the weight of the accessible state with $r$

RNAP molecules bound and $n$ Bicoid molecules bound. The overall transcriptional initiation rate is now

$$\frac{d\mathrm{mRNA}}{dt} = \frac{1}{Z}\left(\sum_{n=0}^{12} P_{1,n} R [Bicoid]^n\right), \tag{18}$$

where $P_{1,n}$ is the statistical weight of each RNAP-bound state and $R$ is the corresponding rate of transcriptional initiation. Note that, as described above, $R$ is still equal to $R_{max}$, the constraint described in Appendix section 1.3, but we no longer use the $R_{min}$ constraint.

The resulting rate of transcriptional initiation is integrated over time to produce a simulated MS2 fluorescence trace using the same procedure as for the models presented in Appendix sections 1.2, 7.1, and 8.1 (see Appendix section 2.2 for details). As with the thermodynamic MWC model, we allow for a mitotic repression time window to account for the lack of transcription early in the nuclear cycle.

## 6.2 Generalized thermodynamic model state space exploration

Due to the high-dimensional parameter space of the generalized thermodynamic model, constraints were necessary to efficiently explore this parameter space (Appendix section 5.1). These constraints were placed on the values of the individual microstate weights $P_{r,n}$, based on dimensional analysis and heuristic arguments. Specifically, each weight $P_{r,n}$ is derived from a product of binding constants $K_d$ for either Bicoid or RNAP, pairwise cooperativity parameters $\omega$, and higher order cooperativity terms. For the purposes of these parameter constraints, we only consider the $K_d$s and $\omega$s, and ignore constraints on higher-order cooperativities. In principle, each Bicoid-binding site possesses a unique $K_d$ and protein-protein interaction terms $\omega$ with other Bicoid molecules and/or with RNAP. However, as described below, these biophysical parameters, once non-dimensionalized, can be constrained to reasonable values by scaling relations through a simple bounding scheme.

For illustrative purposes, consider the microstate with RNAP and one Bicoid molecule bound. Its weight depends on two independent binding constants $p, b$ and a cooperativity term between RNAP and Bicoid $\omega_{bp}$. First, we assume that the $p, b$ terms are non-dimensionalized, that is they take the form $p = [RNAP]/K_p$ and $b = [Bicoid]/K_b$. Although the two individual $p, b$ terms are in principle different since RNAP and Bicoid have can different binding energies, we can be generous about the constraints and assume that the non-dimensionalized forms are both bounded below and above by 0 and 1000, respectively. This strategy is justified by assuming that neither RNAP nor Bicoid exist in concentrations three orders of magnitude above their dissociation constants, and do not exist at negative concentrations (*Estrada et al., 2016*). Similarly, we can be generous about any possible cooperativities and say that $\omega_{bp}$ and $\omega_b$ have a similar bound between 0 and 1000, thus accounting for both positive and negative cooperativities. For this state with RNAP and one Bicoid molecule bound, we can say that

$$P_{1,1} = bp\omega_{bp} \tag{19}$$

which has bounds

$$0 < P_{1,1} < (1000)^2(1000) = 10^9 \tag{20}$$

and thus provide a bound for the possible values that the weight $P_{1,1}$ can take.

In general, this process can be applied to enforce bounds on any microstate weight $P_{r,n}$ through constraining of the possible values of $p$, $b$, $\omega_{bp}$, and $\omega_b$. As a result, the weight of a microstate with more Bicoid bound (i.e. higher values of $n$) will have a more generous dynamic range, due to the larger powers of $b$ and $\omega_b$. In this way, exploration of parameter space can be made more constrained by restricting the possible values of the microstate weights $P_{r,n}$. In addition, the mitotic repression term was constrained like in the thermodynamic MWC model, where $0 < t_{MitRep} < 10$.

As a result of these constraints, the region occupied by the generalized thermodynamic model in the $t_{on}$ offset and average $t_{on}$ delay space does not entirely include that of the thermodynamic MWC model, whose parameters were only constrained to be positive values (Appendix section 1.3).

Nevertheless, this model still fails to capture the delays observed in the *zelda*⁻ data (**Appendix 1—figure 9B**, yellow).

## 6.3 Extended generalized thermodynamic model with transcription factor binding in the inaccessible state

The generalized thermodynamic model (Appendix section 6.1) encompasses all possible thermodynamic models with up to twelve Bicoid-binding sites that can be bound in the accessible state. However, a potentially more general class of models involves those where Bicoid can also bind to the inaccessible state. For example, Bicoid action could conceivably result in some pioneering activity by directly binding to chromatin in the inaccessible state and facilitating RNAP binding and transcription. Here, we show that these models can be reformulated into the generalized thermodynamic model presented above.

If we allow for Bicoid to bind to any of the 12 binding sites in the inaccessible states, then we introduce $l$ new microstates with individual Boltzmann weights $P_l$, one for each Bicoid-bound inaccessible state, in addition to the unbound inaccessible state with weight $P_{inacc}$. Nevertheless, as long as the ensuing transcription rate of each Bicoid-bound inaccessible state is zero, then the net effect of these additional inaccessible states could simply be described by a single effective inaccessible state with Boltzmann weight $P'_{inacc} = P_{inacc} + \sum_l P_l$. The resulting state space exploration (Section 'No thermodynamic model can recapitulate the activation of hunchback by Bicoid alone' and **Figure 5C**, yellow), which explores the whole parameter space of reasonable values of $P_{inacc}$, would thus also capture the behavior of this single effective inaccessible state. As a result, models that consider the binding of Bicoid to the inaccessible states are contained within our generalized thermodynamics model.

## 6.4 Investigation of the failure of thermodynamic models

Here, we provide an intuitive explanation for why thermodynamic models fail to recapitulate the delay in $t_{on}$ for *zelda*⁻ embryos. The combination of the occupancy hypothesis and the assumption of separation of times scales described in Appendix section 6.5 imply that the rate of transcriptional initiation at any moment in time is an *instantaneous* readout of the Bicoid concentration at that time point. Thus, any thermodynamic model is *memoryless*. Intuitively, this means that a thermodynamic model requires transcription to begin as soon as the Bicoid concentration crosses a certain 'threshold' since time delays between input and output require some sense of memory. Examination of the dynamic measurements of MS2 output in *zelda*⁻ embryos reveals that no matter what 'threshold' concentration of Bicoid is assigned for the start of transcription, the model cannot simultaneously describe two values of $t_{on}$ corresponding to different positions along the anterior-posterior axis (**Appendix 1—figure 10A and B**).

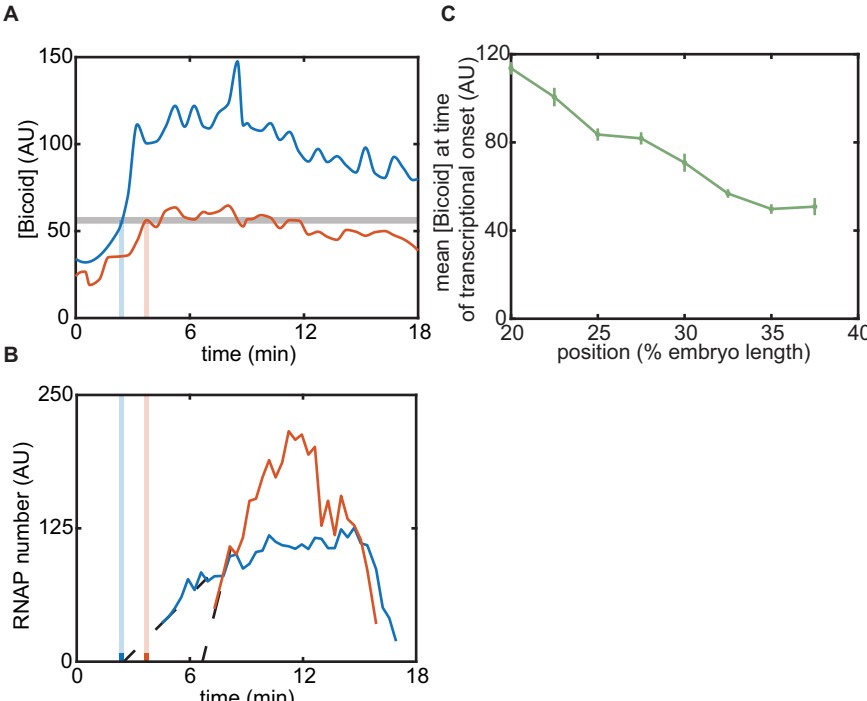

**Appendix 1—figure 10.** Intuition for failure of equilibrium models. (**A**) Mean Bicoid concentrations for two positions along the embryo (blue, red), with a 'threshold' chosen to to attempt to match the corresponding $t_{on}$ in (**B**). (**B**) MS2 fluorescence signal for the two positions shown in (**A**) for the $zelda^-$ experiment. Note that no single threshold value of Bicoid can match the timings in (**A**) with the transcriptional onset times in (**B**). (**C**) Mean Bicoid concentration at $t_{on}$ as a function of position for the $zelda^-$ data.

Another self-consistency check of a thermodynamic model is to examine the concentration of Bicoid at $t_{on}$ for various positions along the embryo. Due to the memoryless nature of thermal equilibrium, a valid thermodynamic model predicts that, at different positions along the embryo, $t_{on}$ will occur when Bicoid reaches the same threshold value. For the $zelda^-$ data, however, the level of Bicoid at each anterior-posterior position's $t_{on}$ value actually *decreases* with increasing $t_{on}$, suggesting the failure of the thermodynamic model (*Appendix 1—figure 10C*). Thus, the strong position-dependent delay in $t_{on}$ for the $zelda^-$ data cannot be explained by an instantaneous Bicoid readout mechanism.

More generally, the memoryless nature of thermodynamic models implies that, given any input-output function that increases monotonically with Bicoid and Zelda concentration, the ensuing onset time of transcription cannot be later than the time at which Bicoid or Zelda reach their maximal values. This is a reflection of a generic feature of thermodynamic models, namely that only instantaneous couplings in time can exist, and that time delays are impossible (*Coulon et al., 2013*; *Wong and Gunawardena, 2020*). By inspecting the nuclear concentrations of Bicoid and Zelda in *Appendix 1—figure 3*, we notice that times of maximal nuclear concentration for both transcription factors all occur around 4.5 min. This time is much earlier than the delayed transcriptional onsets exhibited in the $zelda^-$ data (*Figure 4D*, red points), providing further evidence for the unsuitability of thermodynamic models in describing the observed delay in the transcriptional onset time along the anterior-posterior axis of the embryo.

## 6.5 Re-examining thermodynamic models of transcriptional regulation

Thermodynamic models based on equilibrium statistical mechanics can be seen as limiting cases of more general kinetic models. For example, consider simple activation, where an activator whose concentration is modulated in time regulates transcription by binding to a single site (*Appendix 1—*

*figure 11*). In this generic model, the presence of activator can modulate the rates of activator and RNAP binding and unbinding through the parameters α, β, γ, and δ.

In order to reduce kinetic models to thermodynamic models where the probabilities of each state are dictated by Boltzmann weights such as those in *Figure 2A*, four conditions must be fulfilled. First, the rate of mRNA production must be linearly related to the probability of finding RNAP bound to the promoter (*Appendix 1—figure 11i*). This *occupancy hypothesis* is necessary for Appendix *Equation 2* to hold. Second, the time scales of binding and unbinding of RNAP and transcription factors must be much faster than the time scales of the concentration dynamics of these proteins (*Appendix 1—figure 11ii*). Third, these time scales must also be much faster than the rate of transcriptional initiation and mRNA production (*Appendix 1—figure 11iii*). Under these conditions of *separation of time scales*, the binding and unbinding of proteins quickly reaches steady state while the overall concentrations of these molecular players are modulated (*Segel and Slemrod, 1989*). Fourth, there must be no energy input into the system (*Appendix 1—figure 11iv*). This condition demands 'detailed balance' (*Vilar and Leibler, 2003*; *Ahsendorf et al., 2014*; *Hill, 1985*): the product of state transition rates in the clockwise direction over a closed loop is equal to the product going in the counterclockwise direction, a constraint known as the *cycle condition* (*Estrada et al., 2016*). In the case of *Appendix 1—figure 11*, this requirement implies that

$$k_P^{ON} \delta k_A^{ON} \beta k_P^{OFF} k_A^{OFF} = k_P^{OFF} k_A^{ON} \alpha k_P^{ON} \gamma k_A^{OFF}. \tag{21}$$

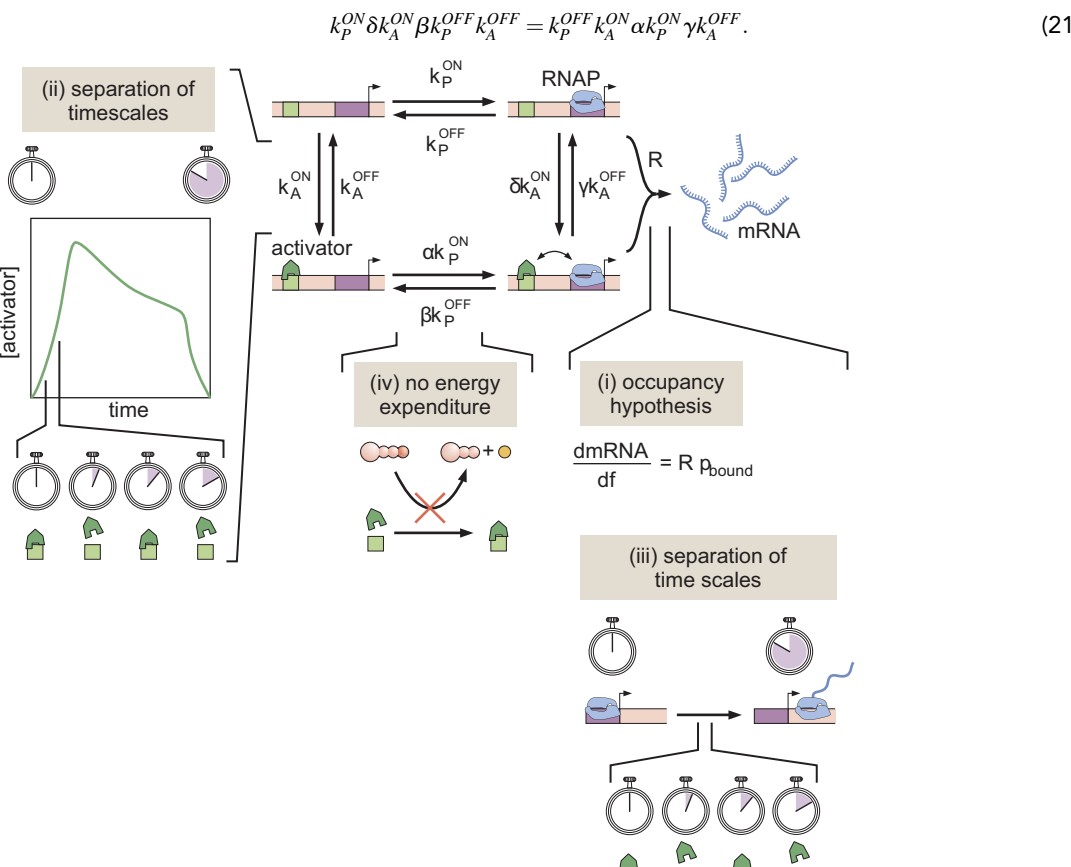

**Appendix 1—figure 11.** A simple kinetic model of transcriptional activation in which activator molecules influence RNAP binding kinetics. The assumptions that make it possible to turn this kinetic model into a thermodynamic one are (**i**) the occupancy hypothesis, (**ii, iii**) a separation of time scales between binding and unbinding rates, and activator and mRNA production dynamics, respectively, and (**iv**) no energy expenditure (detailed balance).

If these four conditions are met, then the system is effectively in equilibrium and the various binding states adopt probabilities that can be calculated using equilibrium statistical mechanics.

## 7. Non-equilibrium MWC model

## 7.1 Non-equilibrium MWC model

The non-equilibrium MWC model is an extension of the thermodynamic MWC model presented in Appendix section 1.2, where we now relax the assumption of separation of time scales (*Appendix 1—figure 11ii and iii*) and make it possible to assume, for example, that the system responds instantaneously to changes in activator concentration. Here, we explicitly simulate the full system of ordinary differential equations (ODEs) that describe the dynamics of the system out of steady state. Additionally, we allow for energy to be expended and thus do not enforce detailed balance through the cycle condition (*Appendix 1—figure 11iv*). We still employ a mitotic repression window term, before which no transcription is allowed.

We consider a generic model with $n$ Bicoid binding sites, and again ignore Zelda since we are only interested in recapitulating the $zelda^-$ mutant data. As a result, this new model has $n + 1$ total binding sites which, together with the closed chromatin state, results in a total of $2^{n+1} + 1 = N$ microstates. In the case of six Bicoid binding sites, this results in $N = 129$ total microstates. We assign each microstate $x_i$ a label $i$ and describe the transition rate from state $j$ to state $i$ using $k_{ij}$, where $i, j$ range from 0 to $N - 1$, inclusive.

In matrix notation, we write the system of ODEs as

$$\frac{d\vec{X}}{dt} = K\vec{X},$$ (22)

where $\vec{X}$ is a vector containing the fractional occupancy of each microstate $x_i$ and $K$ is a matrix containing all the transition rates $k_{ij}$. Normalizing such that the sum of all the components in the vector $\vec{X}$ is unity, we now have a vector representing the instantaneous probability of being in each microstate.

To relate the occupancies of the different states to the rate of transcriptional initiation, we retain the occupancy hypothesis presented earlier: that $p_{bound}$, the probability of being in a microstate with a bound RNAP molecule, is linearly related to the overall average transcriptional initiation rate that we determine from experimental measurements.

For this particular system, it is helpful to define an intuitive microstate labeling system. Because the relevant physical processes are the binding and unbinding of Bicoid and RNAP molecules, we can represent any microstate in binary form, where the total number of digits is the total number of binding sites $n + 1$, and each digit represents an individual binding site. Our convention is to assign the first digit to the promoter, and the subsequent ones to the Bicoid sites. By assigning 0 to an unbound site and one to a bound site, we can rewrite each unique microstate's label $i$ in binary form. For example, for a model with six Bicoid sites, the label for the microstate with no RNAP bound and the first two Bicoid sites occupied is represented with

$$i = \mathrm{bin}(0110000) = 48.$$ (23)

Here, bin() indicates taking the base 2 value of the binary label in the parentheses. The closed chromatin state is added manually and assigned to the last position in our binary label, $x_{N-1}$. This convention allows us to intuitively define each unique label for the system's microstates and provides a way to map the physical contents of a microstate with its associated label $i$.

In general, the overall transition matrix $K$ can be very complex. However, we benefit from the fact that the only non-zero transitions $k_{ij}$ are the ones that correspond to physical processes: modifying the open/closed chromatin state, and binding and unbinding of Bicoid or RNAP molecules. In this binary notation, these constraints imply that the only nonzero transitions are the ones that represent individual flips between 0 and 1, as well as between the open and closed states 0 and $N - 1$. The transition matrix $K$ is then easier to write, since it is clear from the binary representation which transitions must be nonzero. Finally, diagonal elements $k_{ii}$ are entirely constrained because they represent probability loss from a particular state $i$, and must be equal to the negative of the rest of the column $i$, such that the sum over each column in $K$ is zero.

Given that the Bicoid concentration changes as a function of time and that we assume first-order binding kinetics, whichever rates $k_{ij}$ correspond to Bicoid binding rates must be multiplied by this time-dependent nuclear concentration. In contrast, all off-rates are independent of Bicoid

concentration. To keep subsequent parameter exploration simple, we non-dimensionalized the Bicoid concentration by rescaling it by its approximate scale. This was achieved by dividing all Bicoid concentrations by the average Bicoid concentration, calculated by averaging the mean Bicoid nuclear fluorescence across all datasets, anterior-posterior positions, and time points, yielding approximately 35 arbitrary fluorescence units. Thus, all the transition rates $k_{ij}$ in the model here are expressed in units of inverse minutes.

To model transcription specifically, we assumed that at the beginning of the nuclear cycle, the system is in the closed chromatin state: $x_i(t = 0) = 0$ except for the closed chromatin state $x_{N-1}(t = 0) = 1$. We simulated the full trajectory of all the microstates $x_i$ over time by solving the system of ODEs given in *Equation 22*. Finally, we calculated $p_{bound}$ by summing the $x_i$'s that correspond to RNAP-bound states, and then computed the subsequent transcriptional initiation rate by multiplying $p_{bound}$ with the transcription rate $R$. Here, $R$ is the same $R_{max}$ as in Appendix sections 1.2 and 6.1 but again we do not constrain the model using $R_{min}$, just as in Appendix section 6.1.

*Appendix 1—figure 12A* shows an example of this model for a system with only one Bicoid binding site and no closed chromatin state, for simplicity, resulting in a four-state network. The binary indexing labels (shown beneath each state in light pink) can be converted into the base-10 labels (light teal) ranging from 0 to 3. The connection matrix for this system is

$$C = \begin{bmatrix} 0 & 1 & 1 & 0 \\ 1 & 0 & 0 & 1 \\ 1 & 0 & 0 & 1 \\ 0 & 1 & 1 & 0 \end{bmatrix} \tag{24}$$

and the corresponding transition rate matrix $K$ is

$$K = \begin{bmatrix} k_{00} & k_{01} & k_{02} & 0 \\ k_{10} & k_{11} & 0 & k_{13} \\ k_{20} & 0 & k_{22} & k_{23} \\ 0 & k_{31} & k_{32} & k_{33} \end{bmatrix}, \tag{25}$$

where, in this example, $k_{02}$ represents the transition rate from state $j$ to state $i$. The diagonal elements $k_{ii}$ are equal to the negative of the sum of the elements in the rest of the column in order to preserve conservation of probability. For example, $k_{00} = -(k_{10} + k_{20} + k_{30})$.

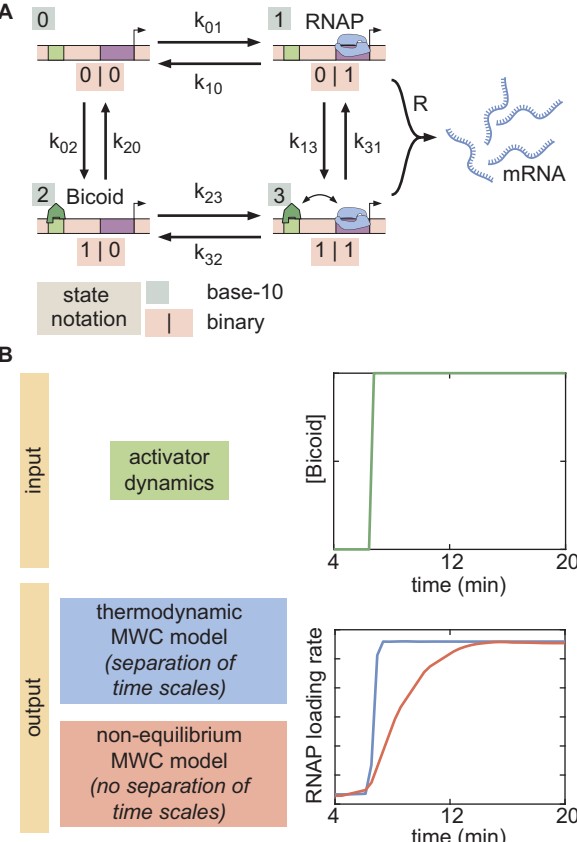

**Appendix 1—figure 12.** Example of a four-state time-dependent model with one Bicoid binding site and no closed chromatin state. (**A**) The binary label for each state (light pink) can be converted into a base-10 label for each state (light teal). The transition rates $k_{ij}$ are defined as the transition rate from state $i$ into state $j$ using this labeling system. (**B**) For an example input activator concentration temporal profile that is a step function, the time-dependent response is compared for the cases of separation of time scales and lack thereof. In the former, the transcriptional initiation rate responds instantaneously to the increase in activator input, while the response is slower in the latter.

With all this information in hand, we solve for the occupancy of each of the four states using the matrix ODE

$$
\begin{bmatrix} \frac{dx_0}{dt} \\ \frac{dx_1}{dt} \\ \frac{dx_2}{dt} \\ \frac{dx_3}{dt} \end{bmatrix} = \begin{bmatrix} k_{00} & k_{01} & k_{02} & 0 \\ k_{10} & k_{11} & 0 & k_{13} \\ k_{20} & 0 & k_{22} & k_{23} \\ 0 & k_{31} & k_{32} & k_{33} \end{bmatrix} \begin{bmatrix} x_0 \\ x_1 \\ x_2 \\ x_3 \end{bmatrix}. \tag{26}
$$

In this case, the occupancy hypothesis relates $p_{bound}$ to the overall transcription rate, resulting in

$$
\frac{d\mathrm{mRNA}}{dt} = R p_{bound} = R \frac{x_1 + x_3}{x_0 + x_1 + x_2 + x_3}. \tag{27}
$$

This model can produce time-dependent behavior not found in the thermodynamic models. *Appendix 1—figure 12B* contains an example of a hypothetical input Bicoid activator concentration that switches instantaneously from zero to a finite value. In the thermodynamic models, the predicted transcriptional initiation rate also responds instantaneously (*Appendix 1—figure 12B*, top). In contrast, for a suitable set of parameters, the non-equilibrium MWC model predicts a slow response over time (*Appendix 1—figure 12B*, bottom).

To produce a simulated MS2 fluorescence trace, the resulting rate of mRNA production is integrated over time using the same procedure (Appendix section 2.2) as the models presented in Appendix sections 1.2, 6.1, and 8.1. As with the thermodynamic MWC model, we allow for a time window of mitotic repression to account for the lack of transcription early in the nuclear cycle. Specifically, this was implemented by allowing the system to evolve over time, but fixing transcription to zero ($R = 0$) until after the mitotic repression time $t_{MitRep}$. An alternative formulation of the model, in which the whole system is frozen such that no transitions between states are allowed until after $t_{MitRep}$, is discussed below in Appendix section 7.3.

## 7.2 Non-equilibrium MWC model state space exploration

In the parameter exploration of this model (Appendix section 5.1), the transition rates $k_{ij}$ were constrained with minimum and maximum values of $k_{min} = 1$ and $k_{max} = 10^5$ respectively, in units of inverse minutes. These bounds were conservatively chosen using the following estimates. First, we estimate the values of the possible unbinding rates $k_{off}$. We assume that RNAP and Bicoid obey the same unbinding kinetics. Estimates of in vivo single-molecule binding kinetics inferred from *Mir et al., 2018* indicate that the lifetime of Bicoid on DNA is on the order of $3s^{-1}$. Second, we estimate the values of the possible on-rates $k_{on}$ using the classic Berg-Purcell equation for the case of a diffusion-limited binding to a perfectly absorbing spherical receptor (*Berg and Purcell, 1977*). In this case, the on-rate of molecule binding is given by

$$k_{on} = 4\pi Dac_o,$$  (28)

where $D$ is the diffusion coefficient of the molecule, $a$ is the estimated size of the spherical receptor, and $c_0$ is the background concentration of the molecular species. Since here we are talking about transcription factor binding to a Bicoid binding site, we assume $a$ to be on the order of 5 nm. We assume that RNAP and Bicoid obey the same diffusion characteristics, leading to a diffusion coefficient of approximately $0.3 \mu m^2 s^{-1}$ (*Gregor et al., 2007b*). Finally, Bicoid is is present at concentrations between 10 nM and 55 nM in the nucleus (*Gregor et al., 2007a*), and we assume that nuclear RNAP concentrations exist within the same range. Plugging these values into Appendix *Equation 28* yields estimates for the maximum and minimum on-rates:

$$
\begin{aligned}
k_{on}^{max} &\sim (4\pi)(0.3\ \mu m^2 s^{-1})(1\ \mu m)(55\ nM) \\
&\sim 0.5\ s^{-1} \sim 30\ min^{-1}.
\end{aligned}
$$

and

$$
\begin{aligned}
k_{on}^{min} &\sim (4\pi)(0.3\ \mu m^2 s^{-1})(1\ \mu m)(10\ nM) \\
&\sim 0.05\ s^{-1} \sim 3\ min^{-1}.
\end{aligned}
$$

Thus, our maximum and minimum transition rate bounds of $k_{min} = 1 min^{-1}$ and $k_{max} = 10^5 min^{-1}$ lie outside these estimated binding and unbinding rates. The mitotic repression term was constrained like in the thermodynamic MWC model, where $0 < t_{MitRep} < 10$.

One caveat of the state-space exploration approach is that the high dimensionality of the non-equilibrium MWC model prevented us from calculating the full state-space boundary using six Bicoid-binding sites. Due to computational costs, we were only able to accurately produce a state-space boundary for this model (Appendix section 7.1) using five Bicoid-binding sites. Running the exploration for a model with six Bicoid-binding sites took over 2 weeks on our own server, and the algorithm had not noticeably converged in the end.

The results of the state space exploration for the non-equilibrium MWC model using five Bicoid-binding sites resulted in larger average $t_{on}$ delays than the thermodynamic models (Appendix sections 1.2 and 6.1). However, this model, like those, failed to reproduce the delays observed in the *zelda⁻* data (*Appendix 1—figure 9B*, cyan).

Interestingly, the total areas covered by each non-equilibrium MWC model did not monotonically increase with Bicoid-binding site number (*Figure 6B*). This phenomenon where the state space of a model does not strictly increase with binding site number has been previously observed (*Estrada et al., 2016*) and the reason for this effect remains uncertain.

## 7.3 Alternative non-equilibrium MWC model with strong mitotic repression

In the main text, we entertained a non-equilibrium MWC model where mitotic repression blocks any productive transcription ($R = 0$) until the mitotic repression window $t_{MitRep}$ has passed. Before this time, in this model, the system can nevertheless transition through its different states over time.

In an alternate formulation of this non-equilibrium MWC model, we consider a form of mitotic repression that we call strong mitotic repression. Here, the system itself is frozen in the initial inaccessible state and not allowed to evolve until after $t_{MitRep}$. After $t_{MitRep}$, the system evolves through time according to the same rules as the original non-equilibrium MWC model.

Repeating the state space exploration for this model, for up to five Bicoid binding sites, yielded similar conclusions. Namely, the model could not describe the average delay and offset in transcriptional onset time in the absence of Zelda (*Appendix 1—figure 13*). The intuition behind this is that, while this stronger form of mitotic repression could potentially achieve longer delays, the crucial feature of the *zelda⁻* data is not merely a delayed transcription onset time, but a *position-dependent* delay that increases towards the posterior of the embryo. This stronger form of mitotic repression does not result in a mechanism capable of achieving such delay. In contrast, the final transcription-factor-driven model (Appendix section 8.1) does provide such a mechanism by coupling the inaccessible-to-accessible transition to the position-dependent Bicoid gradient.

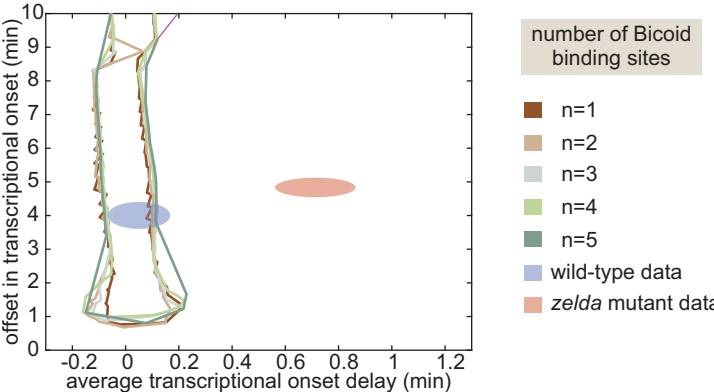

**Appendix 1—figure 13.** State space exploration for non-equilibrium MWC model with strong mitotic repression for up to five Bicoid-binding sites.

# 8. Transcription-factor-driven model of chromatin accessibility

## 8.1 Transcription-factor-driven model of chromatin accessibility

The transcription-factor-driven model of chromatin accessibility is a slight modification of the thermodynamic MWC model (Appendix section 1.2) that replaces the MWC mechanism of chromatin transitions with a direct driving action due to Bicoid and Zelda. Here, we retain the idea of inaccessible vs. accessible states, but no longer demand that these states be in thermodynamic equilibrium. Instead, the system begins in the inaccessible state and undergoes a series of $m$ identical, slow, and effectively irreversible transitions to the accessible state. Once these transitions into the accessible state occur, the system can rapidly and reversibly occupy all of its accessible microstates such that the probability of the system being in any of these microstates is described by thermodynamic equilibrium. The accessible states are governed by the same rules and parameters as the thermodynamic MWC model (Appendix section 1.2), albeit without the $\Delta\varepsilon_{chrom}$ parameter since now the transition from the inaccessible to accessible state is unidirectional.

We consider two possible contributions for these irreversible transitions: a Bicoid-dependent pathway and a Zelda-dependent pathway (*Appendix 1—figure 15A*, see Appendix section 8.2 for a discussion on this choice of parameterization). We assume the transition rates to be first-order in Bicoid and Zelda, respectively, such that

$$\pi_b = c_b[Bicoid] \tag{29}$$

and

$$\pi_z = c_z[Zelda]. \tag{30}$$

Here, $\pi_b$ is the Bicoid-dependent contribution to the transition rates and $\pi_z$ is the corresponding Zelda-dependent contribution. There are two input parameters $c_b$ and $c_z$ that give the relative speed of each transition rate contribution. The overall rate $\pi$ of each irreversible transition is given by the sum

$$\pi = \pi_b + \pi_z = c_b[Bicoid] + c_z[Zelda]. \tag{31}$$

Because the accessible states are in thermodynamic equilibrium with each other, we can effectively treat them as a single state and describe the entire system with $m+1$ states, corresponding to the inaccessible, intermediate, and accessible states. We label the inaccessible state with 0, the $m-1$ intermediate states with one through $m-1$, and the final accessible state with $m$. Thus, we describe the probability $p_i$ of the system being in the state $i$ with the probability vector $\vec{P}$

$$\vec{P} = \begin{bmatrix} p_0 \\ p_1 \\ ... \\ p_m \end{bmatrix}. \tag{32}$$

Calculating the overall RNAP loading rate then simply corresponds to rescaling $p_{bound}$ with the overall probability $p_m(t)$ of being in the accessible state:

$$\frac{\mathrm{dmRNA}}{\mathrm{d}t} = R\, p_{bound}\, p_m, \tag{33}$$

where $R$ is the same maximum rate used in Appendix section 1.2. Note that $p_m(t)$ is a time-dependent quantity that changes over time. To calculate $p_m(t)$, we solve the corresponding system of ODEs that describes the time evolution of $\vec{P}$.

$$\frac{d\vec{P}}{dt} = \Pi\vec{P}, \tag{34}$$

where $\Pi$ is the transition rate matrix describing the time evolution of the system. $\Pi$, by definition, is a square matrix with dimension $m+1$. Given the initial condition that the system begins in the inaccessible state

$$\vec{P} = \begin{bmatrix} 1 \\ 0 \\ ... \\ 0 \end{bmatrix} \tag{35}$$

the system of ODEs can be solved to find the probability of being in the accessible state $p_m(t)$. For example, for $m=3$ irreversible steps, $\Pi$ takes the form

$$\Pi = \begin{bmatrix} -\pi & 0 & 0 & 0 \\ \pi & -\pi & 0 & 0 \\ 0 & \pi & -\pi & 0 \\ 0 & 0 & \pi & 0 \end{bmatrix}, \tag{36}$$

where $\pi$ is given by Appendix *Equation 31*.

For simplicity, the time evolution of $\vec{P}$ was solved using MATLAB's ode15s solver.

With the probability $p_m(t)$ of the system being in the accessible state calculated, we now calculate the probability $p_{bound}$ of RNAP bound to the promoter in the accessible states, which lie in

thermodynamic equilibrium with each other. Because we now only have accessible states, the partition function is

$$Z = (1+z)^{10}(1+p+\sum_{j=0,1}\sum_{i=1}^{6}\binom{6}{i}b^i\omega_b^{i-1}p^j\omega_{bp}^{ij}), \tag{37}$$

where $z$, $p$, and $b$ correspond to the non-dimensionalized concentrations of Zelda, RNAP, and Bicoid, respectively, and $\omega_b$ and $\omega_{bp}$ are the cooperativities between Bicoid molecules and between Bicoid and RNAP, respectively. Thus, the overall transcriptional initiation rate is given by

$$
\begin{aligned}
Rate &= \frac{R}{Z}\left((1+z)^{10}p\left(1+\sum_{i=1}^{6}\binom{6}{i}b^i\omega_b^{i-1}\omega_{bp}^i\right)p_m\right. \\
&= R\frac{(p(1+\sum_{i=1}^{6}\binom{6}{i}b^i\omega_b^{i-1}\omega_{bp}^i))}{(1+p+\sum_{j=0,1}\sum_{i=1}^{6}\binom{6}{i}b^i\omega_b^{i-1}p^j\omega_{bp}^{ij})}p_m.
\end{aligned} \tag{38}
$$

Due to the lack of the inaccessible state in the partition function and because we assume that Zelda does not directly interact with Bicoid or RNAP, now the presence of Zelda mathematically separates out so that only Bicoid influences transcription. The calculation above is a standard equilibrium statistical mechanical calculation, except that we have weighted the final result with $p_m(t)$, the probability of being in the accessible states. The resulting rate is integrated to produce a simulated MS2 fluorescence trace using the same procedure (Appendix section 2.2) as the models presented in Appendix sections 1.2, 6.1, and 7.1.

Interestingly, we found that a mitotic repression term was not necessary to recapitulate the data, since the presence of intermediary states produced the necessary delay to explain the experimentally observed $t_{on}$ values in the data (**Figure 4D**, points).

In order to sufficiently explain the data, we found that a minimum of $m = 3$ irreversible steps was necessary. **Appendix 1—figure 14A and B** show the results of fitting this model to the observed rates of RNAP loading and $t_{on}$ for the wild-type and *zelda⁻* data, for increasing values of $m$ (wild-type results not shown, since all values of $m$ easily explained the wild-type data). We see that while lower values of $m$ do a poor job of recapitulating the data, once we reach $m = 3$ the model sufficiently predicts the experimental data within experimental error. For values of $m$ higher than 3, explanatory power increases marginally. Considering the parameter exploration of this model (Appendix section 8.3) highlights the necessity of having at least $m = 3$ steps.

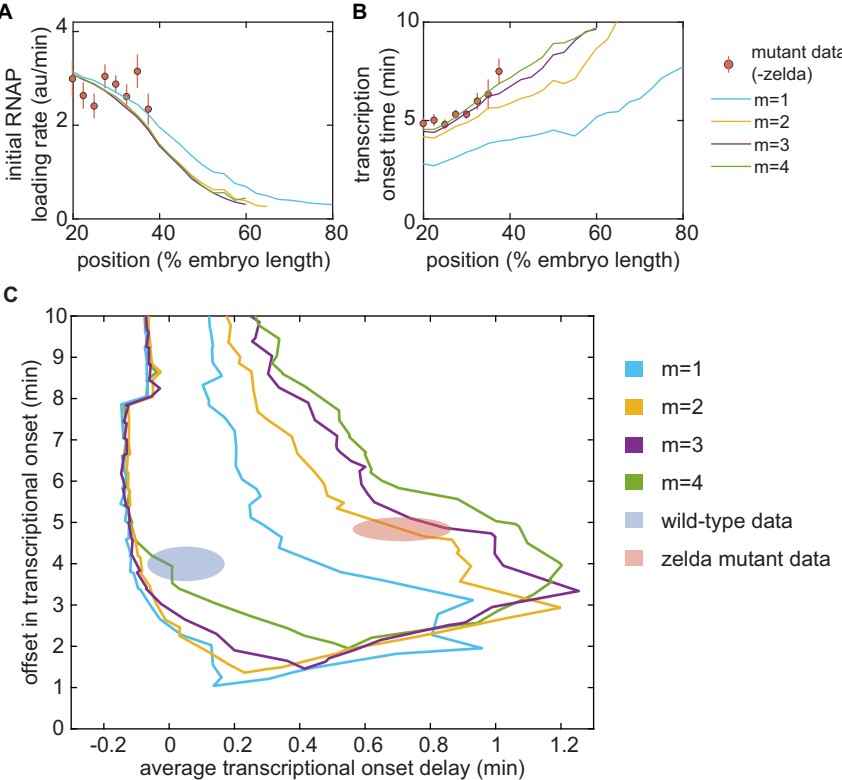

**Appendix 1—figure 14.** Testing the transcription-factor-driven model of chromatin accessibility. (**A**, **B**) Best-fit results of the transcription-factor-driven model to the mutant *zelda⁻* data. (**A**) initial RNAP loading rates, and (**B**) $t_{on}$, for varying numbers $m$ of transcriptionally silent states. (**C**) Parameter exploration in average $t_{on}$ delay and $t_{on}$ offset state space for increasing values of $m$.

## 8.2 Exploring alternatives to the additive transcription-factor-driven transition rate

In Appendix section 8.1, we defined the transition rate between the transcriptionally silent states in our transcription-factor-driven model of chromatin accessibility as

$$\pi = c_b[Bicoid] + c_z[Zelda]. \tag{39}$$

Here, we assumed that Zelda and Bicoid operate independently and in parallel to catalyze the transitions from the inaccessible to accessible state (*Appendix 1—figure 15A*). Our choice in using two independent Zelda- and Bicoid-mediated transitions was primarily motivated by the fact that, to our knowledge, no direct interactions between Bicoid and Zelda have been reported to date. However, this is not the only possible choice of model formulation. Here, we discuss and rule out two alternative mechanisms of Zelda- and Bicoid-mediated transitions from the inaccessible to accessible state.

As a first alternative, instead of an independent and additive mechanism, we could imagine a scenario where Bicoid and Zelda act simultaneously (*Appendix 1—figure 15B*). Here, each stochastic transition is given by

$$\pi = c[Bicoid][Zelda] \tag{40}$$

where $c$ is some constant with units of $[Bicoid]^{-1}[Zelda]^{-1}min^{-1}$.

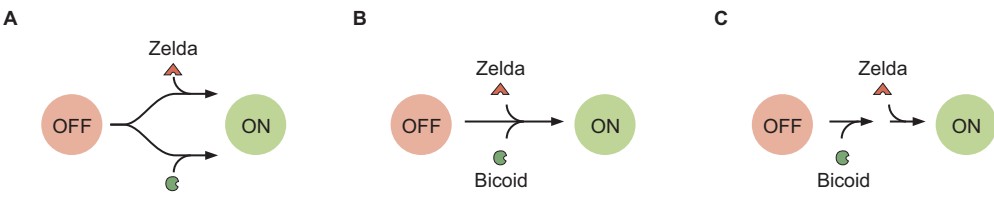

**Appendix 1—figure 15.** Different potential schemes of Bicoid- and Zelda-mediated transition into the accessible state, for a model with $m = 1$ transcriptionally silent state. (**A**) The model used in the main text, where Bicoid and Zelda provide independent pathways for chromatin to transition into the accessible state. (**B**) A scheme where Bicoid and Zelda act simultaneously on the transition. (**C**) A scheme where Bicoid acts first, and then Zelda, on the same pathway.

In a second alternative, Bicoid and Zelda could act sequentially. Here, each stochastic transition contains an intermediary state (*Appendix 1—figure 15C*). In this case, the transition rate will be dependent on Bicoid and Zelda such that

$$\pi \sim \frac{c_1[Bicoid]c_2[Zelda]}{c_2[Zelda] + c_1[Bicoid]}, \tag{41}$$

where $c_1$ and $c_2$ are constants with units of $[Bicoid]^{-1}min^{-1}$ and $[Zelda]^{-1}min^{-1}$.

One critical experimental observation is that transcription occurs even in the absence of Zelda, albeit at a delayed capacity. Since removing Zelda would set $\pi$ to zero in these alternative models, transcription would not occur at all, and so both of the proposed alternative mechanisms can be ruled out. More generally, the existence of transcription in the absence of Zelda requires that there must exist some independence between Bicoid-and-Zelda-mediated transitions from the OFF to the ON state. Otherwise, no transition, and hence no transcription, could occur in the absence of Zelda.

## 8.3 Transcription-factor-driven model of chromatin accessibility state space exploration

In the parameter exploration of this model (Appendix section 5.1), the parameters were constrained as

- $c_b > 0$
- $c_z > 0$.

The parameters shared with the thermodynamic MWC model retained the constraints described in Appendix section 1.3.

*Appendix 1—figure 14C* shows the state space explorations (see Appendix section 5.1) of this transcription-factor-driven model for increasing numbers of intermediate steps $m$. Not until $m = 3$ does the model explain the both the wild-type and *zelda$^-$* data, indicating that $m = 3$ is the minimum number of irreversible steps necessary. In the state space exploration shown in *Figure 7D* and *Appendix 1—figure 9*, the number of irreversible steps was fixed at $m = 3$.

Unlike the other models investigated (Appendix sections 1.2, 6.1, and 7.1), the transcription-factor-driven model of chromatin accessibility occupied a region in state space that encompassed both the wild-type and *zelda$^-$* data (*Appendix 1—figure 9*, purple).

