## [Decision Letter]

**Acceptance summary:**

This is a combined experimental-theoretical investigation of transcription in development using elegant experimental methods and a comprehensive theoretical analysis. The aim is to distinguish between computational models predicting the transcriptional output of a developmental gene from time-varying concentrations of transcription factors. Based on quantitative comparisons between live-cell imaging measurements of transcription in the fly embryo and the predictions from three classes of computational models, the authors conclude that the data can only be accounted for by a non-equilibrium scheme describing the progression towards 'accessible' chromatin state as a multi-step process driven by transcription factors Bicoid and pioneering factor Zelda.

**Decision letter after peer review:**

Thank you for submitting your article "Quantitative dissection of transcription in development suggests transcription factor-driven chromatin accessibility" for consideration by *eLife*. Your article has been reviewed by two peer reviewers, and the evaluation has been overseen by a Reviewing Editor and Naama Barkai as the Senior Editor. The following individual involved in review of your submission has agreed to reveal their identity: Anatoly Kolomeisky (Reviewer #2).

The reviewers have discussed the reviews with one another and the Reviewing Editor has drafted this decision to help you prepare a revised submission.

Both reviewers and the Reviewing editor found the paper very interesting, thought provoking, and very well written. We are convinced that a non-equilibrium approach to transcription is very important. Your experimental approach is elegant and the data are of high quality. There is need for additional experiments. However, reviewers raised concerns about the novelty of your findings, and questioned certain modelling hypotheses. These important issues must be satisfactorily addressed before a decision regarding publication can be made. In our opinion, this can be done without additional wet lab experiments, and within a reasonable timeframe.

As the editors have judged that your manuscript is of interest, but as described below that major revisions are required, in particular regarding modelling and discussion, before it is published, we would like to draw your attention to changes in our revision policy that we have made in response to COVID-19 (https://elifesciences.org/articles/57162). First, because many researchers have temporarily lost access to the labs, we will give authors as much time as they need to submit revised manuscripts. We are also offering, if you choose, to post the manuscript to bioRxiv (if it is not already there) along with this decision letter and a formal designation that the manuscript is "in revision at *eLife*". Please let us know if you would like to pursue this option. (If your work is more suitable for medRxiv, you will need to post the preprint yourself, as the mechanisms for us to do so are still in development.)

Summary:

This is a combined experimental-theoretical investigation of transcription in development using elegant experimental methods and a comprehensive theoretical analysis.

The aim is to distinguish between computational models predicting the transcriptional output of a developmental gene from time-varying concentrations of transcription factors. Based on quantitative comparisons between live-cell imaging measurements of transcription in the fly embryo and the predictions from three classes of computational models, the authors conclude that the data can only be accounted for by a non-equilibrium scheme describing the progression towards 'accessible' chromatin state as a multi-step process driven by transcription factors Bicoid and pioneering factor Zelda.

Essential revisions:

Novelty and context

1) The novelty of the work in comparison with exiting studies reaching similar conclusions must be clarified. In particular in comparison [Dufourt et al., 2018] in which a different gene is studied, but within the same organism, in the same nuclear cycle and controlled by the same pioneering factor Zelda.

2) The Discussion must be extended to put the results in a broader context of existing literature. Non-equilibrium view on transcriptional regulation has been discussed before. is presented as new, although these ideas have been around and studied for a number of years already. See for instance Coulon et al., 2013, and discuss the parallels with the studies cited therein, which points to the same mechanisms of step-wise, irreversible, TF-driven progression of chromatin state towards a Pol II-accessible configuration.

Dufourt et al., 2018; Fritzsch et al., 2018; Coulon et al., 2013.

Further data analysis

3) The data seem under-exploited: The authors have access to single-cell information but only analyze ensemble-average traces. This is surprising since the model they put forward is intrinsically stochastic (governed by a few rate-limiting steps). Signatures of such stochastic behavior should be clear in the data. For instance: Given that the model posits that transcription starts after m=5 equivalent first order reactions, then a key prediction is that the first appearance of the MS2 signal among all individual traces at a given position along the embryo should be Gamma-distributed (with shape parameter m=5). Even if the first appearance of the MS2 spot is a noisy measurement, this should be measurable (as in [Fritzsch et al., 2018] Figure 4A) and the authors already have this data at hand.

Concerns regarding the model

3) In the equilibrium MWC model (Figure 2A), Zelda, although being a pioneering factor, is not represented as such. Here it acts by binding the 'accessible' state, hence simply by keeping chromatin from closing. Instead, a pioneering factor should be able to bind in the 'inaccessible' state and make this state unfavorable (e.g. weight = exp(∆_chrom / kT) * z * w_{z,chrom} with w_{z,chrom } > 0). How would this more realistic scenario affect the conclusions?

4) The model (Equation 6) essentially assumes that bicoid and *zelda* act independently, and in an additive manner. Is this supported by data? Does transcription happen without bicoid, ,as Equation 6 suggests. If the two factors act simultaneously, one would expect the rate π=[Bicoid][Zelda]. If they act sequentially, one would expect π=[Zelda][Bicoid]c1[Zelda]+c2[Bicoid]. The use of Equation 6 must be justified better, and the difference between the different possible microscopic scenario should be checked numerically and discussed.

---

## [Author Response]

Essential revisions:Novelty and context1) The novelty of the work in comparison with exiting studies reaching similar conclusions must be clarified. In particular in comparison [Dufourt et al., 2018] in which a different gene is studied, but within the same organism, in the same nuclear cycle and controlled by the same pioneering factor Zelda.

We apologize for not having compared and contrasted our work with that of Dufourt et al. more explicitly. Indeed, they found that, just like with *hunchback*, the removal of Zelda resulted in a delay of transcriptional onset of the *snail* gene (and that the addition of Zelda binding sites led to an earlier onset of transcription). Dufourt *et al.* then proposed a model for explaining this delay by positing the existence of a series of identical, irreversible states that each promoter needs to traverse at the start of the nuclear cycle before transcription can ensue, and find the number of states and the transition rate most consistent with their data. Our transcription factor-driven model of chromatin accessibility (Figure 7) is very much an extension of the model proposed by Dufourt *et al.*

Despite the similarities and inspiration from this previous work, there are key differences between Dufourt *et al.* and our manuscript. First, we correlated output *hunchback* transcriptional activity—both onset time and the rate of mRNA production upon transcriptional onset—with the concentration dynamics of the Bicoid and Zelda inputs, a first in the fruit fly to our knowledge. In contrast, the work by Dufourt *et al.* only examined output transcriptional dynamics and did not visualize or incorporate into their model the dynamics of the various regulatory inputs to their *snail* reporter gene.

Second, these direct quantitative investigations of the relationship between input concentrations and output dynamics made it possible to investigate progressively more complex models of transcriptional regulation (equilibrium model MWC, non-equilibrium MWC model, and transcription factor-driven model of chromatin accessibility) that can predict both the transcriptional onset time and the rate of transcriptional initiation. Such systematic dissection ultimately brought us to the transcription factor-driven model of chromatin accessibility inspired by the work of Dufourt *et al.* In contrast, their work focused on using the single-cell distribution of transcriptional onset times to investigate how Zelda modulates the number of hypothesized transcriptionally silent states and their transition rates. While our work stands on their shoulders and we do not want to diminish their contributions by any means, we believe that our work is broader and more systematic.

Finally, we test the transcription factor-driven model of chromatin accessibility from a different perspective than that of Dufourt *et al.* They focus on examining the single-cell distribution of transcriptional onset times. A key assumption of their model is that the concentrations of input transcription factors and, hence, the number of states and their transition rates, do not change over time. However, as we show in our manuscript, these concentrations do change significantly. By considering concentration dynamics, we are able to examine the model under more realistic assumptions, and constrain it using the mean transcriptional onset time.

In summary, by performing dynamical measurements of both inputs and outputs, our work systematically explores a broad swath of models of transcriptional regulation, including that proposed by Dufourt et al. We then go beyond their work by dissecting the model from the perspective of transcription factor dynamics and complement their testing of the model by examining the mean predicted transcriptional onset time as well as the rate of transcriptional initiation. We apologize for our lack of scholarship in contrasting our work to that of Dufourt et al. and have now made this comparison more explicit in the Discussion, starting in the ninth paragraph.

2) The Discussion must be extended to put the results in a broader context of existing literature. Non-equilibrium view on transcriptional regulation has been discussed before. is presented as new, although these ideas have been around and studied for a number of years already. See for instance Coulon et al., 2013 and discuss the parallels with the studies cited therein, which points to the same mechanisms of step-wise, irreversible, TF-driven progression of chromatin state towards a Pol II-accessible configuration.Dufourt et al., 2018; Fritzsch et al., 2018; Coulon et al., 2013.

We thank the reviewer for pointing out our oversight in addressing similar studies of non-equilibrium transcriptional regulation. We now more clearly acknowledge these works in the Introduction (last paragraph). We have also added an additional paragraph in the Discussion (ninth paragraph) that contextualizes our findings in the broader setting of non-equilibrium transcriptional regulation. Specifically, we acknowledge that the idea of step-wise, transcription factor-driven transitions of chromatin into a transcribing state is not new, but do highlight the novelty of our work in quantitatively investigating the relationship between dynamic transcription factor concentrations and the rates of these transitions.

Further data analysis3) The data seem under-exploited: The authors have access to single-cell information but only analyze ensemble-average traces. This is surprising since the model they put forward is intrinsically stochastic (governed by a few rate-limiting steps). Signatures of such stochastic behavior should be clear in the data. For instance: Given that the model posits that transcription starts after m=5 equivalent first order reactions, then a key prediction is that the first appearance of the MS2 signal among all individual traces at a given position along the embryo should be Gamma-distributed (with shape parameter m=5). Even if the first appearance of the MS2 spot is a noisy measurement, this should be measurable (as in [Fritzsch et al., 2018] Figure 4A) and the authors already have this data at hand.

We would have loved to perform the analysis suggested by the reviewers in this manuscript. The analysis of transcription onset times using Gamma-distributed stochastic models, as the reviewer points out, is not uncommon in the field, and has been performed in numerous studies, for example in Dufourt et al., 2018, and Fritzsch et al., 2018. We acknowledge that by only examining the means of the transcription onset time, we are under-utilizing the data, which are taken at single-cell resolution. However, after carefully considering the possibility of extending our work to account for full distributions and capture stochastic effects of the model, we concluded that the results of such an analysis would be inconclusive given the current data and will require a future follow-up study that lies outside the scope of this paper. Below, we outline our reasoning in full detail. In essence, the explicit coupling of our model’s transition rates to dynamic transcription factor concentrations complicates the resultant theoretical analysis, as well as drastically reduces the statistical size of our datasets to a level where meaningful conclusions about higher moments of the distribution are difficult to draw.

One key difference between our study and comparable works is that our model explicitly accounts for how transcription factor concentrations modulate the transition rates between the silent states that precede transcriptional onset. Specifically, we posit that Bicoid and Zelda concentrations dictate these transition rates. Thus, in order for our analysis to be meaningful, it must take into account transcription factor concentrations. For example, in the context of explaining the *zelda^-^* mutant data, our analysis must take the spatiotemporal modulation of the Bicoid concentration gradient into account. As a result, for each position along the length of the embryo, a different dynamic Bicoid profile will cause the transition rates in the stochastic model to be time-varying. Thus, the resultant distribution of transcription onset times will no longer be a simple Gamma-distribution that reflects the correct shape parameter.

Author response image 1 demonstrates how acknowledging Bicoid concentration dynamics significantly changes the shape of the single-cell transcriptional onset times distribution and its interpretation. Here, we took the transcription factor-driven model used in the main text and numerically simulated the *zelda^-^* transcription onset times for two cases, (1) using the full dynamic Bicoid input at each embryo position, and (2) using a static Bicoid input obtained by averaging the dynamic Bicoid concentration in the first 8 minutes of the nuclear cycle (author response image 1A). For model parameters, we chose the same best-fit parameters used in Figure 7 of the main text. author response image 1B shows the resulting predicted distributions of transcription onset times for the dynamic and static cases (red and blue, respectively) for a model with k=5 steps. The figure reveals that Bicoid dynamics cause the overall distribution to shift to the right. Nevertheless, both distributions still fit well to Gamma-distributions (purple and yellow curves).

Repeating these simulations for models ranging from k=1 to k=6 steps at a particular embryo position and fitting Gamma-distributions to both static and dynamic cases shows that, as expected, the static input model yields Gamma-distribution fits that match the underlying truth (Author response image 1, blue), where the effective shape parameter (i.e. number of steps) matches to the correct value. However, we see that the dynamic input model systematically yields effective shape parameters that are higher than the actual number of states assumed in our simulation (Author response image 1, red). Thus, the dynamic nature of the Bicoid concentration leads to a higher effective shape parameter. This trend holds for various embryo positions as well. Author response image 1 shows the k=5 steps model for various positions along the embryo, demonstrating that the dynamic input systematically produces distributions with higher effective shape parameters than the static input.

While the need to consider transcription factor dynamics complicates the analysis from the standpoint of analytical expressions for the probability distribution, we can still carry forward stochastic numerical simulations of transcription onset times taking into account Bicoid dynamics. However, after doing this, we encountered a severe experimental barrier, namely that the statistical size of each sample (i.e. each position along the embryo with unique Bicoid concentration dynamics) was very low.

In an attempt to analyze the single-cell results, we followed the analysis carried out by Dufourt et al., 2018. Specifically, we calculated the effective number of intermediary steps at each embryo position, defined as the square of the mean transcription onset time divided by the variance. For a true Gamma-distribution, this yields the shape parameter (the number of intermediary steps). Here, due to the dynamic nature of Bicoid, we interpreted this as an effective number of intermediary steps to be compared with the stochastic simulation of the model.

To calculate the transcription onset time for a single cell, we invoked the technique used in the main text consisting of fitting a straight line to the initial rise in the MS2 signal, thus defining the x-intercept as the onset time (Author response image 1). After grouping the single cell fits into bins along embryo position, we calculated the effective number of intermediary steps (Author response image 1, datapoints). We then simulated N=500 cells for the transcription factor-driven model ranging from k=1 to k=6 steps between the inaccessible and accessible state, using the dynamic Bicoid concentration at each embryo position and the parameters used in Figure 7 of the main text. From these single-cell simulations, we calculated the effective number of intermediary steps (Author response image 1, black dashed line). While the k=1 and k=2 models appear to clearly fall outside the data, it is difficult to distinguish between models with higher step numbers, simply because the data were sparse enough that the bootstrapped error in the effective step number was too high. For example, at 20% along the embryo, the effective step number in the data possessed a mean of approximately 7, with a bootstrapped standard error of almost 5. This high error prevents us from drawing precise conclusions about the numbers of intermediary steps that best fit the data.

The reason for these high uncertainties is the low statistical size of the dataset (Author response image 1, bars). At most, a single position bin possessed ~25 cells’ worth of data. For this reason, we did not place much faith into the analysis of higher moments of the transcription onset times. This low statistical size, in our opinion, makes such a single-cell analysis inconclusive in the context of this work and calls for a new experiment where the number of data sets is dramatically increased. However, germline clone *zelda^-^* mutant experiments are incredibly time-consuming to perform. Simply doubling the amount of data would take >2 months of full-time imaging and, we believe, would still be insufficient for rigorous statistical analysis. Thus, we hope that the reviewer will agree that, while exciting and certainly a next step, a thorough analysis of single-cell distributions of transcription onset times will require a more efficient experimental scheme that lies outside of this work’s scope. A more detailed discussion of the potential for future insights to be gained from examining single-cell distributions can be found in the twelfth paragraph of the Discussion. Regardless, though we believe this falls outside the scope of our current work, we would be happy to incorporate the analysis described here in the Appendix if the reviewers deem it necessary.

**Author response image 1. respfig1:** Single cell analysis of transcription onset times for *zelda^-^* data. (**A**) Comparison of Bicoid input profiles for dynamic (red) or static (blue) cases at 22.5% along the embryo length. Static profile was obtained for each position along the embryo length by computing the mean of the dynamic profile between 0 and 8 minutes. (**B**) Simulated transcription onset times of the transcription factor-driven model for k=5 intermediary states using dynamic (red) or static (blue) Bicoid inputs, along with Gamma-distribution fits (purple and yellow, respectively). (**C**) Effective shape parameter for simulated transcription onset times of the transcription factor-driven model for k=1 to k=6 intermediary transitions using dynamic (red) or static (blue) Bicoid inputs. (**D**) Effective shape parameter for k=5 intermediary transitions as a function of embryo position for dynamic (red) or static (blue) Bicoid inputs. (**E**) Extraction of single-cell transcription onset times using linear fit (red) from a single-cell MS2 signal (black). (**F**) Effective step number computed from single-cell transcription onset times as a function of embryo position, for *zelda^-^* data (red) as well as for predictions of the transcription factor-driven model with k=1 to k=6 intermediary steps and dynamic Bicoid inputs (black dashed lines). The total number of cells per position bin is shown in light blue bars, for a total of n=12 datasets. (C, D, error bars reflect 95% confidence interval; F, error bars reflect bootstrapped standard error).

Concerns regarding the model3) In the equilibrium MWC model (Figure 2A), Zelda, although being a pioneering factor, is not represented as such. Here it acts by binding the 'accessible' state, hence simply by keeping chromatin from closing. Instead, a pioneering factor should be able to bind in the 'inaccessible' state and make this state unfavorable (e.g. weight = exp(∆_chrom / kT) * z * w_{z,chrom} with w_{z,chrom } > 0). How would this more realistic scenario affect the conclusions?

We agree with the reviewer that our equilibrium MWC model does not encompass all the possible mechanisms of Zelda as a pioneering factor. However, it is important to note that the initial equilibrium MWC model was sufficient to explain our wild-type data, and it was only the *zelda^-^* mutant data that could not be described by any of our thermodynamic models. Thus, it was actually the removal of Zelda that seemed to demonstrate the failure of the equilibrium paradigm. As a result, the specific theoretical action of Zelda will not affect the conclusion that the thermodynamic framework fails to explain the key result that removing Zelda drastically delays onset times of transcription in our system (Figure 4D, red points).

Nevertheless, one could extend this reasoning to wonder if a potential pioneering activity of Bicoid would change the conclusions. After pursuing this line of thought, we concluded that this would also not change our conclusions, since such a mechanism is largely captured in our generalized thermodynamic model, which only included Bicoid and RNAP binding to attempt to explain the *zelda^-^* mutant data (subsection “No thermodynamic model can recapitulate the activation of hunchback by Bicoid alone” and Appendix subsection “Generalized thermodynamic model”). Here, the Boltzmann weights for each microstate were left as an arbitrary coefficient, with the weight of the inaccessible state denoted by Pinacc. Extending the model to allow Bicoid to bind to the inaccessible state would introduce l new microstates with individual Boltzmann weights Pl, one for each Bicoid-bound inaccessible state. Nevertheless, as long as the ensuing transcription rate of each inaccessible state with bound Bicoid molecules is zero, an assumption that we believe is reasonable, then the net effect of these additional inaccessible states could simply be described by a single effective inaccessible state with Boltzmann weight Pinacc′=Pinacc+∑lPl. The resulting state space exploration (subsection “No thermodynamic model can recapitulate the activation of hunchback by Bicoid alone” and Figure 5C), which explores the whole parameter space of reasonable values of Pinacc, would thus also capture the behavior of this single effective inaccessible state.

Finally, as pointed out in the reviewer comment #9, the constrained dynamics of both Bicoid and Zelda, which reach a peak value around 4.5 min, imply that any thermodynamic model will not be able to produce onset times significantly later than that, due to the separation of time scales assumption. Thus, our main conclusion that the thermodynamic framework is insufficient to explain the *zelda^-^* data is a rather general point that is independent of the specific implementation of the model. We have incorporated these ideas as a new section in the Appendix (Appendix subsection “Extended generalized thermodynamic model with transcription factor binding in the inaccessible state”) and referenced it in the main text (subsection “No thermodynamic model can recapitulate the activation of hunchback by Bicoid alone”).

4) The model (Equation 6) essentially assumes that bicoid and zelda act independently, and in an additive manner. Is this supported by data? Does transcription happen without bicoid, ,as Equation 6 suggests. If the two factors act simultaneously, one would expect the rate π=[Bicoid][Zelda]. If they act sequentially, one would expect π=[Zelda][Bicoid]c1[Zelda]+c2[Bicoid]. The use of Equation 6 must be justified better, and the difference between the different possible microscopic scenario should be checked numerically and discussed.

First, we would like to clarify a possible misunderstanding. Equation 6 only describes the rate of *transitioning* between the silent states preceding transcription and the transcriptionally active state, and holds no information about the ensuing rate of transcription, which is then given by a standard thermodynamic model involving Bicoid (Figure 7A). Equation 6 alone does not suggest that transcription happens without Bicoid, only that the system can transition to the ON state with Zelda alone. Once in the ON state, the absence of Bicoid will thus result in merely whatever basal transcriptional activity is supported by the presence of RNAP.

Second, the reviewer brings up salient points about the specific form of Equation 6. We would first like to stress that the main goal of this section of the paper was to propose a simple non-equilibrium extension to the thermodynamic framework that could explain all aspects of the experimental data. To our knowledge, there have been no works suggesting any direct interactions between Bicoid and Zelda. In the absence of evidence, we thus chose a model where Bicoid and Zelda acted independently and additively. A more comprehensive comparison of various non-equilibrium models is a project that we believe to be beyond the scope of this paper. Indeed, the exploration of these models still remains challenging even in the simpler case of bacterial transcriptional regulation as exemplified by the work by Hammar et al., 2014, and the recent bioRxiv preprint of Morrison et al. (2020). Nevertheless, we can at least comment on the two alternative mechanisms proposed in this comment.

If Bicoid and Zelda act simultaneously, then one would indeed expect Equation 6 to have the form π=[Bicoid][Zelda]. However, one critical experimental fact is that transcription occurs even in the absence of Zelda, albeit at a delayed capacity. Thus, this purely simultaneous action mechanism can be ruled out, since removing Zelda would set all transition rates π to zero. Similarly, this rules out a sequential mechanism where π=[Bicoid][Zelda]c1[Bicoid]+c2[Zelda], since removing Zelda would similarly forbid any transitions from the OFF to ON state.

More generally, the existence of transcription in the absence of Zelda implies that there must exist some independence between Bicoid-and-Zelda-mediated transitions from the OFF to ON state. If Bicoid and Zelda did not work in parallel to catalyze the transition into the ON state, transcription could not occur in the absence of Zelda. To discuss this point explicitly, we have added this argument as an appendix section in Appendix subsection “Exploring alternatives to the additive transcription factor-driven transition rate” and Appendix—figure 15, and have referenced it in the subsection “Transcription factor-driven chromatin accessibility can capture all aspects of the data”.